EMBO
Molecular Medicine

# Modulating CCTG repeat expansion toxicity in DM2 Drosophila model through *TDP1* inhibition

Yingbao Zhu [iD][1], Shengwei Xiao[1], Xinxin Guan[1], Haitao Deng[1], Liqiang Ai[1], Kaijing Fan[1], Jin Xue[1], Guangxu Li[1], Xiaoxue Bi[1], Qiao Xiao[1], Yuanjiang Huang[1], Lin Jiang[1], Wen Huang[1,2,3], Peng Jin [iD][4✉] & Ranhui Duan [iD][1,2,3✉]

## Abstract

**Myotonic dystrophy type 2 (DM2), caused by CCTG repeat expansion, is a common adult-onset disorder characterized by myotonia and progressive muscle degeneration with no effective treatment. Here, we identified Tyrosyl-DNA phosphodiesterase 1 (*TDP1*) as a novel modifier for DM2 therapeutic intervention through a high-throughput chemical screening of 2160 compounds. Moreover, we detailed how both genetic and pharmacological inhibition of *TDP1* translates to a cascade of beneficial effects, including improved motor functions, amelioration of progressive muscle degeneration, repair of muscle fiber damage, and normalization of aberrant molecular pathology. Remarkably, the TDP1 inhibition led to substantial CCTG repeat contractions, a mechanism that underlies the observed muscle toxicity and neurodegeneration. Our results highlighted the potential of *TDP1* as a molecular target for addressing the complex interplay between repeat expansions and neuromuscular degeneration in DM2, hinting at broader applicability in a spectrum of repeat expansion disorders.**

**Keywords**  DM2; CCTG Repeat Expansion; Chemical Screen; *TDP1*; Repeat Instability
**Subject Categories** Genetics, Gene Therapy & Genetic Disease; Musculoskeletal System

## Introduction

Myotonic dystrophy is the most common form of muscular dystrophy in adults, affecting numerous tissues but predominantly skeletal muscle, with an incidence of 1 in 8000 worldwide (Cho and Tapscott, 2007; Kumar et al, 2013; Timchenko, 2013). The disorder encompasses two genetically distinct types: myotonic dystrophy type 1 (DM1, OMIM 160900) and myotonic dystrophy type 2 (DM2, OMIM 602668), attributable to CTG expansion in 3'UTR of the *DMPK* gene and CCTG repeats in the first intron of the *CNBP* gene, respectively (Kumar et al, 2013). The range of expanded CCTG allele sizes is extremely broad, from 75 to 11,000 CCTG repeats in DM2 (Liquori et al, 2001). Mutant RNAs with extensive CUG or CCUG repeats disrupt RNA metabolism and cause DM symptoms by sequestering MBNL proteins in nuclear RNA foci, leading to abnormal embryonic splicing patterns (Thomas et al, 2018).

DM1 and DM2 share a common mechanism involving toxic RNA repeats, however, DM1 and DM2 differ in certain pathogenic mechanisms, including the muscle groups affected and the absence of a congenital form in DM2. In addition, DM1 pathogenesis involves the hyperactivation of the GSK3β-CUGBP1 pathway, a mechanism not shared by DM2 (Meola, 2020). Differential interaction with RNA-binding proteins, such as RbFOX1's specific affinity for expanded CCUG but not CUG repeats, partially explain these disparities (Sellier et al, 2018). These distinctions suggest that treatments developed for DM1 may not be directly translatable to DM2. The majority of therapeutic studies in DM are currently focused on DM1, leaving DM2 in need of more attention. Clinical trials for DM1 include Tideglusib targeting GSK3β pathway (phase III), Metformin for the splicing defect (phase III), Mexiletine for myotonia (phase III), and antisense oligonucleotides like AOC 1001 (phase III), and NTC0231 targeting repeat secondary structures to displace sequestered proteins (IND enabling phase) (Heatwole et al, 2021; Horrigan et al, 2020; Pascual-Gilabert et al, 2023). A more comprehensive list of DM1 drugs currently in development can be found in the review by Pascual-Gilabert et al (Pascual-Gilabert et al, 2023). The difference in toxic mechanisms underlying DM1 and DM2 render many established DM1 treatment strategies with limited effectiveness for DM2 (Yenigun et al, 2017). The therapeutic options for DM2 remain symptomatic, necessitating extensive efforts to discover new treatments.

DM2 Drosophila models, expressing uninterrupted CCTG repeats ranging from 16 to 720, were developed by Bonini et al (Yu et al, 2015). The DM2 models recapitulate key features observed in patients, such as RNA repeat-induced structure disruption in the compound eye, formation of ribonuclear foci, and splicing alterations. In 2017, a DM2 Drosophila model expressing 1100 CCTG repeats was constructed, with histological

[1]Furong Laboratory, Center for Medical Genetics, School of Life Sciences, Central South University, Changsha, Hunan, China. [2]Hunan Key Laboratory of Medical Genetics, Changsha, Hunan, China. [3]Hunan Key Laboratory of Animal Models for Human Diseases, Changsha, Hunan, China. [4]Department of Human Genetics, Emory University School of Medicine, Atlanta, GA 30322, USA. ✉E-mail: peng.jin@emory.edu; duanranhui@sklmg.edu.cn

analysis revealing severe muscle reduction and cardiac dysfunction (Cerro-Herreros et al, 2017). A Drosophila model with 106 CCTG repeats was developed by Bergmann et al, which displayed disruption of external eye morphology and an apoptotic response but lacked a muscle phenotype (Yenigun et al, 2017). *Drosophila melanogaster*, as a multicellular organism, offers a powerful platform for modeling complex phenotypes, including neurodegenerative and muscular toxicities. Drosophila models for drug screening have been utilized to successfully investigate the pathological effects of repeat expansions (Chang et al, 2008; Garcia-Lopez et al, 2011). In this study, we conducted a large-scale unbiased chemical screen and identified the Tyrosyl-DNA phosphodiesterase 1 (*TDP1*) as a modifier for muscle toxicity of expanded CCUG repeats. *TDP1* is involved in resolving stalled topoisomerase 1 (TOP1) cleavage complex (TOPcc), consequently facilitating the initiation of the DNA repair process during replication and transcription (Bhattacharjee et al, 2022; Pommier et al, 2014). To mitigate the DNA supercoils in replication and transcription, TOP1 cleaves and reseals DNA molecules, forming a transient TOPcc (Pommier et al, 2016). As a DNA end-processing enzyme, TDP1 excises stalled TOP1cc, thus preventing DNA damage accumulation and ensuring genomic stability (Comeaux and van Waardenburg, 2014).

Our chemical screening showed that inhibition of *TDP1* significantly alleviates the lethal phenotype observed in DM2 Drosophila models. Moreover, we extensively examined the muscular toxicity characteristic of DM2, assessing the impact of

*TDP1* suppression on various aspects, including motor function, muscle morphology, muscle fiber, and molecular pathologies. Given the function of TDP1, the effect of TDP1 on the number of CCTG repeats was observed, which underpinned the improvements in muscle toxicity and neurodegeneration in DM2 models. These findings suggested that the inhibition of TDP1 represents a promising therapeutic strategy for expanded CCTG repeats.

# Results

## Development of chemical screen for modifier of DM2 muscle toxicity

To find small molecules capable of mitigating muscle toxicity, we conducted a chemical screen of 2160 compounds from the Johns Hopkins Clinical Compound Library (JHCCL) of FDA-approved or approvable drugs (Chong et al, 2006; Chong and Sullivan, 2007; Lin et al, 2020; Qurashi et al, 2012). Given that muscle degeneration is the prominent phenotype in DM2 patients, muscular toxicity was used as a basis for our screen. The expression of 720 CCTG repeats across all somatic muscles, under the control of the muscle-expressed *24B-Gal4* driver, led to pupal lethality. A total of 140 compounds were identified to rescue the viability of *24B-Gal4 > UAS-(CCTG)₇₂₀*, resulting in the emergence of adults (Fig. 1A). To verify the efficacy of these compounds without their effects on muscle development, another

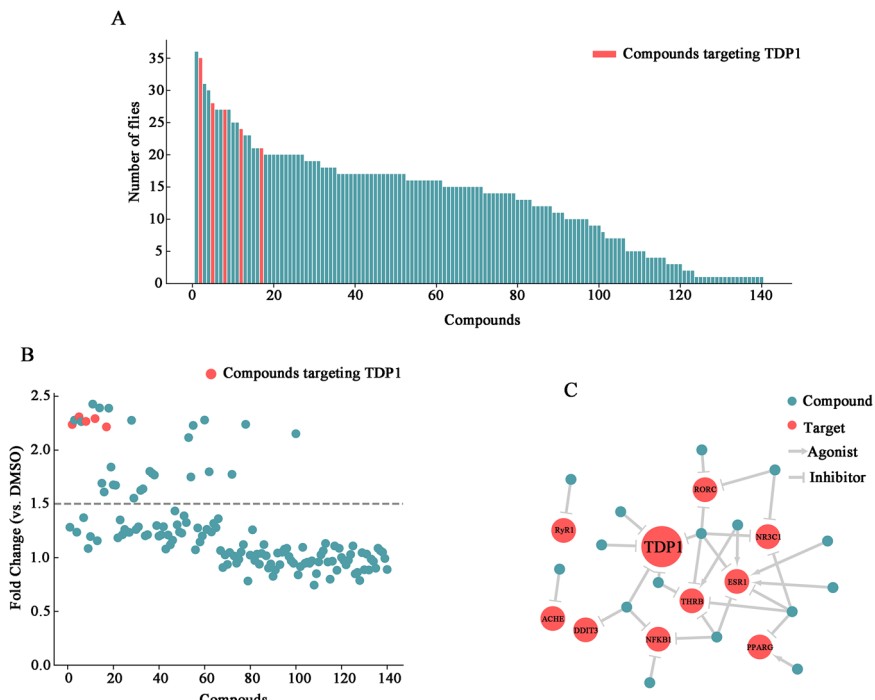

**Figure 1. Development of chemical screen for identifying modulators of DM2 muscle toxicity.**

(A) The primary screen result of 2160 compounds. On the *Y* axis reported the number of *24B-Gal4 > UAS-(CCTG)₇₂₀* that survive to adult. (B) The validation results of 140 selected compounds identified from the primary screen. On the *Y* axis reported the fold change of survival compared with DMSO-treated *Mef2-Gal4 > UAS-(CCTG)₇₂₀*. The dotted line represents the threshold (1.5-fold change). Data show the mean value from three replicates. (C) Network diagram of 16 compounds with specific target information and target interactions. Source data are available online for this figure.

**Table 1. List of the compounds targeting TDP1 selected from the confirmatory screening.**

| No. | PubChem CID | Compound name | 2D structure | Primary screen (number of flies) | Second screen (fold change vs. DMSO) |
|---|---|---|---|---|---|
| 1 | 37907 | Climbazole | | 35 | 2.23 |
| 2 | 6708755 | Deoxysappanone B 7,3'-dimethyl ether acetate | | 28 | 2.31 |
| 3 | 65464 | Benzydamine hydrochloride | | 27 | 2.26 |
| 4 | 64971 | Betulinic acid | | 24 | 2.29 |
| 5 | 8813 | Pentamidine isethionate | | 21 | 2.21 |

Data show the mean value from three replicates.

muscle-specific *Mef2-Gal4* driver was utilized. The expression of 720 CCTG repeats across late-stage muscle development and mature muscle, under the control of the *Mef2-Gal4* driver, resulted in semi-lethality. The 31 screened compounds showed a significant increase ($\geq$ 1.5-fold change compared to DMSO) in survival rates (Fig. 1B). Further analysis, referencing the PubChem Compound database, narrowed these to 16 with target information (Fig. 1C; Table EV1). While ESR1 was pinpointed by six compounds, it was excluded due to the inconsistent activity types among these compounds, with three acting as agonists and the others as inhibitors. The inhibition of TDP1 by five compounds, at the top of the screening list, exhibited the greatest potential for modifying muscle toxicity (Table 1).

## *TDP1/gkt* knockdown alleviated muscle atrophy and neurodegeneration in DM2 flies

TDP1 is evolutionarily conserved across species and is orthologous to Drosophila's glaikit (gkt), with 53% similarity between the two proteins (Barclay et al, 2014; Guo et al, 2014). To assess TDP1's role in DM2 pathology, we examined the impact of CCTG repeat expansion on *gkt* expression, as well as the effects of *TDP1/gkt* knockdown on locomotion, progressive muscle degeneration, and

neurodegeneration in DM2 Drosophila model. There is no difference in *gkt* expression between (CCTG)$_{720}$ and (CCTG)$_{16}$ flies (Appendix Fig. S1A–D). Using muscle-specific *Mef2-Gal4* driver, DM2 flies showed a 40% reduction in locomotor activity from 15 to 25-days-old compared to controls (Fig. 2A,C). Knockdown of *gkt*, through two RNAi lines and one mutant line, restored locomotor activity, climbing flight ability in DM2 files, while *gkt* knockdown alone did not produce any noticeable phenotype in control flies (Figs. 2B,C and EV1A–C,I; Appendix Fig. S2A,B,D; Movie EV1). Moreover, indirect flight muscles (IFM) of flies with (CCTG)$_{720}$ repeats experienced progressive degeneration (Fig. EV1D,E). At 1-day-old, a noticeable tear and roughly 10% reduction in muscle area were observed in DM2 flies compared to control. By 7-days-old, the DM2 flies showed more severe muscle damage, with visible holes and approximately a 15% decrease in muscle area, leaving few muscles intact. By 15-days-old, absolute damage, including large holes and complete fragmentation of the muscle mass, was apparent, alongside an approximate 30% reduction in muscle area. *Gkt* knockdown mitigated the degenerative phenotypes in IFM, resulting in a ~25% increase in muscle area compared to DM2 flies at 15-days-old (Figs. 2D,E and EV1D,E; Appendix Fig. S2C).

In addition to muscle degeneration, we investigated neurodegenerative phenotypes within the compound eyes. The expression

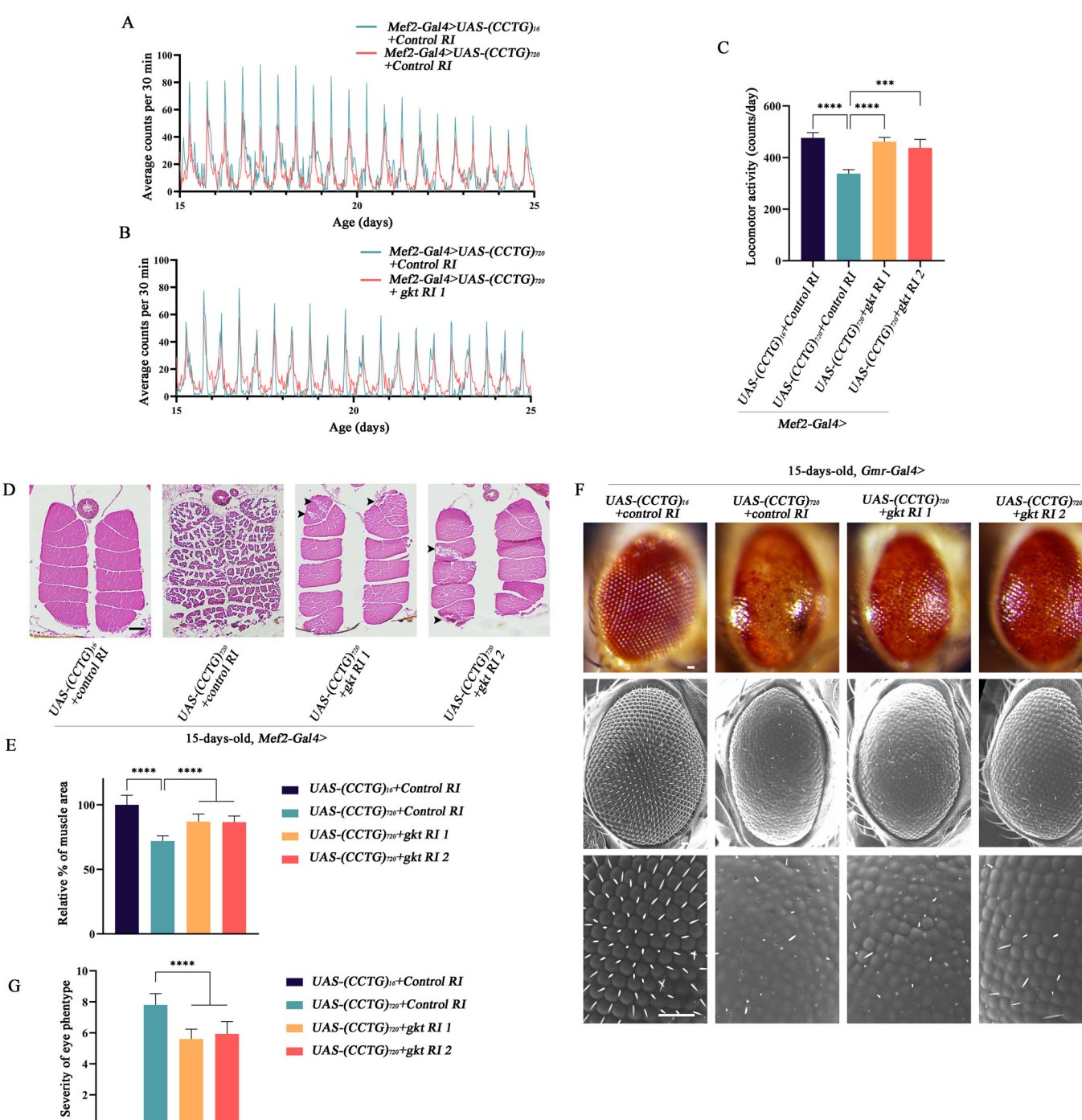

**Figure 2. TDP1/gkt knockdown alleviated neurodegeneration and muscle atrophy in DM2 flies.**

(A, B) The average counts of per 30 min in flies across different genotypes from 15-days-old to the 25-days-old. Fifteen animals per sample. (C) Quantification of (A, B) and *Mef2-Gal4 > UAS-(CCTG)$_{720}$+gkt RI 2* from five biological replicates. Data are mean ± SD. Two-tailed, unpaired *t* test. ****$P$ < 0.0001 (CCTG)$_{16}$ versus (CCTG)$_{720}$, ****$P$ < 0.0001 (CCTG)$_{720}$ versus (CCTG)$_{720}$ + *gkt RI 1*, ***$P$ = 0.0003 (CCTG)$_{720}$ versus (CCTG)$_{720}$ + *gkt RI 2*. (D, E) Representative images of paraffin-embedded adult thoraces showing IFMs of flies at 15-days-old. Black arrows, holes, and fragmentations. Scale bars 100 μm. (E) Quantification at least ten independent thoraces of each genotype/experiment, three experiments. Data are mean ± SD. Two-tailed, unpaired *t* test. ****$P$ < 0.0001 (CCTG)$_{16}$ versus (CCTG)$_{720}$, ****$P$ < 0.0001 (CCTG)$_{720}$ versus (CCTG)$_{720}$ + *gkt RI 1*, ****$P$ < 0.0001 (CCTG)$_{720}$ versus (CCTG)$_{720}$ + *gkt RI 2*. (F, G) Representative images of the compound eyes across different genotypes at 15-days-old. Top: LM images. Middle and bottom: SEM images. Scale bars 20 μm. (G) Quantification of at least 60 independent eyes of each genotype/experiment, three experiments. Data are mean ± SD. Two-tailed, unpaired *t* test. ****$P$ < 0.0001 (CCTG)$_{720}$ versus (CCTG)$_{720}$ + *gkt RI 1*, ****$P$ < 0.0001 (CCTG)$_{720}$ versus (CCTG)$_{720}$ + *gkt RI 2*. Source data are available online for this figure.

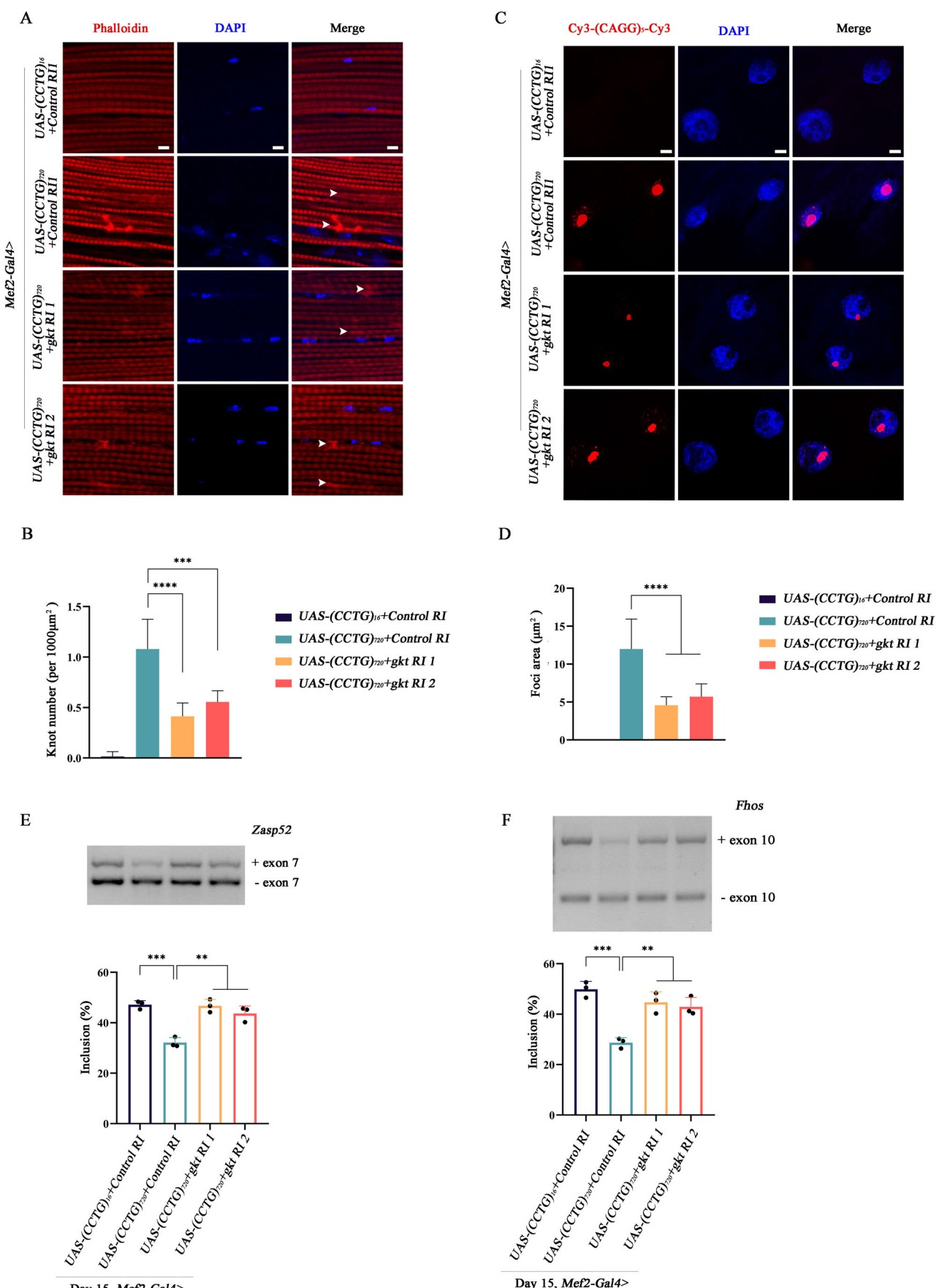

**Figure 3.  TDP1/gkt loss of function rescues molecular pathology of DM2.**

(A, B) Representative immunofluorescent (IF) images of thoracic muscle fibers across different genotypes at 15-days-old. Scale bars 5 μm. (B) Quantification at least ten animals of each genotype/experiment, three experiments. Data are mean ± SD. Two-tailed, unpaired $t$ test. ****$P$ < 0.0001 $(CCTG)_{720}$ versus $(CCTG)_{720} + gkt RI 1$, ***$P$ = 0.0001 $(CCTG)_{720}$ versus $(CCTG)_{720} + gkt RI 2$. (C, D) Representative Confocal images of CCUG-repeat foci with $(CAGG)_5$ probe across different genotypes at body-wall muscles of third instar larvae. Scale bars 5 μm. (D) Quantification of at least ten animals of each genotype/experiment, three experiments. Data are mean ± SD. Two-tailed, unpaired $t$ test. ****$P$ < 0.0001 $(CCTG)_{720}$ versus $(CCTG)_{720} + gkt RI 1$, ****$P$ < 0.0001 $(CCTG)_{720}$ versus $(CCTG)_{720} + gkt RI 2$. (E) Representative gel image of endogenous Zasp52 exon 7 with RT-PCR in flies at 15 days old. Tissue from ten animals per sample/experiment, five experiments. Lower panel: quantification of Zasp52 exon 7 inclusion from three biological replicates. Data are mean ± SD. Two-tailed, unpaired $t$ test. ***$P$ = 0.0005 $(CCTG)_{16}$ versus $(CCTG)_{720}$, **$P$ = 0.0015 $(CCTG)_{720}$ versus $(CCTG)_{720} + gkt RI 1$, **$P$ = 0.0052 $(CCTG)_{720}$ versus $(CCTG)_{720} + gkt RI 2$. (F) Representative gel image of endogenous Fhos exon 10 with RT-PCR in flies at 15-days-old. Tissue from ten animals per sample/experiment, five experiments. Lower panel: quantification of Fhos exon 10 inclusion from three biological replicates. Data are mean ± SD. Two-tailed, unpaired $t$ test. ***$P$ = 0.0006 $(CCTG)_{16}$ versus $(CCTG)_{720}$, **$P$ = 0.0037 $(CCTG)_{720}$ versus $(CCTG)_{720} + gkt RI 1$, **$P$ = 0.0043 $(CCTG)_{720}$ versus $(CCTG)_{720} + gkt RI 2$. Source data are available online for this figure.

of $(CCTG)_{720}$ in the retina, under the control of the *Gmr-Gal4* driver, disrupted eye morphology. By 15-days-old, notable abnormalities were observable, including diminished pigmentation, necrosis, ommatidial fusion, and loss of interommatidial bristles (Fig. 2F). Knockdown of *gkt* mitigated the loss of pigmentation and necrosis observed in DM2 flies. In addition, *gkt* knockdown partially restored the hexagonal shape of individual ommatidia, indicating preservation of eye integrity (Figs. 2F and EV1J). A quantitative evaluation of eye phenotypes demonstrated significant improvement in the condition of DM2 flies with *gkt* knockdown (Figs. 2G and EV1K). Thus, *TDP1/gkt* loss of function not only mitigates muscular toxicity but also suppresses neurodegeneration in the DM2 Drosophila model.

## TDP1/gkt loss of function rescues molecular pathology of DM2

To further explore the impact of *TDP1/gkt* on myotoxicity in DM2 model, we evaluated the myofiber damage, RNA toxicity, and splicing alterations. Flies expressing $(CCTG)_{720}$, compared to those with $(CCTG)_{16}$, displayed disorganized myofibrillar structures and widespread actin deposits within muscle fibers, termed "Actin blobs" (Fig. 3A) (Klein et al, 2014). Knockdown of *gkt* improved myofibrillar organization and reduced the incidence of actin blobs (Figs. 3B and EV2A,B).

A hallmark of DM2 pathology is the formation of nuclear foci containing expanded CCUG RNA sequences bound to RNA-binding proteins like MBNL1 (Ozimski et al, 2021). Using fluorescence in situ hybridization with a Cy3-labeled $(CAGG)_5$ probe, we found that *gkt* knockdown led to an approximately 50% reduction in the area of nuclear foci and a decrease in the sequestration of hMBNL1 within *gkt* RNAi/mutant flies (Figs. 3C,D and EV2C,D; Appendix Fig. S3A–C). In addition, the presence of these RNA foci results in splicing alterations of muscle-associated mRNAs. We then examined the splicing patterns of *Zasp52* and *Fhos*, known for their aberrant splicing in DM2 and importance for muscle function (Liao et al, 2016; Sellier et al, 2018; Shwartz et al, 2016; Yamashita et al, 2014). Our analyses indicated a partial restoration of alternative splicing, including the inclusion of exon 7 in *Zasp52*, exon 10/16 in *Fhos*, and exon 13 in *Serca*, following *gkt* knockdown (Figs. 3E,F and EV2E,F; Appendix Fig. S4A–C). These findings highlight the comprehensive impact of *TDP1/gkt* knockdown in addressing both structural and molecular abnormalities in DM2 pathology.

## Knockdown of TDP1/gkt could induce time-dependent contraction of large CCTG repeat via TOP1

Inspired by *TDP1*'s role in DNA damage repair and its improvements in both neurodegeneration and muscle toxicity, we suspected that *TDP1* might affect the stability of CCTG repeats. We used high GC content PCR and the Agilent high-sensitivity DNA chip to measure the size of the CCTG repeats (Appendix Fig. S5). A substantial reduction in the average length of the CCTG repeat in flies was observed with diminished *TDP1* expression (Figs. 4A,B and EV3A–C). *TDP1/gkt* knockdown affected the number of CCTG repeats rather than CCTG expression or transgene stability (Appendix Figs. S3D and S1E).

Given that Drosophila undergoes three developmental stages—egg, larva, and pupa—before reaching maturation, we found the developmental stage at which CCTG repeat contraction occurs. In the background of *gkt* knockdown, CCTG repeat contraction was not observed in embryos (using *mat-TubGal4*) or in pupal muscle, but it was noted in larval muscle (Figs. 4C,D and EV3D). This phenomenon may be explained by the distinct cell division states across the stages. During the embryonic and pupal stages, cells are in a state of continuous division and proliferation. In contrast, muscle cells remain in a non-dividing state during the larval stage and subsequently undergo apoptosis at the early pupal stage (Bothe and Baylies, 2016; Laurichesse and Soler, 2020). Apart from knockdown, *gkt* overexpression did not affect CCTG repeat size or locomotor in DM2 flies (Appendix Fig. S6).

We then examined how repeat size changes in adult flies during the aging process. By analyzing the muscle repeat lengths in *gkt* knockdown flies at 1, 7, 15 and 30-days-old, we observed a time-dependent contraction, with repeat lengths reaching half of $(CCTG)_{720}$ (Figs. 4E–G and EV3E). To determine whether the contraction also occurs in smaller CCTG repeat sizes, the $(CCTG)_{475}$ and $(CCTG)_{200}$ were used to test. Our analysis showed a minor contraction in $(CCTG)_{475}$ flies at 30-days-old, but not in those with $(CCTG)_{200}$. (Fig. EV3F–I). These results suggested that larger sizes are more prone to contraction. We next assessed the effects of *gkt* knockdown in endogenous short tandem repeat (STR) loci. With TRFinder software, we selected two loci with high GC content and the largest repeat number in the Drosophila genome, respectively (Benson, 1999). Knockdown of *gkt* did not affect the stability of these loci within the $(CCTG)_{720}$ background (Fig. EV3J,K).

Considering *TDP1*'s function as a repair enzyme that removes TOP-cc, we sought to understand TOP1's involvement in repeat

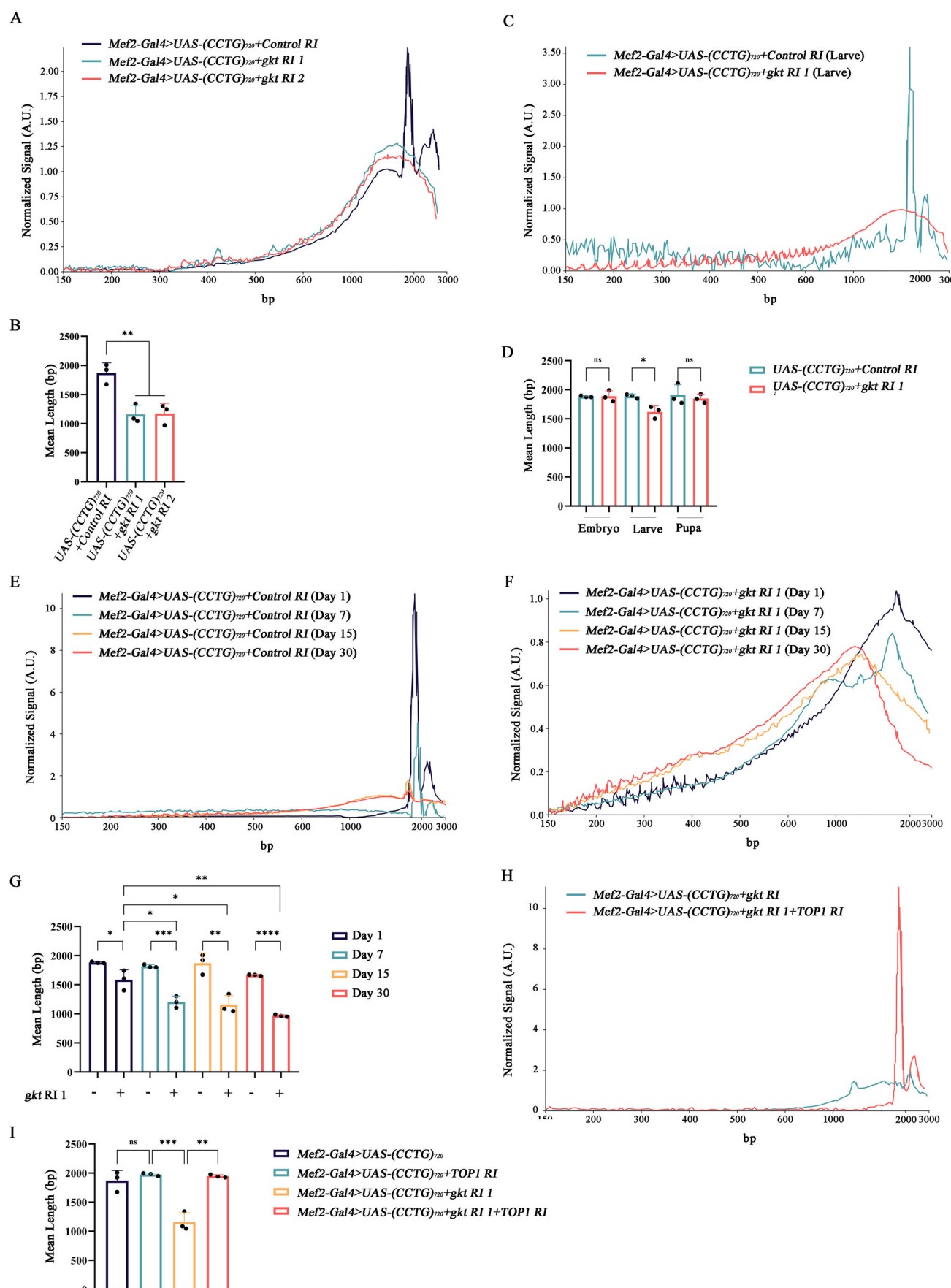

**Figure 4. Knockdown of TDP1/gkt could induce time-dependent contraction of CCTG repeat via TOP1.**

(A, B) Bioanalyzer quantitation of CCTG repeat lengths across different genotypes at 15-days-old. Tissue from three animals per sample. (B) Quantification of mean length of CCTG repeats from three biological replicates. Data are mean ± SD. Two-tailed, unpaired $t$ test. **$P = 0.0046$ $(CCTG)_{720}$ versus $(CCTG)_{720} + gkt\ RI\ 1$, **$P = 0.0082$ $(CCTG)_{720}$ versus $(CCTG)_{720} + gkt\ RI\ 2$. (C, D) Bioanalyzer quantitation of CCTG repeat lengths across $Mef2\text{-}Gal4 > UAS\text{-}(CCTG)_{720} + Control\ RI$ and $Mef2\text{-}Gal4 > UAS\text{-}(CCTG)_{720} + gkt\ RI\ 1$ at third instar larvae. Tissue from five animals per sample. (D) Quantification of mean length of CCTG repeats from three biological replicates. Data are mean ± SD. Two-tailed, unpaired $t$ test. *$P = 0.0132$. (E–G) Bioanalyzer quantitation of CCTG repeat lengths across $Mef2\text{-}Gal4 > UAS\text{-}(CCTG)_{720} + Control\ RI$ and $Mef2\text{-}Gal4 > UAS\text{-}(CCTG)_{720} + gkt\ RI\ 1$ at 1, 7, 15, 30-days-old. Tissue from three animals per sample. (G) Quantification of mean length of CCTG repeats from three biological replicates. Data are mean ± SD. Two-tailed, unpaired $t$ test. *$P = 0.0392$ 1 d $(CCTG)_{720}$ versus 1 d $(CCTG)_{720} + gkt\ RI\ 1$, ***$P = 0.0005$ 7 d $(CCTG)_{720}$ versus 7 d $(CCTG)_{720} + gkt\ RI\ 1$, **$P = 0.0046$ 15 d $(CCTG)_{720}$ versus 15 d $(CCTG)_{720} + gkt\ RI\ 1$, ****$P < 0.0001$ 30 d $(CCTG)_{720}$ versus 30 d $(CCTG)_{720} + gkt\ RI\ 1$, *$P = 0.0292$ 1 d $(CCTG)_{720} + gkt\ RI\ 1$ versus 7 d $(CCTG)_{720} + gkt\ RI\ 1$, *$P = 0.0340$ 1 d $(CCTG)_{720} + gkt\ RI\ 1$ versus 15 d $(CCTG)_{720} + gkt\ RI\ 1$, **$P = 0.0034$ 1 d $(CCTG)_{720} + gkt\ RI\ 1$ versus 30 days $(CCTG)_{720} + gkt\ RI\ 1$. (H, I) Bioanalyzer quantitation of CCTG repeat lengths across different genotypes at 15-days-old. Tissue from three animals per sample. (I) Quantification of mean length of CCTG repeats from three biological replicates. Data are mean ± SD. Two-tailed, unpaired $t$ test. ***$P = 0.0009$ $(CCTG)_{720} + TOP1\ RI$ versus $(CCTG)_{720} + gkt\ RI\ 1$, **$P = 0.0011$ $(CCTG)_{720} + gkt\ RI\ 1$ versus $(CCTG)_{720} + gkt\ RI\ 1 + TOP1\ RI$. Source data are available online for this figure.

contraction. The simultaneous knockdown of *TOP1* and *TDP1* mitigated the *TDP1*-induced repeat contraction (Figs. 4H,I and EV3L). This indicates that the CCTG repeat contraction observed in DM2 flies with reduced *TDP1/gkt* is mediated through TOP1 activity.

## Small molecules predicted to inhibit TDP1 activity and promote repeat contraction

To search for compounds that effectively inhibit TDP1 activity, we evaluated the binding affinity of 2,306,534 commercialized small molecules to the binding pocket of TDP1 (PDB: 6DIE) via a customized virtual screen (Figs. 5A,B and EV4A,B) (Eberhardt et al, 2021; Lountos et al, 2019). The top 20 compounds from virtual screening were subjected to functional screening to test their capacity to inhibit TDP1 enzyme activity in vitro (Table EV2). A specially designed DNA sensor, capable of detecting the removal of 3'-adducts by TDP1, was utilized for the functional screening (Jensen et al, 2013). The screening highlighted Lawsone and SPI-112 as the most potent inhibitors of TDP1 (Fig. 5C,D).

Camptothecin (CPT) can stabilize Top-cc, leading to DNA breaks (Beidler and Cheng, 1995). Given that TDP1 deficiency increases γ-H2AX foci following CPT treatment, we explored the capacity of Lawsone and SPI-112 to exacerbate the baseline accumulation of γ-H2AvD foci in Drosophila larvae. Lawsone and SPI-112 increased the staining intensity of γ-H2AvD foci in larval body-wall muscles, showing their potential to inhibit TDP1 activity in vivo (Figs. 5E and EV4C). To assess the cytotoxicity of these compounds, Hek293 cell was treated with Lawsone and SPI-112 for 72 h, establishing IC$_{50}$ values at 51.36 μM for Lawsone and 59.83 μM for SPI-112 (Fig. EV4D).

Lawsone and SPI-112 effectively inhibited TDP1 activity. We then investigated these compounds' effects on CCTG repeat instability. CPT was included in the study for its mechanism similar to TDP1 inhibitors, while Amikacin was incorporated due to its established role as a TDP1 inhibitor (Hubert et al, 2011; Liao et al, 2006). The compound eye phenotype was used to determine the optimal concentration (Fig. EV4E–H). Administering 15 μM CPT, 200 μM Amikacin, 0.5 μM Lawsone, and 1 μM SPI-112 to $Mef2\text{-}Gal4 > UAS\text{-}(CCTG)_{720}$ flies, we found all TDP1 inhibitors facilitated repeat contraction, with our compounds showing comparable efficacy at the lowest concentrations (Figs. 5F,H and EV4I). Consistent with genetic knockdown observations, *TOP1* knockdown neutralized the contraction triggered by Lawsone and SPI-112 (Figs. 5G,H and EV4E). Together, these findings

underscore that our compound not only significantly inhibits TDP1 activity both in vitro and in vivo but also promotes CCTG repeat contraction.

## *TDP1/gkt* inhibitors suppress the muscular toxicity and neurodegeneration

To test muscular toxicity and neurodegeneration, DM2 flies treated with DMSO, CPT, Amikacin, Lawsone, and SPI-112 were evaluated. All compounds except Amikacin reduced the locomotor deficits in DM2 flies compared to the DMSO control (Fig. 6A–C). The weaker TDP1 inhibition by Amikacin may explain the observation. We then analyzed the effect of the compounds on muscle paraffin sections. All compounds except Amikacin increased muscle area and improved structural integrity at least in one or two sections of the muscle (Fig. 6D,E). Regarding neurodegeneration within the compound eye, all compounds except CPT improved eye phenotypes, including pigment loss and degeneration severity, while CPT exacerbates necrosis due to its cytotoxicity (Fig. 6F,G).

Next, we investigated the impact of TDP1 inhibitors on the molecular pathology associated with DM2. Excluding CPT due to its toxicity, all compounds reduced actin blobs in DM2 flies (Fig. 6H,I). In terms of RNA toxicity, all compounds decreased the area of RNA foci (Fig. 6J,K). In addition, all compounds expect Amikacin to partially correct the splicing alterations caused by expanded repeat RNA (Fig. 6L,M). Among these, Lawsone and SPI-112 emerged as ideal TDP1 inhibitors, successfully mitigating the complex phenotypes of DM2, ranging from muscular toxicity and neurodegeneration to molecular alterations.

In addition to the DM2 Drosophila model, the effects of TDP1/gkt inhibition were validated in other repeat expansion models. To investigate the impact of TDP1 inhibition on CTG repeats, a DM1 Drosophila model expressing $(CTG)_{270}$ repeats was tested (Fig. EV5A,B; Appendix Fig. S7). Capillary electrophoresis results demonstrated that genetic knockdown of *gkt* facilitated CTG repeat contraction. Further validation was performed using HEK293 cell lines stably expressing $(CTG)_{400}$ or $(CCTG)_{400}$ repeats, confirming that TDP1 inhibition promotes repeat contraction across species (Fig. EV5C,D). After siRNA-mediated TDP1 knockdown combined with CPT treatment, both CTG and CCTG repeats exhibited reductions in length. Beyond transgenic models, DM1 patient-derived lymphocytes were also examined. Treatment with TDP1 inhibitors, Lawsone and SPI-112, effectively induced repeat contraction (Fig. EV5E,F).

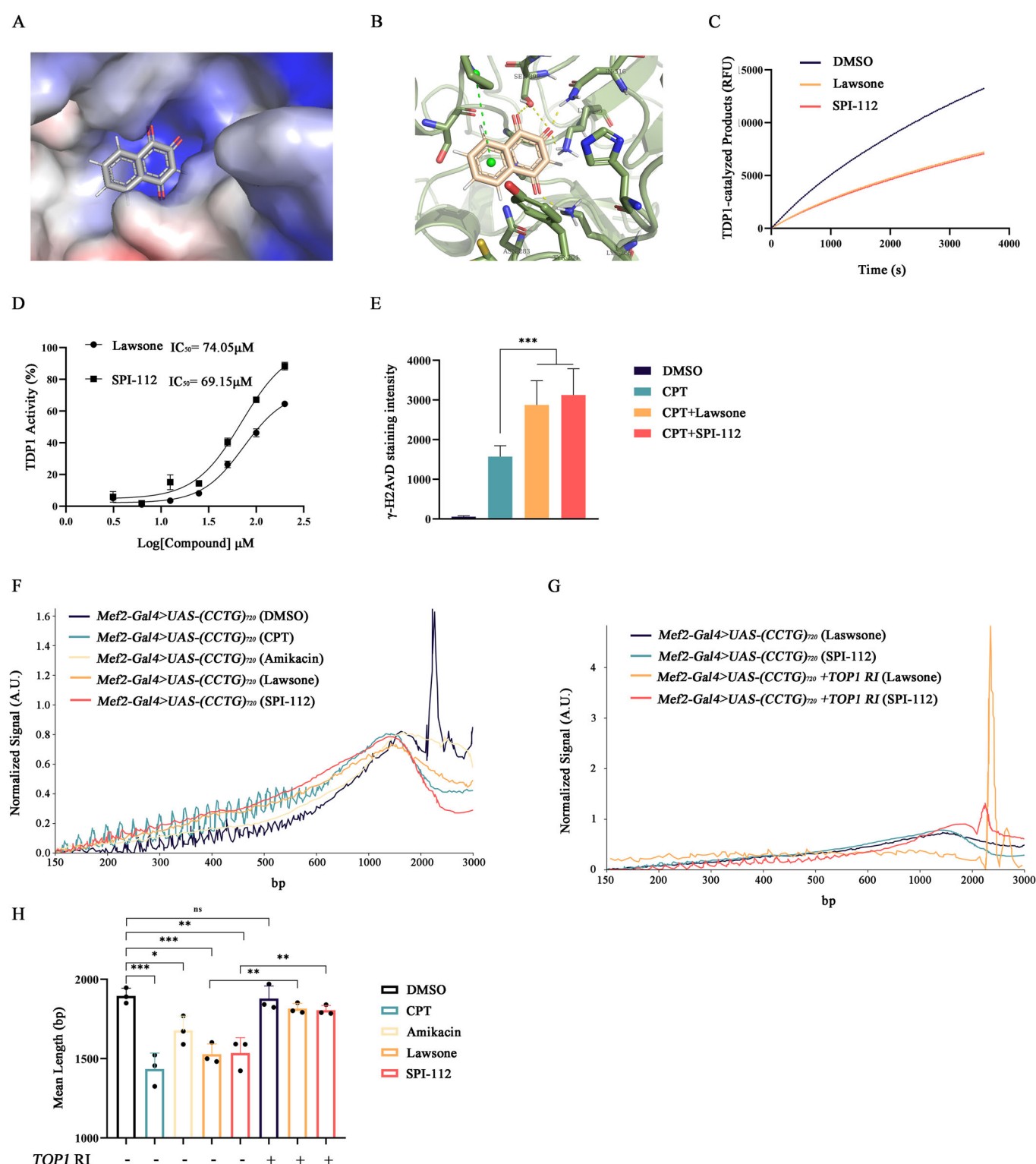

## Discussion

Through extensive chemical screening in the DM2 Drosophila model, we have uncovered a critical role for *TDP1* as a modifier of muscular toxicity in DM2. Genetic and pharmacological inhibition of *TDP1/gkt* have a substantial protective effect against a variety of

muscular toxicity ranging from locomotion to molecular pathology. Intriguingly, our data highlight that the contraction of CCTG repeats upon *TDP1/gkt* inhibition is the key mechanism driving the significant improvements in the disease phenotypes.

The disruption of the compound eye and reduced muscle area have been described in previous studies. Given that muscle

**Figure 5. Small molecules predicted to inhibit TDP1 activity and promote repeat contraction.**

(A) Surface electrostatic potentials of predicted crystal structure of human TDP1/Lawsone complex. Red depicts a positive partial charge on the surface, blue depicts negative partial charge and gray shows neutral/lipophilic areas. (B) Docking model of Lawsone bind to TDP1 and Lawsone. Dotted yellow line: hydrogen bond, Dotted green line: Pi bond. Blue: N atom, Red: O atom, White: H atom, Green: protein skeleton, Gold: compound skeleton. (C) Representative graphs showing measurement of fluorescence development over time. The slope of the initial linear phase is a representation of the TDP1 activity assay. (D) Dose–response analysis of Lawsone and SPI-112. The activity of TDP1 was treated with purified TDP1 enzyme 3.125, 6.25, 12.5, 25, 50, 100, 200 μM. Data are mean ± SD from three biological replicates. IC$_{50}$ values were calculated using log(inhibitor) compared with normalized response (variable slope), using the program Prism, and are given in graphs. (E) Quantification of γ-H2AvD staining intensity in nuclear of body-wall muscles at third instar larvae of *Wild-type*. Quantification at least five animals of each sample/experiment, five experiments. Data are mean ± SD. Two-tailed, unpaired *t* test. ***P = 0.0007 CPT versus CPT + Lawsone, ***P = 0.0003 CPT versus CPT + SPI-112. (F) Bioanalyzer quantitation of CCTG repeat lengths at 15-days-old *Mef2-Gal4 > UAS-(CCTG)$_{720}$* with treatment of different compounds. Tissue from three animals per sample. (G) Bioanalyzer quantitation of CCTG repeat lengths across 15-days-old different genotypes with treatment of different compounds. Tissue from three animals per sample. (H) Quantification of (F, G) from three biological replicates. Data are mean ± SD. One-way ANOVA ***P = 0.0002 DMSO versus CPT, *P = 0.00286 DMSO versus Amikacin, ***P = 0.0010 DMSO versus Lawsone, **P = 0.0011 DMSO versus SPI-112, **P = 0.0024 Lawsone versus Lawsone (*TOP1 RI*), **P = 0.0099 SPI-112 versus SPI-112 (*TOP1 RI*). Source data are available online for this figure.

involvement is the predominant clinical feature of DM2, this study delved into the aspects of muscle toxicity. We observed reductions in locomotion, progressive muscle degeneration, abnormalities in muscle fibers, and signs of molecular pathology. DM2 flies showed a decline in locomotor abilities compared to controls, with age-dependent morphological defects emerging in the IFM. All six muscles of the IFM remain morphologically intact at 1-day-old flies. The CCUG toxic RNA progressively induces muscle apoptosis with aging, eventually leading to the formation of vacuoles, disorganization of muscle fibers, and fragmentation of muscle tissue. Additionally, actin blobs frequently appeared within the muscle fibers of DM2 flies, reminiscent of those observed in troponin T mutant flies (Schnorrer et al, 2010; Singh et al, 2014). Troponin T, a subunit of the troponin complex, regulates muscle contraction and serves as a biomarker of DM (Bose et al, 2019; Salvatori et al, 2009). Furthermore, RNA toxicity was evident in muscles, marked by the presence of foci in nuclei. Abnormalities were observed in the splicing patterns of *Fhos* and *Zasp52*, which are important for sarcomeric thin-filament array assembly and as a core Z-disc protein in IFM, respectively (Liao et al, 2020; Shwartz et al, 2016). The phenotypes we characterized, reflecting muscular toxicity, were significantly improved by targeting TDP1 via genetic knockdown and pharmacological inhibition.

Since the discovery of CGG and CAG repeat expansions as causes of genetic disorders in 1991, more than 50 repeat expansion disorders have been reported (Depienne and Mandel, 2021; La Spada et al, 1991; Oberle et al, 1991). Healthy individuals also possess these repeats, with disease onset occurring when the size of the repeats exceeds a certain threshold. Furthermore, a strong correlation has been established between the size of these expansions and both the age at onset and the severity of repeat expansion disorders (Gousse et al, 2018; Pearson et al, 2005). The larger the expansions are associated with the earlier the age at onset and the more severe the phenotype. Repeat expansions trigger a series of concurrent molecular processes, such as repeat-induced transcriptional silencing, toxic RNA-mediated RNA-binding protein sequestration, AUG-initiated repeat-harboring native proteins and repeat-associated non-AUG translation of toxic repetitive peptides, all of which are derived from the expanded repeat sequences (Depienne and Mandel, 2021). Furthermore, the expanded nucleotide repeats are unstable in germline and somatic cells, leading to intergenerational instability and somatic mosaicism that contribute to disease progression (Joosten et al, 2020; Nolin

et al, 2019). Developing therapeutic interventions that directly reduce the number of expanded repeats in affected tissues could profoundly alleviate the repeat-mediated toxicity and potentially delay the onset of repeat expansion diseases by decades or even offer lifetime safeguarding.

In this study, we found that targeting TDP1 could induce CCTG repeat contraction. TDP1 helps repair 3'-termini at DNA single-strand breaks (SSBs), which caused by TOP1 during replication and transcription. Studies on primary cultured neurons from *TDP1* knockout mice and non-dividing lymphoblastoid cells from patients with *TDP1* mutations have shown an accumulation of SSBs after exposure to CPT, which induces breakage via stalled TOP-cc. These findings indicate that *TDP1* deficiency leads to an accumulation of SSBs (El-Khamisy et al, 2005; Katyal et al, 2007). The association between *TDP1* and repeat instability has been reported in two studies. *TDP1* knockdown resulted in a decrease in CAG repeat numbers from 95 to fewer than 38, and the contraction occurred in approximately 0.08% of cells (Hubert et al, 2011). A subsequent investigation using a cell model harboring 800 CTG•CAG repeats found that *TDP1* knockdown led to a slight decrease in repeat length. No significant difference was observed compared to the control group, which also exhibited a reduction in repeat length without undergoing TDP1 knockdown (Nakatani et al, 2015). These two studies on dividing human fibrosarcoma cells did not show substantial changes in repeat numbers. Our study, performed in the multicellular organism, revealed that *TDP1*-induced repeat contraction occurs within muscle tissue, the predominant site of involvement in DM2. The involvement of DNA breakage and R-loop formation in repeat contraction is a possibility that warrants additional exploration. The secondary structures and high GC content of the expanded CCTG repeats tend to facilitate RNA-DNA hybrid (R-loop) formation during transcription (Wu et al, 2023). The non-template DNA strands within these R-loops are highly susceptible to damage (Su and Freudenreich, 2017). This suggests that a diminished TDP1 activity might contribute to an accumulation of SSBs and/or the formation of R-loops within the repeat sequence, which could, in turn, prompt repeat contraction. Exploring the complex interactions, encompassing the impact of R-loops, the range of DNA damage, and the detailed repair processes, demand a comprehensive suite of future studies.

Targeting TDP1 in humans could represent a safe therapeutic strategy. Homozygous mutation of *TDP1* causes spinocerebellar ataxia with axonal neuropathy-1, resulting in the progressive degeneration of post-mitotic neurons (Salih et al, 2007). The

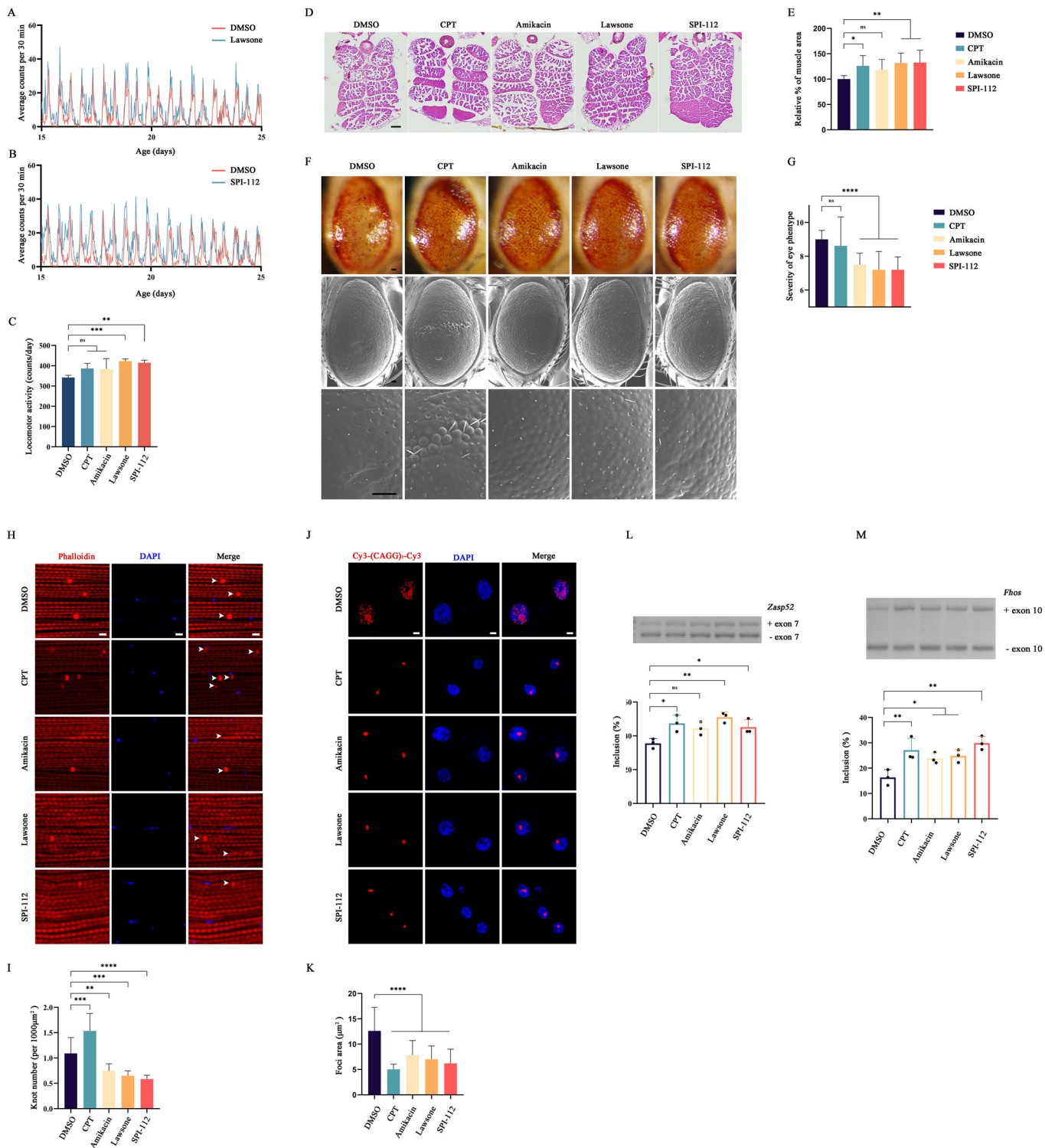

heterozygous mutation carriers, in both humans and model organisms, exhibit no apparent phenotype (El-Khamisy et al, 2005; Guo et al, 2014; Hirano et al, 2007). In addition, the application of TDP1 inhibitors has been explored in tumor resistant to TOP1 inhibitors, such as metastatic ovarian, cervical, and small-cell lung cancers (Hu et al, 2021). The development of TDP1

inhibitors is pursued as a potential adjunct therapy to address the resistance. The effect of TDP1 inhibition on repeat expansion requires substantial time and effort for validation in other models, such as iPSC-derived muscle tissue from patients, higher animal models, and organoid models, in the future. In summary, our study highlights TDP1 as a potential target for reducing the size of CCTG

**Figure 6. TDP1 inhibitors suppress the muscular toxicity and neurodegeneration.**

(A–C) The average counts per 30 min in *Mef2-Gal4 > UAS-(CCTG)$_{720}$* with treatment of different compounds from 15-days-old to the 25-days-old. Fifteen animals per sample. (C) Quantification from five biological replicates. Data are mean ± SD. One-way ANOVA. ***$P = 0.005$ DMSO versus Lawsone, **$P = 0.0014$ DMSO versus SPI-112. (D, E) Representative images of paraffin-embedded adult thoraces showing IFMs of *Mef2-Gal4 > UAS-(CCTG)$_{720}$* with treatment of different compounds at 15-days-old. Scale bars 100 μm. (E) Quantification of at least ten independent thoraces of each sample/experiment, three experiments. Data are mean ± SD. One-way ANOVA. *$P = 0.00148$ DMSO versus CPT, **$P = 0.0023$ DMSO versus Lawsone, **$P = 0.0018$ DMSO versus SPI-112. (F, G) Representative images of the compound eyes of *Gmr-Gal4 > UAS-(CCTG)$_{720}$* with the treatment of different compounds at 15-days-old. Top, LM images. Middle and Bottom, SEM images. Scale bars 20 μm. (G) Quantification of at least sixty independent eyes of each sample/experiment, three experiments. Data are mean ± SD. One-way ANOVA. ****$P < 0.0001$ DMSO versus Amikacin, ****$P < 0.0001$ DMSO versus Lawsone, ****$P < 0.0001$ DMSO versus SPI-112. (H, I) Representative IF images of thoracic muscle fibers of *Mef2-Gal4 > UAS-(CCTG)$_{720}$* with treatment of different compounds at 15-days-old. White arrows, action blobs. Scale bars 5 μm. (I) Quantification of at least ten animals of each sample/experiment, three experiments. Data are mean ± SD. One-way ANOVA. ***$P = 0.0006$ DMSO versus CPT, **$P = 0.0078$ DMSO versus Amikacin, ***$P = 0.0006$ DMSO versus Lawsone, ****$P < 0.0001$ DMSO versus SPI-112. (J, K) Representative Confocal images of CCUG-repeat foci with (CAGG)$_5$ probe of *Mef2-Gal4 > UAS-(CCTG)$_{720}$* with treatment of different compounds at body-wall muscles of third instar larvae. Scale bars 5 μm. (K) Quantification of at least ten animals of each sample/experiment, three experiments. Data are mean ± SD. One-way ANOVA. ****$P < 0.0001$. (L) Representative gel image of endogenous *Zasp52* exon 7 with RT-PCR in *Mef2-Gal4 > UAS-(CCTG)$_{720}$* with treatment of different compounds at 15-days-old. Tissue from ten animals per sample/experiment, 5 experiments. Lower panel, quantification of *Zasp52* exon 7 inclusion. Data are mean ± SD. One-way ANOVA. *$P = 0.0217$ DMSO versus CPT, **$P = 0.0022$ DMSO versus Lawsone, *$P = 0.0404$ DMSO versus SPI-112. (M) Representative gel image of endogenous *Fhos* exon 10 with RT-PCR in *Mef2-Gal4 > UAS-(CCTG)$_{720}$* with treatment of different compounds at 15-days-old. Tissue from ten animals per sample/experiment, five experiments. Lower panel, quantification of *Fhos* exon 10 inclusion. Data are mean ± SD. One-way ANOVA. **$P = 0.0061$ DMSO versus CPT, *$P = 0.0470$ DMSO versus Amikacin, *$P = 0.0249$ DMSO versus Lawsone, **$P = 0.0012$ DMSO versus SPI-112. Source data are available online for this figure.

repeat sequences, offering a new strategy to treat DM2 and possibly applicable to a broad spectrum of repeat expansion diseases.

## Limitations of the study

This study highlights the potential of TDP1 inhibition in promoting repeat contraction. Further study is needed to fully elucidate the molecular mechanisms underlying repeat contraction. Validation in patient-derived muscle fibers and mouse models will be essential for clinical translation. Moreover, investigating whether TDP1 inhibition could have therapeutic effects in DM1 and even other repeat expansion disorders remains an important direction for future research.

## Methods

### Reagents and tools table

| Reagent/resource | Reference or source | Identifier or catalog number |
|---|---|---|
| **Experimental models** | | |
| *24B-Gal4* | Prof. ZhuoHua Zhang, Central South University | N/A |
| *Gmr-Gal4* | Prof. ZhuoHua Zhang, Central South University | N/A |
| *mat-Tub-Gal4* | Prof. Kai Yuan, Central South University | N/A |
| *UAS-(CCTG)$_n$* | Prof. Nancy M. Bonini, Boston University | N/A |
| *Mef2-Gal4* | Bloomington Drosophila Stock Center | 27390 |
| *W1118* | Bloomington Drosophila Stock Center | 6326 |
| *gkt mut* | Bloomington Drosophila Stock Center | 11321 |
| *UAS-(CTG)$_{19}$* | Bloomington Drosophila Stock Center | 79576 |
| *UAS-(CTG)$_{270}$* | Bloomington Drosophila Stock Center | 79581 |

| Reagent/resource | Reference or source | Identifier or catalog number |
|---|---|---|
| *gkt RNAi 1* | Vienna Drosophila RNAi Center | 109757 |
| *gkt RNAi 2* | Vienna Drosophila RNAi Center | 46268 |
| *Control RNAi* | Vienna Drosophila RNAi Center | 60200 |
| *TOP1 RNAi* | Tsing Hua Fly Center | TH02218.N |
| *UAS-gkt-HA* | This study | N/A |
| HEK-293 cells (*H. sapiens*) | ATCC | CRL-1573 |
| **Recombinant DNA** | | |
| pcDNA3.1(+) | ThermoFisher | V79020 |
| **Antibodies** | | |
| Rhodamine-phalloidin | ThermoFisher | R415 |
| DAPI | Sigma | D9542 |
| Anti-Histone H2AvD | Rockland | 600-401-914 |
| Anti-rabbit Cy3 | Jackson Immuno Research | 711-165-152 |
| **Oligonucleotides and other sequence-based reagents** | | |
| PCR primers | This study | Dataset EV3 |
| Probes | This study | Dataset EV3 |
| **Chemicals, enzymes, and other reagents** | | |
| High Sensitivity DNA Kit | Agilent | 5067-4626 |
| Lawsone | MCE | HY-N2493 |
| SPI-112 | MCE | HY-101964 |
| Camptothecin (CPT) | TargetMol | T1123 |
| Amikacin | TargetMol | T1013 |
| Dimethyl sulfoxide (DMSO) | TargetMol | T0341 |
| Hi-Pure Animal Genomic DNA Kit | Tsingke | MD101 |
| FastPure Gel DNA Extraction Kit | Vazyme | DC301 |
| AmpliTaq Gold DNA Polymerase | ThermoFisher | 4398823 |

| Reagent/resource | Reference or source | Identifier or catalog number |
|---|---|---|
| CCK-8 Kit | Beyotime | C0037 |
| Lipo2000 | ThermoFisher | L3000015 |
| **Software** | | |
| GraphPad Prism | https://www.graphpad.com | |
| Python | https://www.python.org/ | |
| AutoDock Vina | https://autodock-vina.readthedocs.io/en/stable/index.html | |
| PyMOL | https://www.pymol.org/ | |
| **Other** | | |
| Drosophila Activity Monitor | TriKinetics | DAM2 Drosophila Activity Monitor |
| Bioanalyzer 2100 | Agilent | G2939BA |

## Fly work

Fly culture and crosses were performed on standard food according to standard procedures and raised at 25 °C.

## Compounds treatment and chemical screening

The compounds utilized in this study are as follows: 2160 compounds from the JHCCL, Lawsone (MCE), SPI-112 (MCE), Camptothecin (CPT, TargetMol), Amikacin (TargetMol), and DMSO (TargetMol).

The compounds from the JHCCL were used for chemical screen at a dose of 40 μM. Chemical screening was performed in 5 mL centrifuge tubes due to the limited availability of compounds. In the primary screen, 140 compounds were tested using the *24B-Gal4* driver, which induces broad expression of $(CCTG)_{720}$ from the mesoderm to muscle cells. The expression caused complete lethality in flies, allowing for the identification of compounds that could potentially rescue toxicity by influencing muscle development.

For the secondary screen, the *Mef2-Gal4* driver, primarily expressed in mature muscle, was employed. This driver resulted in semi-lethality, providing an easily quantifiable phenotype for further analysis. Compounds with survival counts showing a 1.5-fold increase compared to DMSO-treated flies ( > 30–40, approximately the top 20%) were selected.

In total, 31 compounds were identified in the secondary screen that exhibited varying degrees of rescue effects against muscle toxicity. Among these, 16 compounds with target information available in the bioassay of PubChem Compound database were selected for subsequent analysis.

The CPT, Amikacin, Lawsone, and SPI-112 were dissolved in DMSO and added to standard food at 15, 200, 0.5, 1 μM, respectively.

## Behavioral assays

For locomotion assays, 15-days-old male flies were collected and continuously monitored using the Drosophila Activity Monitor system (TriKinetics). The Drosophila Activity Monitor (DAM) utilizes centrally positioned 5 mm activity tubes, which are equipped with fly food at one end and an air-permeable closure at the other. This setup houses individual flies for experiments lasting 15–25 days. The activity tubes are fitted with infrared monitoring system that records whenever a fly crosses the midpoint of the tube. As described previously, each interruption when a fly crossed the infrared beam was recorded as an activity event. For each individual genotype, at least 15 flies were simultaneously monitored to obtain the locomotion activity data.

For climbing assays, 15-days-old male flies were tested. For each genotype, more than 3 groups of 10 males were transferred into 1.25-cm-diameter and 28-cm-height plastic tubes and incubated for 1 h at room temperature to relieve anesthesia and acclimate to the environment. For each group, Tubes were tapped until all flies were at the bottom and the time that it took for the fifth fruit fly to climb 15 cm was recorded.

For Flight assay, 15-days-old male flies were tested. Briefly, paraffin oil is applied to the inner wall of a 100 ml graduated cylinder. Using a funnel, 20 flies are poured into the cylinder from top to bottom. Once the flies stick to the wall, the number of flies falling within each interval ( < 5 cm, 5–10 cm, 10–15 cm, …, 30–35 cm, >35 cm) is recorded, and the average height and distribution are calculated.

## Paraffin section

The histological sections of the muscle were prepared from wax-embedded material. Male flies were fixed in Carnoy's solution at 4 °C overnight. The sample was dehydrated using increasing concentrations of ethanol for 10 min (40%, 60%, 80%, 100%, and 100%). Next, the flies were incubated in Methyl benzoate (MB, Sangon Biotech), MB + paraffin solution (V:V = 1:1), paraffin I and paraffin II for 30 min in each at 65 °C. The collar was rapidly relocated into a plastic box and filled with melted paraffin (65 °C), and placed at room temperature (RT) overnight. The paraffin blocks were sectioned at 6 μm intervals using Leica paraffin microtome. The sections were stained with hematoxylin (Sangon Biotech) and eosin (Sangon Biotech) and analyzed using light microscopy (DM5000B, Leica Microsystems).

## Light microscopy (LM), scanning electron microscopy (SEM) and scoring compound eye degeneration

Whole flies were analyzed with an OLYMPUS DP72 microscope (OLYMPUS) for LM images and a JEOL JSM-6360-LA scanning electron microscope (JEOL) for SEM images. The scoring criteria to apply quantitative analyses for Drosophila ommatidial phenotypes by light microscopy had been described previously (Lin et al, 2020). Higher scores denote more severe phenotypes. Three independent experimenters, blinded to the experimental conditions, conducted the scoring analyses.

## Immunoblotting assay

The thorax of male flies was fixed in 4% paraformaldehyde (PFA) for 1 h at RT. Then, the muscular tissue was obtained from the dissected thorax in 0.3% PBST and blocked for 1 h in 0.3% PBST with 10% normal goat serum for 1 h at RT. The muscle fibers were labeled with rhodamine-phalloidin (1:1000, ThermoFisher Scientific), and nuclei were counterstained with DAPI (1 μg/mL; Sigma Aldrich). The samples were observed with Zeiss LSM880 laser scanning confocal microscopy.

## In situ hybridization

The Cy3-(CAGG)$_5$-Cy3 oligo (synthesized by Tsingke) used to detected CCUG repeats RNA foci as described. Briefly, the body-wall muscles of third instar larvae were fixed in 4% PFA for 30 min at 4 °C, hybridized with probe (2 ng/μl) for 16 h at 42 °C in SSC buffer (Sangon Biotech). The nuclei were counterstained with DAPI (1 μg/mL) for 10 min at RT. The samples were observed with Zeiss LSM880 laser scanning confocal microscopy.

## RT-PCR

Total RNA from the thoracic muscle of ten male flies was isolated by TriReagent (ThermoFisher). cDNAs were generated using the HiScript II Q RT SuperMix for qPCR kit (Vazyme) for quantification of mRNAs. PCR was performed with 2×Taq Master Mix (Novoprotein), one denaturation step at 94 °C for 1 min 30 s, 26 cycles of amplification 94 °C for 20 s, 60 °C for 20 s, 72 °C for 1 min and a final step at 72 °C for 5 min. The PCR products were analyzed by electrophoresis on a 2% agarose gel, stained with GelStain (Transgen,) and quantified with a Gel image analysis system (Tannon 1600). The primers used for the PCR are listed in Table EV3.

## High GC content PCR and Bioanalyzer analysis

The DNA was extracted from the thoracic muscle of three male flies (for the embryo stage, ten complete embryos. For the third instar stage, body-wall muscles of five files. For pupae, thoracic muscle of five files) using the Hi-Pure Animal Genomic DNA Kit (Tsingke). The CCTG repeat number was quantified using High GC content PCR with AmpliTaq Gold DNA polymerase (Applied Biosystems) and specific primers (CCTG)$_{720}$-F/R, Table EV3). The thermal cycling conditions were as follows: initial denaturation at 98 °C for 10 min, followed by 35 cycles at 98 °C for 30 s, 60 °C for 30 s, and 68 °C for 4 min, with a final extension at 68 °C for 10 min.

High GC content PCR products were purified by using FastPure Gel DNA Extraction Mini Kit (Vazyme), and the purified samples were further analyzed by High-sensitivity DNA kit (Agilent) with Bioanalyzer 2100 (Agilent) to quantify the length of CCTG fragments. The estimated fragment length was calculated by the migration time to size conversion based on the ladder migration. Raw signal data was normalized by the average intensity between 300 bp to 3 kb. The mean length is calculated based on the signal intensity between 300 bp to 3 kb.

## Virtual screening

The compounds databases were respectively downloaded from MCE (HY-L001V, HY-L032V, and HY-L0051V), Selleck (L3800, L3900), and Chemdiv (100k Diverse Compounds Pre-Plated Set). The compounds were screened to the crystal structure of TDP1 (6DIE, resolution 1.78 Å), which was obtained from the Protein Data Bank (PDB). The Pymol was used to prepare the crystal structure for docking, including adding hydrogen atoms, and removing the co-crystallized benzene-1,2,4-tricarboxylic acid as well as crystallographic water molecules. The center of the binding pocket was defined as the position of the His263 and His493 with 16 Å radius, respectively. The docking runs were performed in AutoDock Vina using the default scoring function. The order of compounds is based on their docking with His263 and His493, respectively.

## TDP1 in vitro activity assay

This method has been previously described in detail. Briefly, In vitro testing of the small molecules' capability of TDP1 inhibition was carried out by using the DNA sensor (Table EV3, synthesized by Sangon Biotech). In total, 40 ng of purified human TDP1 protein (purified as previously described) was incubated with the potential TDP1 inhibitors in a final concentration of 40 μM for each compound or DMSO equivalent to the DMSO content of the potential TDP1 inhibitors (0.8%) (Jensen et al, 2013). A final concentration of 0.5 μM DNA sensor was added on ice. The final volume of 25 μL reaction was carried out using the CFX Real-Time PCR Detection System (BioRad), and fluorescence was measured every 30 s for 1 h. The initial linear slope was used as a relative measure of TDP1 activity.

## γ-H2AvD assay

The body-wall muscles of third instar larvae were fixed in 4% PFA for 1 h at RT, Then, the muscular tissue was blocked for 1 h in 0.3% PBST with 10% normal goat serum for 1 h at RT. The Rabbit-anti-Histone H2AvD (1:1000, Rockland) was used as the primary antibody, secondary antibody was anti-rabbit Cy3 (1:200, Jackson Immuno Research). The nuclei were counterstained with DAPI (1 μg/mL) for 10 min at RT. The samples were observed with Zeiss LSM880 laser scanning confocal microscopy.

## CCK-8 assay

This method has been previously described in detail (Cai et al, 2019). Briefly, cell viability was determined by Cell Counting Kit-8 (Beyotime). Hek293 cells were seeded in 96-well plates for 24 h. The medium was then replaced with 100 μL of opti-MEM containing different compounds as described, followed by a 24 h incubation. 20 μL of CCK-8 was added in each well and kept for an additional 2 h. Absorbance at 450 nm was measured using the microplate Reader (BioTek Instruments).

## Cell culture

The construction of the Hek293 cell model expressing (CTG)$_{400}$ or (CCTG)$_{400}$ has been described previously (Ordway and Detloff, 1996). In brief, Hek293 cells were transfected with a plasmid pCDNA3.1 containing 400 CTG/CCTG repeats. Transfection was performed with lipo3000 (Thermofisher), and stably transfected clones were selected with G418. For siRNA treatment, the Hek293 cells were plated in a six-well plate, and 1 μM TDP1 siRNA (Tsingke, Table EV3) was added 6 h later. This was followed by the subsequent addition of 0.02 μM CPT. Cells were incubated at 37 °C with 5% $CO_2$ in DEME Media with 5% fetal bovine serum. Cells were passaged twice weekly with continuous exposure to 1 μM siRNA. The primers used for the PCR are DM1-F/R.

The DM1 patient-derived lymphoblastoid cell lines were established after EBV transformation. This study was approved by the Ethics Committee of Center for Medical Genetics, the School of Life Sciences, Central South University, and informed

## The paper explained

### Problem

Myotonic dystrophy type 2 (DM2), caused by CCTG repeat expansion, is a common adult-onset disorder characterized by myotonia and progressive muscle degeneration. Current therapeutic efforts have predominantly focused on DM1, leaving DM2 patients with only symptomatic management options. Identifying novel therapeutic targets through drug screening may help elucidate key molecular pathways involved in DM2 and offer potential strategies for treatment.

### Results

Through a high-throughput chemical screening of 2160 compounds, we identified Tyrosyl-DNA phosphodiesterase 1 (TDP1) as a novel modifier for DM2 therapeutic intervention. Both genetic and pharmacological inhibition of TDP1 resulted in significant improvements, including motor functions, progressive muscle degeneration, muscle fiber damage, and aberrant molecular pathology. Notably, TDP1 inhibition led to substantial CCTG repeat contractions, directly mitigating the toxic mechanisms driving muscle toxicity and neurodegeneration. These findings suggest that targeting TDP1 could be a promising strategy to counteract the effects of repeat expansions in DM2.

### Impact

Our findings highlight TDP1 as a promising therapeutic target for DM2. By alleviating muscle toxicity and neurodegeneration, TDP1 inhibition offers a novel approach to mitigating DM2 pathology. Given TDP1's role in DNA repair and genomic stability, these results may extend beyond DM2, providing insights into DM1 and even other repeat expansion disorders.

consent was obtained from all participants (approval number: 2022-1-4). Proper informed consent was obtained from all human subjects, and the experiments conformed to the principles set out in the WMA Declaration of Helsinki and the Department of Health and Human Services Belmont Report. For compound treatment, the CPT, Amikacin, Lawsone, and SPI-112 were dissolved in DMSO and added at 10, 520, 42, 135 μM, respectively. Cells were incubated at 37 °C with 5% $CO_2$ in RPMI-1640 Media with 10% fetal bovine serum. The primers used for the PCR are pcDNA-F/R.

## Statistical methods

Quantification of replicate experiments is presented as mean ± SD as described in figure legends. Statistical tests are described in figure legends. GraphPad Prism version 9 or Python were used for all analyses, with $P$ values less than 0.05 indicating statistically significant differences. The $IC_{50}$ values were determined by fitting dose–response curves with four-parameter logistic regression.

## Data availability

This study includes no data deposited in external repositories.

The source data of this paper are collected in the following database record: biostudies:S-SCDT-10_1038-S44321-025-00217-3.

## Peer review information

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

## Acknowledgements

The authors would like to thank Nancy M. Bonini (Department of Biology, University of Pennsylvania, Philadelphia, PA, USA.) for providing (CCTG)n flies. We thank Bloomington Drosophila Stock Center, Vienna Drosophila Resource Center, and Tsing Hua Fly Center for fly stocks with the help of Dr. Xuan Guo. The authors also thank The authors thank the Electron Microscope Rooms at Hunan Normal University and Hunan Agricultural University for assistance with SEM imaging. The funding for this project was provided by the grant of the National Natural Science Foundation of China (grant no. 82071273, 82271906).

## Author contributions

**Yingbao Zhu**: Conceptualization; Data curation; Software; Formal analysis; Supervision; Validation; Investigation; Visualization; Methodology; Writing—original draft; Writing—review and editing. **Shengwei Xiao**: Resources; Data curation. **Xinxin Guan**: Data curation. **Haitao Deng**: Data curation. **Liqiang Ai**: Validation; Visualization; Methodology. **Kaijing Fan**: Methodology. **Jin Xue**: Conceptualization. **Guangxu Li**: Data curation. **Xiaoxue Bi**: Formal analysis; Supervision. **Qiao Xiao**: Data curation. **Yuanjiang Huang**: Methodology. **Lin Jiang**: Data curation. **Wen Huang**: Conceptualization; Data curation. **Peng Jin**: Formal analysis; Funding acquisition; Methodology. **Ranhui Duan**: Conceptualization; Resources; Supervision; Funding acquisition; Writing—original draft.

Source data underlying figure panels in this paper may have individual authorship assigned. Where available, figure panel/source data authorship is listed in the following database record: biostudies:S-SCDT-10_1038-S44321-025-00217-3.

## Disclosure and competing interests statement

The authors declare no competing interests.

# Expanded View Figures

**Figure EV1. TDP1/gkt mutation alleviated neurodegeneration and muscle atrophy in DM2 flies.**

(A) Quantification of the average counts per day in flies across different genotypes from 15-days-old to the 25-days-old. Five biological replicates. Data are mean ± SD. Two-tailed, unpaired *t* test. (B, C) The average counts of per 30 min in flies across different genotypes from 15-days-old to the 25-days-old. Fifteen animals per sample. (C) Quantification from five biological replicates. Data are mean ± SD. Two-tailed, unpaired t-test. ****$P < 0.0001$ $(CCTG)_{720}$ versus $(CCTG)_{720} + gkt\ mut$. (D, E) Representative images of paraffin-embedded adult thoraces showing IFMs of flies at 1, 7, 15-days-old. Scale bars 100 μm. (E) Quantification at least ten independent thoraces of each sample/experiment, 3 experiments. Data are mean ± SD. Two-tailed, unpaired *t* test. **$P = 0.084$ 1-day-old versus 7-days-old $(CCTG)_{720}$, ***$P = 0.00042$ 7-days-old versus 15-days-old $(CCTG)_{720}$. (F–H) Representative images of paraffin-embedded adult thoraces showing IFMs of flies at 15-days-old. Black arrows, holes and fragmentations. Scale bars 100 μm. (G, H) Quantification at least ten independent thoraces of each sample/experiment, three experiments. Data are mean ± SD. Two-tailed, unpaired *t* test. ****$P < 0.0001$ $(CCTG)_{16}$ versus $(CCTG)_{720}$, ****$P < 0.0001$ $(CCTG)_{720}$ versus $(CCTG)_{720} + gkt\ mut$. (I) Quantification of climbing index across different genotypes at 15-days-old from five biological replicates. Data are mean ± SD. Two-tailed, unpaired *t* test. ****$P < 0.0001$ $(CCTG)_{16}$ versus $(CCTG)_{720}$, ****$P < 0.0001$ $(CCTG)_{720}$ versus $(CCTG)_{720} + gkt\ RI\ 1$, ****$P < 0.0001$ $(CCTG)_{720}$ versus $(CCTG)_{720} + gkt\ RI\ 2$. (J, K) Representative images of the compound eyes across different genotypes at 15-days-old. Top, LM images. Middle and Bottom, SEM images. Scale bars 20 μm. (K) Quantification at least sixty independent eyes of each genotype/experiment, three experiments. Data are mean ± SD. Two-tailed, unpaired *t* test. ****$P < 0.0001$.

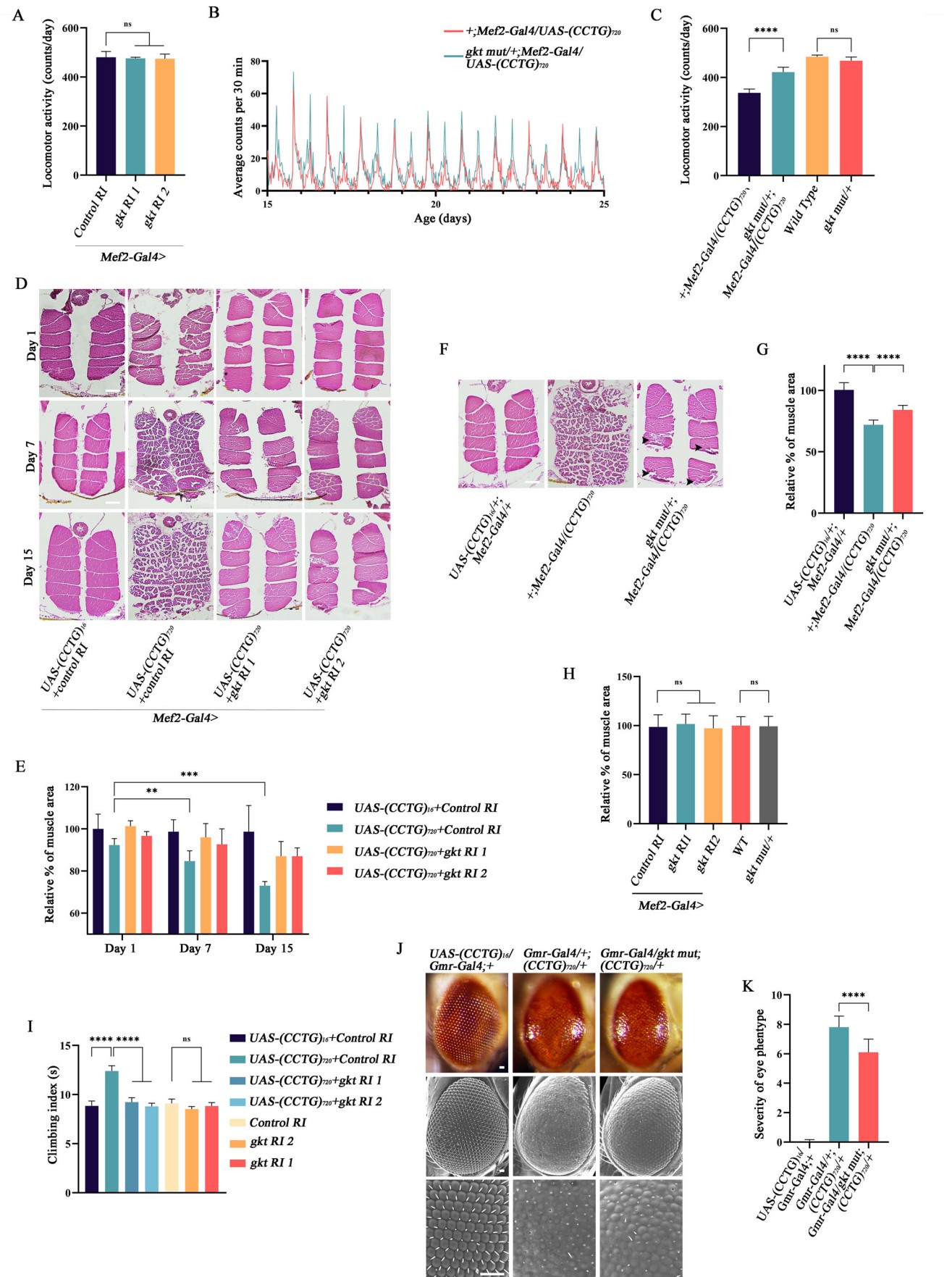

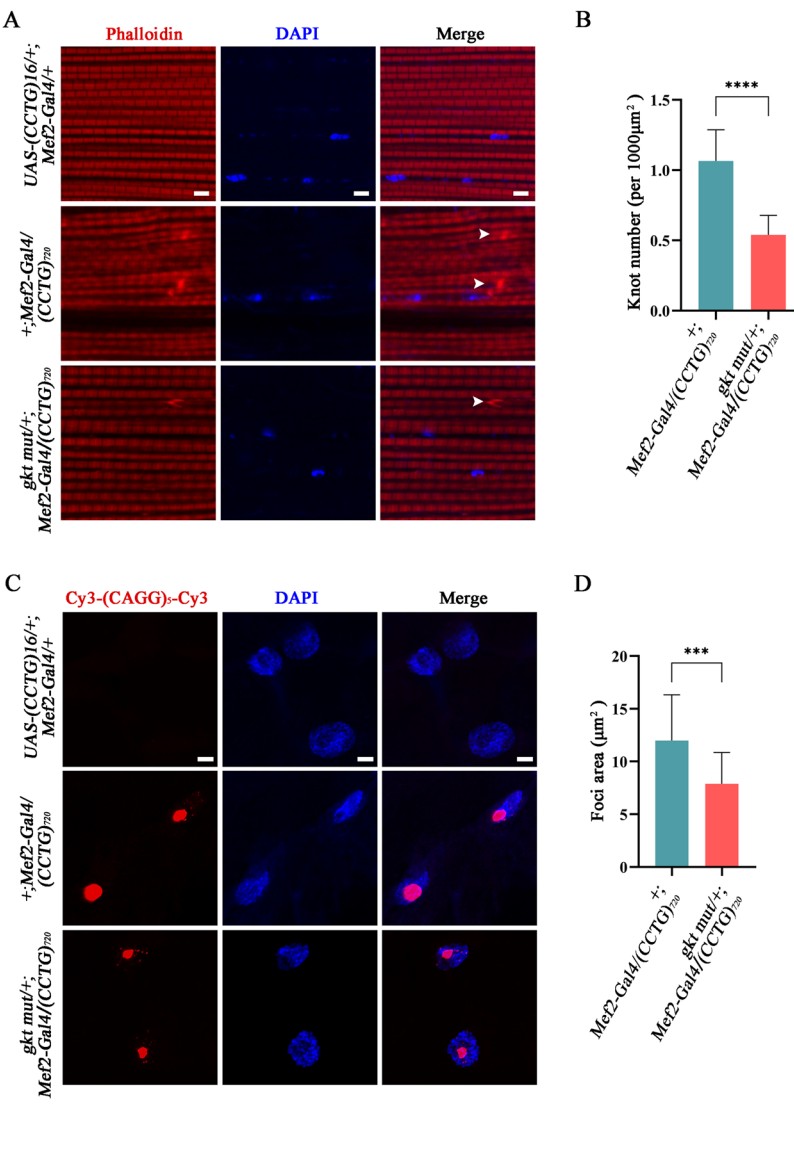

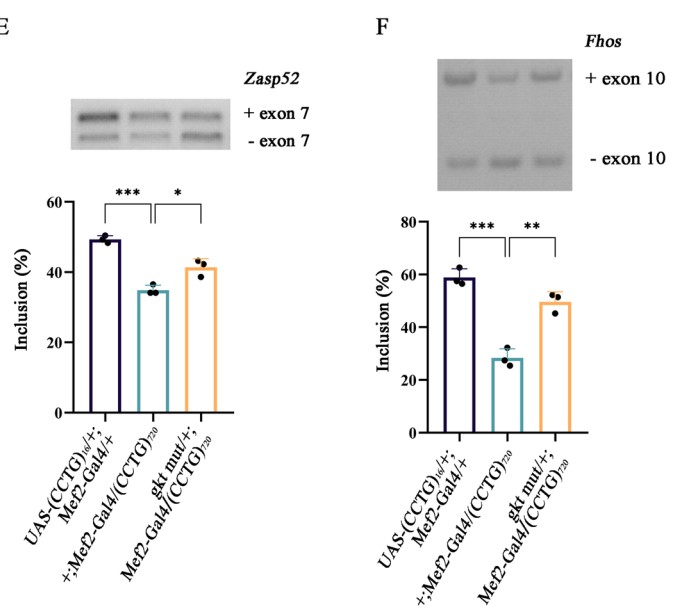

**Figure EV2.  TDP1/gkt mutation rescues molecular pathology of DM2.**

(A, B) Representative IF images of thoracic muscle fibers across different genotypes at 15-days-old. White arrows, action blobs. Scale bars 5 μm. (B) Quantification at least ten animals of each genotype. Data are mean ± SD. Two-tailed, unpaired *t* test. ****$P$ < 0.0001. (C, D) Representative Confocal images of CCUG-repeat foci with (CAGG)$_5$ probe across different genotypes at body-wall muscles of third instar larvae. Scale bars 5 μm. (D) Quantification at least ten animals of each genotype/experiment, three experiments. Data are mean ± SD. Two-tailed, unpaired *t* test. ***$P$ = 0.0003. (E) Representative gel image of endogenous *Zasp52* exon 7 with RT-PCR in flies at 15-days-old. Tissue from ten animals per sample/experiments, 5 experiments. Lower panel, quantification of *Zasp52* exon 7 inclusion from three biological replicates. Data are mean ± SD. Two-tailed, unpaired *t* test. *$P$ = 0.0162, ***$P$ = 0.0001. (F) Representative gel image of endogenous *Fhos* exon 10 with RT-PCR in flies at 15-days-old. Tissue from ten animals per sample/experiment, 5 experiments. Lower panel, quantification of *Fhos* exon 10 inclusion from three biological replicates. Data are mean ± SD. Two-tailed, unpaired *t* test. **$P$ = 0.021,***$P$ = 0.0004.

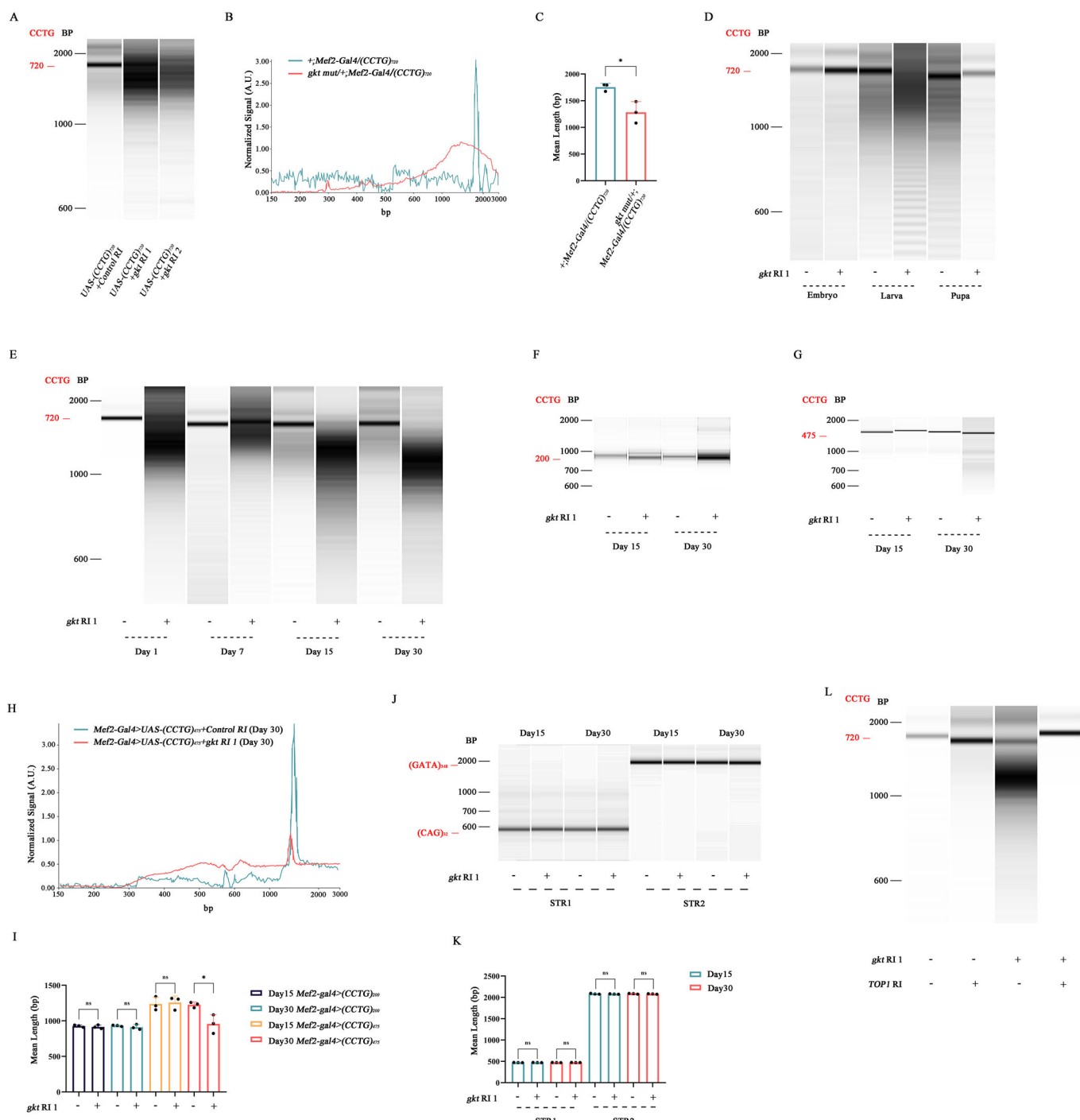

**Figure EV3.  TDP1/gkt loss of function induce large CCTG repeats contraction but not affect stability of endogenous STR loci.**

(A) Representative gel image of PCR across different genotypes at 15-days-old from bioanalyzer. Tissue from three animals per sample. (B, C) Bioanalyzer quantitation of CCTG repeat lengths across different genotypes at 15-days-old. Tissue from three animals per sample. (B) Quantification of mean length of CCTG repeats from three biological replicates. Data are mean ± SD. Two-tailed, unpaired *t* test. *P = 0.0182. (D) Representative gel image of PCR across different genotypes at embryo, larva, and pupa. Embryo from ten animals per sample. Larva and pupa from five animals per sample. (E) Representative gel image of PCR across *Mef2-Gal4 > UAS-(CCTG)₇₂₀ +Control RI* and *Mef2-Gal4 > UAS-(CCTG)₇₂₀+gkt RI 1* at 1, 7, 15, 30-days-old. Tissue from three animals per sample. (F) Representative gel image of PCR across *Mef2-Gal4 > UAS-(CCTG)₂₀₀ +Control RI* and *Mef2-Gal4 > UAS-(CCTG)₂₀₀+gkt RI 1* at 15-days-old and 30-days-old. Tissue from three animals per sample. (G) Representative gel image of PCR across *Mef2-Gal4 > UAS-(CCTG)₄₇₅ +Control RI* and *Mef2-Gal4 > UAS-(CCTG)₄₇₅+gkt RI 1* at 15-days-old and 30-days-old. Tissue from three animals per sample. (H) Bioanalyzer quantitation of CCTG repeat lengths across *Mef2-Gal4 > UAS-(CCTG)₄₇₅ +Control RI* and *Mef2-Gal4 > UAS-(CCTG)₄₇₅+gkt RI 1* at 30-days-old. Tissue from three animals per sample. (I) Quantification of mean length of CCTG repeats from three biological replicates. Data are mean ± SD. Two-tailed, unpaired *t* test. *P = 0.0263. (J, K) Representative gel image of STR1 and STR2 PCR across *Mef2-Gal4 > UAS-(CCTG)₇₂₀ +Control RI* and *Mef2-Gal4 > UAS-(CCTG)₇₂₀+gkt RI 1* at 15-days-old and 30-days-old. Tissue from three animals per sample. (K) Quantification of mean length of GATA and CAG repeats from three biological replicates. Two-tailed, unpaired *t* test. (L) Representative gel image of across different genotype at 15-days-old. Tissue from three animals per sample.

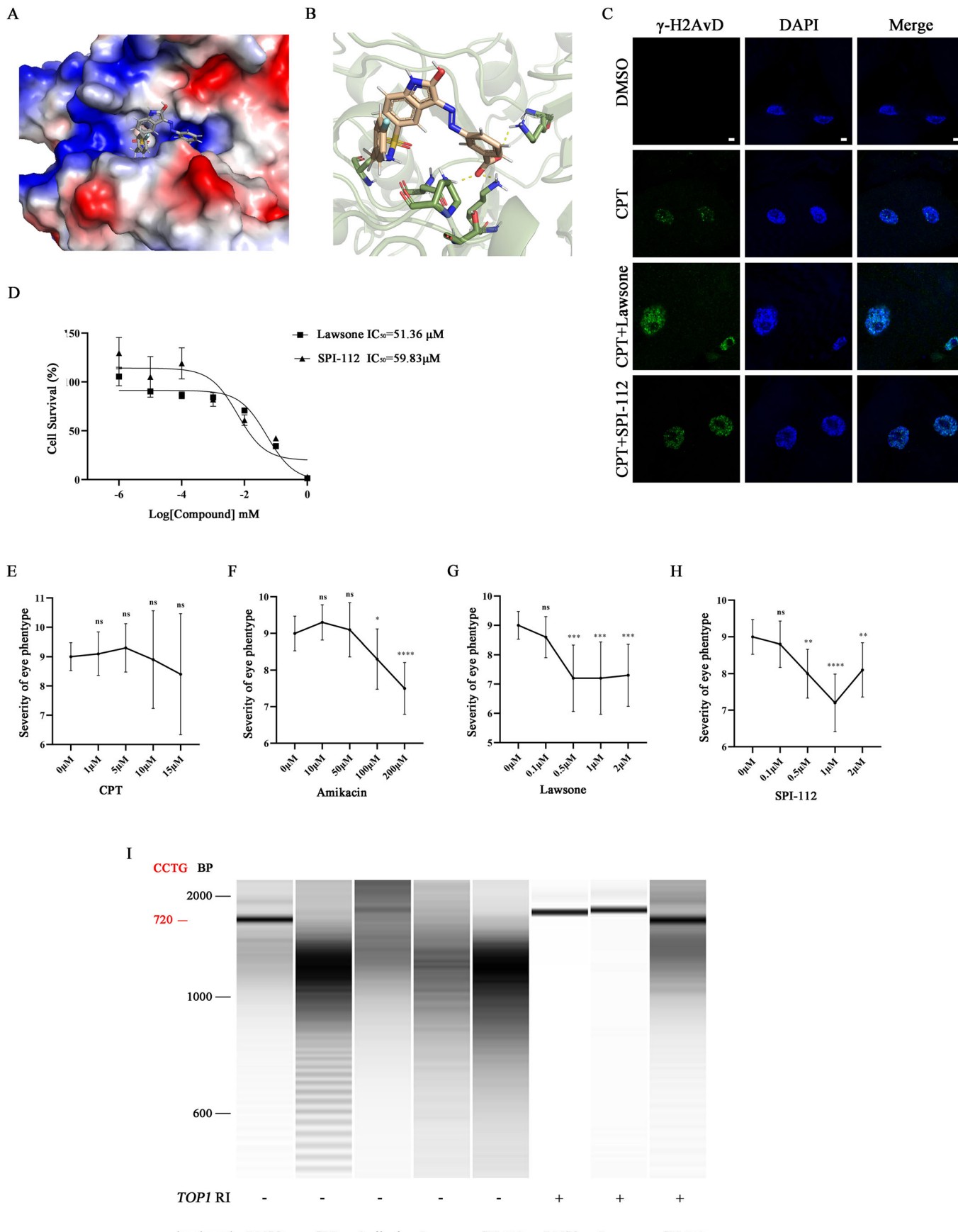

◀ **Figure EV4.   Small molecules predicted to bind TDP1 and cytotoxicity.**

(A) Surface electrostatic potentials of predicted crystal structure of human TDP1/SPI-112 complex. Red depicts a positive partial charge on the surface, blue depicts negative partial charge and gray shows neutral/lipophilic areas. (B) Docking model of Lawsone bind to TDP1 and SPI-112. Dotted yellow line: hydrogen bond, Dotted green line: Pi bond. Blue: N atom, Red: O atom, White: H atom, Green: protein skeleton, Gold: compound skeleton. (C) Representative IF images of body-wall muscles of third instar larvae from WT flies. Scale bars 5 μm. (D) Dose-survival analysis of Lawsone and SPI-112. The HEK293 cells were treated with purified TDP1 enzyme 0, 0.001, 0.01, 0.1, 1, 10, 100, 1000 μM for 72 h. Data are mean ± SD from three biological replicates. $IC_{50}$ values were calculated using log(inhibitor) compared with normalized survival rate (variable slope), using the program Prism, and are given in graphs. (E) Quantitative analysis of compound eye phenotypic severity of *Gmr-Gal4 > UAS-(CCTG)$_{720}$* after treatment with 0, 1, 5, 10, 15 μM CPT, at least twenty independent eyes of each sample/experiment, 3 experiments. Flies treated with 20 μM CPT were lethal. Data are mean ± SD. Two-tailed, unpaired *t* test. (F) Quantitative analysis of compound eye phenotypic severity of *Gmr-Gal4 > UAS-(CCTG)$_{720}$* after treatment with 0, 10, 50, 100, 200 μM Amikacin, at least twenty independent eyes of each sample/experiment, 3 experiments. Flies treated with 400 μM Amikacin were lethal. Data are mean ± SD. Two-tailed, unpaired *t* test. *$P = 0.0314$, ****$P < 0.0001$. (G) Quantitative analysis of compound eye phenotypic severity of *Gmr-Gal4 > UAS-(CCTG)$_{720}$* after treatment with 0, 0.1, 0.5, 1, 2 μM Lawsone, at least twenty independent eyes of each sample/experiment, 3 experiments. Data are mean ± SD. Two-tailed, unpaired *t* test. ***$P = 0.0002$ 0 μM versus 0.5 μM, ***$P = 0.0004$ 0 μM versus 1 μM, ***$P = 0.0002$ 0 μM versus 2 μM. (H) Quantitative analysis of compound eye phenotypic severity of *Gmr-Gal4 > UAS-(CCTG)$_{720}$* after treatment with 0, 0.1, 0.5, 1, 2 μM SPI-112, at least twenty independent eyes of each sample/experiment, 3 experiments. Data are mean ± SD. Two-tailed, unpaired *t* test. **$P = 0.0011$ 0 μM versus 0.5 μM, ****$P < 0.0001$ 0 μM versus 1 μM, **$P = 0.0044$ 0 μM versus 2 μM. (I) Representative gel image of across different genotype with treatment of different compounds at 15-days-old. Tissue from three animals per sample.

     

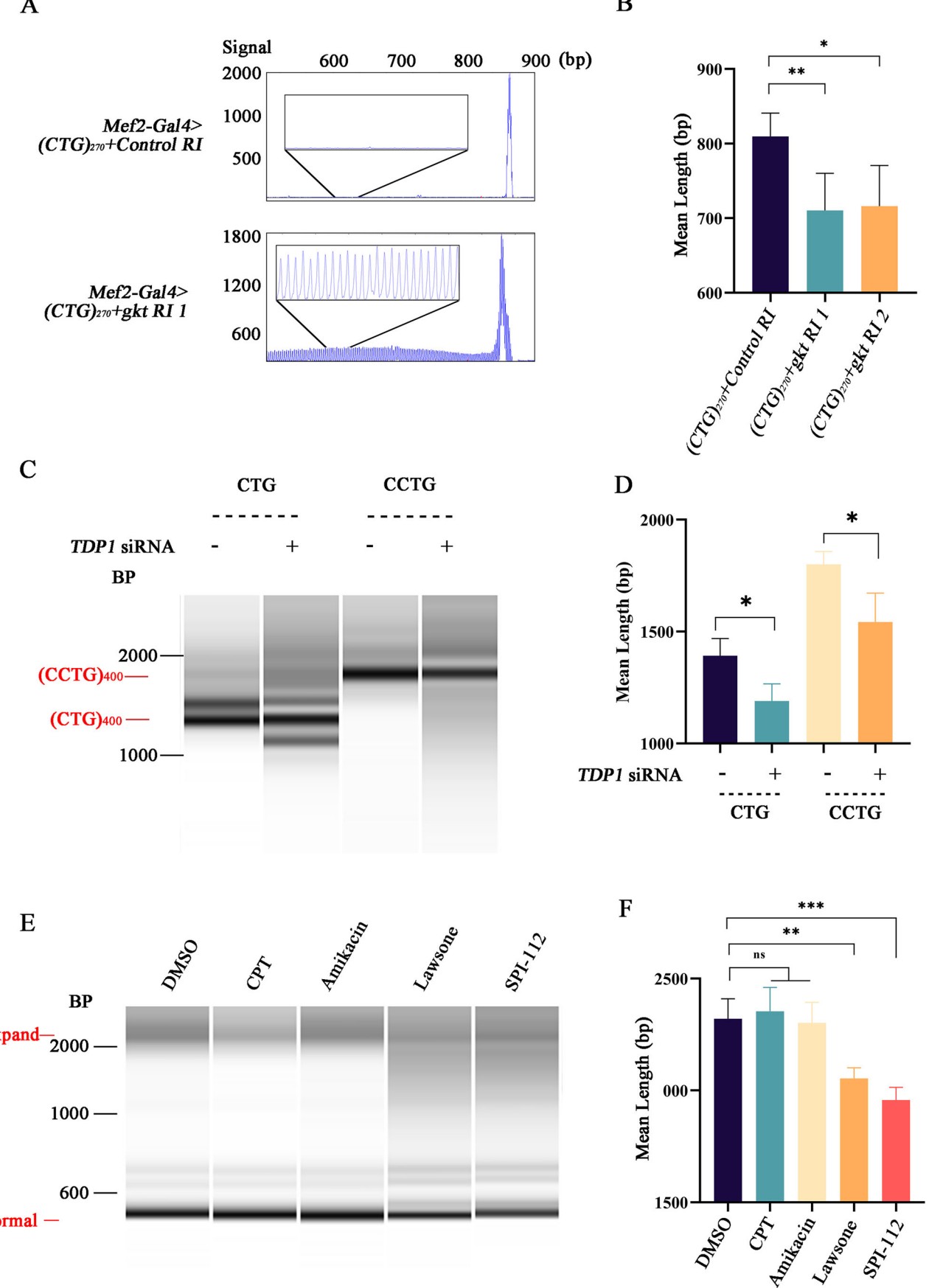

◀  **Figure EV5.  TDP1/gkt inhibition induce CTG/CCTG repeats contraction in other repeat expansion models.**

(A, B) Representative capillary electropherograms across different genotypes at 60-days-old. Tissue from three animals. (B) Quantification of mean length of CTG repeats from five biological replicates. Data are mean ± SD. Two-tailed, unpaired *t* test. *$P = 0.0103$, **$P = 0.0054$. (C, D) Representative gel image of PCR across different genotypes Hek293 cell line following 7 days of treatment from bioanalyzer. (D) Quantification of mean length of CTG/CCTG repeats from three biological replicates. Data are mean ± SD. Two-tailed, unpaired *t* test. *$P = 0.0324$ (CTG)$_{400}$ cell line, *$P = 0.0341$ (CCTG)$_{400}$ cell line. (E, F) Representative gel image of DM1 patient-derived lymphocytes following 15 days treatment of different compounds from bioanalyzer. (F) Quantification of (E) from three biological replicates. Data are mean ± SD. One-way ANOVA. **$P = 0.0082$ DMSO versus Lawsone, ***$P = 0.001$ DMSO versus SPI-112.

