## [Peer Review File · EMBO Molecular Medicine]

Modulating CCTG Repeat Expansion Toxicity in DM2 Drosophila Model Through TDP1 Inhibition

Yingbao zhu, Shengwei Xiao, Xinxin Guan, Haitao Deng, Liqiang Ai, Kaijing Fan, Jin Xue, Guangxu Li, Xiaoxue Bi, Qiao Xiao, Yuanjiang Huang, Lin Jiang, Wen Huang, Peng Jin, and Ranhui Duan

Corresponding authors: Ranhui Duan (duanranhui@sklmg.edu.cn) , Peng Jin (peng.jin@emory.edu)

Review Timeline:

Submission Date:	4th Apr 24
Editorial Decision:	10th May 24
Revision Received:	18th Dec 24
Editorial Decision:	27th Jan 25
Revision Received:	1st Mar 25
Accepted:	5th Mar 25

Editor: Zeljko Durdevic

Transaction Report:

10th May 2024

Dear Prof. Duan,

Thank you for the submission of your manuscript to EMBO Molecular Medicine. We have now received feedback from the three reviewers who agreed to evaluate your manuscript. All three referees recognize interest of the study but also raise important and partially overlapping concerns that should be addressed in a major revision. During our cross-commenting session referees agreed that the focus of the revision should be on validating main findings in a different disease model (e.g. human cell lines), evaluating results in other repeat diseases, providing controls for what is happening to the transgenic construct used and key factors around myotonic dystrophy, and providing more details about the methodologies and experimentation. If you would like to discuss further the points raised by the referees, I am available to do so via email or video. Let me know if you are interested in this option.

We would welcome the submission of a revised version within three to six months for further consideration. Please let us know if you require longer to complete the revision.

I look forward to receiving your revised manuscript.

Yours sincerely,

Zeljko Durdevic

We require:

- 1) A .docx formatted version of the manuscript text (including legends for main figures, EV figures and tables). Please make sure that the changes are highlighted to be clearly visible.
- 2) Individual production quality figure files as .eps, .tif, .jpg (one file per figure). For guidance, download the 'Figure Guide PDF': (<https://www.embopress.org/page/journal/17574684/authorguide#figureformat>).
- 3) A .docx formatted letter INCLUDING the reviewers' reports and your detailed point-by-point responses to their comments. As part of the EMBO Press transparent editorial process, the point-by-point response is part of the Review Process File (RPF), which will be published alongside your paper.
- 4) A complete author checklist, which you can download from our author guidelines (<https://www.embopress.org/page/journal/17574684/authorguide#submissionofrevisions>). Please insert information in the checklist that is also reflected in the manuscript. The completed author checklist will also be part of the RPF.
- 5) Please note that all corresponding authors are required to supply an ORCID ID for their name upon submission of a revised

manuscript.

6) It is mandatory to include a 'Data Availability' section after the Materials and Methods. Before submitting your revision, primary datasets produced in this study need to be deposited in an appropriate public database, and the accession numbers and database listed under 'Data Availability'. Please remember to provide a reviewer password if the datasets are not yet public (see <https://www.embopress.org/page/journal/17574684/authorguide#dataavailability>).

13) Author contributions: You will be asked to provide CRediT (Contributor Role Taxonomy) terms in the submission system. These replace a narrative author contribution section in the manuscript.

14) A Conflict of Interest statement should be provided in the main text.

Please also suggest a striking image or visual abstract to illustrate your article as a PNG file 550 px wide x 300-800 px high.

***** Reviewer's comments *****

Referee #1 (Remarks for Author):

Myotonic dystrophy (DM), the most common muscular dystrophy in adults, is an inherited muscle disease caused by expanded CTG (causing DM type 1) or CCTG (cause of DM type 2) repeats that are transcribed into mutant RNA titrating the MBNL splicing regulators. Consequently, myotonic dystrophy is characterized by numerous splicing alterations due to the lower quantity of functional MBNL splicing factors. Importantly, there is no cure for myotonic dystrophy (Thornton et al., Lancet Neurology 2024).

In this work, Zhu and collaborators performed a screen of small molecules in a drosophila model expressing expanded CCTG repeats and identified the TDP1 pathway as a potential therapeutic approach for DM type 2. As TDP1 is involved in DNA repair, this work is timely considering recent publications demonstrating the importance of somatic expansions in tri-nucleotide expansion diseases, as well as recent evidences that modulating mismatch repair proteins, RPA, FAN1 and other DNA repair mechanisms could represent a therapeutic hope for these diseases (Gall-Duncan et al., 2023; O'Reilly et al., 2023; Matlik et al., 2024, etc.). Overall, this is an interesting, well controlled and well performed work. My main concern is that this draft is entirely focused on DM2 and fly model, without validation in a human model (cf. below).

Major points:

As DM1 is the major subtype of myotonic dystrophy, some experiments assessing the effect of modulating TDP1 pathway in CTG expressing flies would be a strong plus to this work.

Similarly, modulating TDP1 in human muscle cell models of DM1 or DM2 through either inhibitory drug, siRNA or CRISPR Cas9 approaches would be an important validation. Of technical interest, splicing alterations and delay in differentiation are observed in DM1 differentiated myoblasts, while DM2 are showing only very mild alterations, but RNA foci and recruitment of MBNL1 can be assessed in both types.

Minor points:

Results of the chemical screen is brief and lack details, as only the final 16 compounds are briefly cited and presented in sup table 1, without any detailed indication of their respective efficiency. Similarly, the original 140 positive molecules, and the 31 selected for further analyzes are not or very briefly described.

Figure 3C, RNA FISH could be completed with immunofluorescence against the Mbl protein to show its titration and lesser capture in GKT flies.

Figure 3E, other splicing could be tested (Fhos exon 16, Serca exon 13, cf Cerro-Herreros et al., 2017, etc).

Introduction sentences 103 to 119, this is a well written and interesting part, but ill-placed and maybe more appropriate for the discussion ?

Referee #2 (Comments on Novelty/Model System for Author):

The manuscript submitted by Zhu et al. presents a fly-based chemical screening approach where a new target is identified and validated as potentially therapeutic for rare DM2 disease, with some small molecules also identified as potential target modifiers.

Novelty of the results is high in the area of myotonic dystrophy pathologies (DM1 and DM2), since no drugs are still available for patients. Moreover, the manuscript focuses on the less studied, DM2.

The medical impact at this point can be considered medium since the results presented involve a project still in an early stage of development. Results presented, although promising, still need of more confirmative results.

Otherwise, the manuscript displays deficiencies in the technical aspect, with poor methodological descriptions for some approaches that make it difficult to follow and understand. Important to note, that for the large number of experiments executed, always the same Two-tailed unpaired t-test statistical approach is used. Not clear that appropriated for all of them.

Finally, the repeat expansion fly models were considered "repeat stable" in most cases, with only one to two reports indicating some low instability levels. This description was also done for the specific model used here. The manuscript results are changing this paradigm, which is highly surprising, and not well-addressed in the manuscript. The use of additional confirmatory models is highly suggested for the publication of the results.

Referee #2 (Remarks for Author):

The manuscript submitted by Zhu et al. and titled "Modulating CCTG Repeat Expansion Toxicity in DM2 Drosophila Model Through TDP1 Inhibition" introduces a novel therapeutic target in a repeat expansion disease, DM2, from a fly-based chemical screening approach. The results presented are promising, not only because of identifying a totally novel disease target, but also presenting molecules able to efficiently target it, and the most exciting result, the in vivo description of a DNA repeat contractions as a result of the target modulation.

However, the whole manuscript, including the writing of some sections but also, the experimental approach displayed, contains several major concerns at different levels, making difficult to fully validate the results presented. The manuscript needs of large extended or/and additional experimental description and execution before publication.

Major concerns

- The introduction section is written in a dispersed way, without order, and lacking some key recent information. Some examples are:

- (i)The random rationale to choose DM2, including too much information about DM1; that connects with...

- (ii)...not explanation at all about why drugs being developed for DM1 are not going to be useful for DM2 in case of success

- (iii)Poor and non-updated description of drug development in DM

- (iv)Poor description of fly DM2 models, lacking description for some already published, and also poor description for the one used, for example, writing about some of the lethal phenotypes observed.

- (v)Too much writing in the intro, involving results or discussion text (lines 94-102 and 120-126). This text should be relocated.

- The Methods section needs to include larger and more detailed information for different aspects:

- (i)The manuscript turns mainly around fly-based experiments with large number of stocks used. It is implied that several of them needed of the generation and use of new transgenic flies including the expression of up to three different transgenes at the same time. The manuscripts lacks description of the crosses and new lines generated making it very difficult to follow and understand the results achieved.

- (ii)The hits from the second screening (semi-lethality) is based on compounds able to increase >1.5 change fold in the flies that can reach the adult stage. But, how variable was the semi-lethality assay in terms of born flies' percentage? Are the authors seeding always the same number of embryos/larvae? More detail is needed to know about the hits identified.

- (iii)In vivo functional fly-based assays are poorly described, and results have difficult interpretation. Interesting to add videos of the locomotor or climbing assays as Supplementary data. Examples of difficult to evaluate data are: Figure 2A: it is clear present an aging effect for the repeat stocks, more pronounced for the 16 repeat flies that during the 10 days of the assays display a strong reduction of their locomotor levels, also observed for the 720 flies. How are the authors considering it? Also, how

considering that during all this time, the 720 flies should be dying... or not? Is there a semi-lethality phenotype, but once the flies born they have the same survival rate during adults? Also these locomotor peaks are so heterogeneous during the whole assay. Otherwise, the level of IFM muscle destruction in 720 repeat stock (Figure 2B and others) is so high at 15 days that these flies shouldn't fly. Why not doing a flight assay? Climbing and locomotor could be linked to other muscles involvement. Why not check what is happening in other fly muscles (histology)?

(iv) Figure 3 is displaying the presence of ribonuclear foci and alternative splicing (AS) issues in the DM2 720 repeat expansion stock. However, the foci observed are so high and show a strong signal. Very different from what is observed in human or mouse DM samples. What is the number of foci x cell? Could be transcription signal what it is observed? Downregulation of TDP1 is displaying reduction of the "foci" area, not the number. Explanation? Regarding AS, the gels are not compare the targeted genes with gene control expressions (actin, GAPDH).

(v) The amplification of large repeat tracts by standard (PCR) methods has been displayed as highly difficult for different motifs. No publications indicating the amplification of so large CCTG repeat tract (in this case a tetranucleotide). Currently, having the possibility of long read sequencing methods successful with long repeat tracts, it should be here included to confirm, again, that the results shown are not PCR amplification issues since the lengths representation is so different depending on the stocks, some are displaying repeated peaks, and in other cases we see a straighter line (Fig. 4) Authors should explain about it. Similar concerns with the visualization of repeat bands with the presence of black areas where sizing seems very difficult, or in other cases, very regular bands displayed, shorter, but also larger (expansions?). Here one "silly" question. In the figures the 720 repeats band is located below the 2000 bp. How is this possible? ($720 \times 4 \text{ bases} = 2880 \text{ bp}$)

•Results

(i) The manuscript points out TDP1 as a novel target for DM2. Its downregulation triggers repeat contraction and reduction of toxic phenotypes in flies. BUT, the manuscript does not include data showing the expression/protein status of the TDP1 homolog in the CCTG repeat stocks in the different approaches executed. It is not shown what happens with the expression of this gene in each CCTG repeat stock or in each developmental stage evaluated in flies, where significant contraction differences are observed. Also at the end, after the computational screening, when some TDP1 inhibitors were evaluated in flies. Thus, it is very important to have pieces of evidence indicating that TDP1 modulation is happening in the model system used, but in the manuscript, no figures or writing showing how effective was the expression or protein reduction levels when using these stocks to correlate with DM2 phenotypes readings. Also not known if differences between the different TDP1 downregulations.

(ii) TDP1 downregulation is strongly connected to repeat contraction and alleviation of DM2-related phenotypes through all the manuscript. If this true, why not test overexpression of TDP1 and check for repeat expansions and the appearance of stronger disease-based phenotypes? This would confirm the direct connection between TDP1 and repeat stability.

(iii) Likewise, to confirm the hypothesis proposed, the manuscript needs to include also biological evidence for the target and the molecules proposed in different DM2 models (cell lines or mouse models).

(iv) The mechanism proposed related to the downregulation of TDP1 is not presented as CCTG-specific. Other repeat motifs must be evaluated for TDP1 downregulation to support results presented in DM2 flies.

(v) Comparing with DM1, DM2 is characterized by less severe human phenotypes. However, DM2 flies display robust toxic phenotypes, even higher than DM1 flies. New locomotor, climbing, and sarcomere histology phenotypes are here described by first time for the DM2 flies pointing out stronger toxic phenotypes. Good characterization and controls are needed. For example, no controls indicate that muscle Gal4 itself is introducing some toxicity degree in the presence of the large repeat. Together, these DM2 flies seems not to be a good model regarding the whole DM2 pathology, although not discarding its valid use for targeting repeat toxicity issues.

(vi) The project rationale lacks of detail in some key steps. Starts with a small molecule screening, with some hits identified after two rounds of screening. But here, the authors do not move forward, as is common for this type of screenings, validating some hits in further steps. Can authors explain why? Here, it is proposed a new selection step based in choosing compounds with notable bioactivity levels. What authors mean with "notable bioactivity levels"? What was the specific rationale/threshold to choose the 16 from the 31? And finally, authors check for biological target coincidence, moving to finally define TDP1 as the gene target, common for five of the hits defined as all inhibitors, to be validated. The manuscript has to include references or links that must show the connection for the five compounds and their TDP1 inhibitor role. Also, at this point, the authors explain that ESR1 target is discarded because some hits inhibitors but other activators. Fig 1C shows that some other genes that showed up, including ESR1, share hits with TRP1, suggesting a pathway involved, not only a gene. But explanation for this is lacking into the rationale.

(vii) The images for the IFM muscle packages from the 720 stock displays a very "destroyed" muscle. This is not seen from the DM1 models, where atrophy is evident but as of loss of muscle area. Would be great to know what is contained in the "holes" between the muscles that still remains. Additional staining to see if the holes contain connective tissue (Masson or other histology labelling) should be performed to discard technical issues (poor fixation, etc) during the protocol. Linked to the last,

some fusion of muscle packages is observed indicating technical issues (less than 6 packages observed in Fig 2D and Fig EV1D indicating bad fixation process).

(viii) The project finish with a virtual screening looking for molecules binding TDP1 and acting as inhibitors. Some were found and tested. But, why not using the ones identified in the first screening, also pointed out as TDP1 inhibitors? Did they show up in the virtual screening? They should, at least their MoA different but also triggering TDP1 downregulation

Other concerns

(i) What does it mean? Line 158-160 "...restored locomotor activity and climbing ability without affecting normal locomotor activity"

(ii) The manuscript is presenting consistent results indicating the identification of a novel biological target, some molecules able to modulate it in a very promising direction, triggering repeat contractions. This data is the type of data that patients are waiting for, with clear translatability to drugs developments. However, the manuscript not indicate if these assets are protected (patent?). This should be done previously to publication to have chances of reaching patients successfully.

Referee #3 (Remarks for Author):

1) The technical quality and statistical analysis is good with sufficient numbers and repetitions. For the locomotor graphs, it would help to know what the counts are about (I am assuming it is the number of times the alarm was set off by a fly breaking the beam). This could be made clearer in the figure legend.

2) The identification of TDP1 through genetic analysis as well as compounds inhibiting its activity and their subsequent effects on multiple phenotypes was novel, convincing and impactful.

3) The medical impact is too early to comment on, though these experiments provide avenues for further investigations in mouse models, for example.

What is lacking and I think is essential, is to show:

1) what happened to the expression of the toxic RNA (transgene RNA)

2) what happened to Mbl (drosophila MBNL) expression

3) was the stability of the toxic RNA affected?

4) was the stability of the transgene construct affected. That is, is it just the repeat length was affected or was the transgene degraded in other ways.

There are a few grammatical errors and typos throughout that need to be checked.

Referee #1 (Remarks for Author):

Myotonic dystrophy (DM), the most common muscular dystrophy in adults, is an inherited muscle disease caused by expanded CTG (causing DM type 1) or CCTG (cause of DM type 2) repeats that are transcribed into mutant RNA titrating the MBNL splicing regulators. Consequently, myotonic dystrophy is characterized by numerous splicing alterations due to the lower quantity of functional MBNL splicing factors. Importantly, there is no cure for myotonic dystrophy (Thornton et al., Lancet Neurology 2024).

In this work, Zhu and collaborators performed a screen of small molecules in a drosophila model expressing expanded CCTG repeats and identified the TDP1 pathway as a potential therapeutic approach for DM type 2. As TDP1 is involved in DNA repair, this work is timely considering recent publications demonstrating the importance of somatic expansions in tri-nucleotide expansion diseases, as well as recent evidences that modulating mismatch repair proteins, RPA, FAN1 and other DNA repair mechanisms could represent a therapeutic hope for these diseases (Gall-Duncan et al., 2023; O'Reilly et al., 2023; Matlik et al., 2024, etc.). Overall, this is an interesting, well controlled and well performed work. My main concern is that this draft is entirely focused on DM2 and fly model, without validation in a human model (cf. below).

Response: We sincerely thank you for the valuable suggestions regarding both the writing and experimental aspects of our manuscript. We have made modifications to the introduction section, supplemented the materials and methods section, and added relevant experiments involving the DM1 Drosophila model, Hek293 cell line containing repeat expansion, and patient lymphocyte model.

Major points:

As DM1 is the major subtype of myotonic dystrophy, some experiments assessing the effect of modulating TDP1 pathway in CTG expressing flies would be a strong plus to this work. Similarly, modulating TDP1 in human muscle cell models of DM1 or DM2 through either inhibitory drug, siRNA or CRISPR Cas9 approaches would be an important validation. Of technical interest, splicing alterations and delay in differentiation are observed in DM1 differentiated myoblasts, while DM2 are showing only very mild alterations, but RNA foci and recruitment of MBNL1 can be assessed in both types.

Response: Using the accessible model including DM1 Drosophila model, Hek293 cell models containing repeat traits, and DM1 patient-derived lymphocytes, we have made the following attempts (Fig EV5).

- 1、 In the DM1 Drosophila model, capillary electrophoresis revealed a distinct peak at 842 bp in (CTG)₂₇₀+control RI Drosophila. In contrast, a diffuse peak was observed in (CTG)₂₇₀+gkt RI1 Drosophila, indicating CTG repeat contraction (Fig EV5A,B). Additionally, FISH results showed that *gkt* RI reduced the area of RNA foci at body-wall muscles of third instar larvae (Appendix FigS7).
- 2、 Stable HEK293 cell models were developed to express (CTG)₄₀₀ and (CCTG)₄₀₀ repeats.

After siRNA-mediated *TDP1* inhibition combined with CPT treatment, both CTG and CCTG repeats exhibited contraction (Fig EV5C,D).

- 3、 Due to our budget and technical constraints, experiments involving iPSC-induced myoblasts require collaboration with external experts in the near future. Additionally, as there are no DM2 patients in China, we identified five DM1 patients, two of whom provided peripheral blood to establish lymphoid cell lines, selecting the one with longer repeat length for further experiments.

An immortalized lymphocyte line with (CTG)₅₀₀ was generated by EBV infection of peripheral blood lymphocytes from DM1 patients. Upon treatment with TDP1 inhibitors, we observed that the screened TDP1 inhibitors, *Lawson* and *SPI-112*, promoted CTG repeat contraction (Fig EV5E,F).

The mechanism of repeat contraction will be explored in the next steps, with the hypothesis that DNA breaks caused by TDP1 inhibition and R-loop structures work together to promote repeat contraction. It is suggested that TDP1 inhibition may reduce other types of repeats as well. Lymphocytes with CGG and CAG repeat expansions are currently being treated with TDP1 inhibitors. In the near future, iPSC-induced myoblasts will be established, either independently or in collaboration with partners, and these models will be used to investigate the mechanisms underlying repeat contraction across different repeat types.

Figure EV5. TDP1/gkt inhibition induce CTG/CCTG repeats contraction in other repeat expansion models. (A) Representative capillary electropherograms across different genotypes at 60-days-old. Tissue from three animals. (B) Quantification of mean length of CTG repeats from five biological replicates. Data are mean \pm SD. Two-tailed, unpaired t-test. *P=0.0103, **P=0.0054. (C) Representative gel image of PCR across different genotypes Hek293 cell line following 7 days of treatment from bioanalyzer. (D) Quantification of mean length of CTG/CCTG repeats

from three biological replicates. Data are mean \pm SD. Two-tailed, unpaired t-test. * P = 0.0324 (CTG)₄₀₀ cell line, * P = 0.0341 (CCTG)₄₀₀ cell line. (E) Representative gel image of DM1 patient-derived lymphocytes following 15 days treatment of different compounds from bioanalyzer. (F) Quantification of (F-G) from three biological replicates. Data are mean \pm SD. One-way ANOVA. ** P = 0.0082 DMSO versus Lawsone, *** P = 0.001 DMSO versus SPI-112.

Appendix Figure S7. *TDPI/gkt* knockdown reduced CUG toxic RNA aggregation. (A) Representative Confocal images of CUG-repeat foci with (CAG)₅ probe across different genotypes at body-wall muscles of third instar larvae. Scale bars 5 µm. (B) Quantification at least ten animals of each genotype. Data are mean \pm SD. Two-tailed, unpaired t-test. * P = 0.0433.

Minor points:

Results of the chemical screen is brief and lack details, as only the final 16 compounds are briefly cited and presented in sup table 1, without any detailed indication of their respective efficiency. Similarly, the original 140 positive molecules, and the 31 selected for further analyzes are not or very briefly described.

Response: Details of chemical screening have been added to the Material Methods section.

Added: “In the primary screen, 140 compounds were tested using the *24B-Gal4* driver, which induces broad expression of (CCTG)₇₂₀ from the mesoderm to muscle cells. This expression caused complete lethality in flies, allowing for the identification of compounds that could potentially rescue toxicity by influencing muscle development.” in line 410-414.

“In total, 31 compounds were identified in the secondary screen that exhibited varying degrees

of rescue effects against muscle toxicity. Among these, 16 compounds with target information available in the PubChem Compound database were selected for subsequent analysis.” in line 419-421.

Figure 3C, RNA FISH could be completed with immunofluorescence against the Mbl protein to show its titration and lesser capture in GKT flies.

Response: The FISH and Immunofluorescence has been added in Appendix Fig S3 A-B. Inspired by the work of Nancy et al., *UAS-hMBNL1-HA* drosophila was used to assess the ability of toxic RNA to sequester MBNL1/mbl proteins. As shown in Appendix Fig S3 A-B, MBNL1 protein colocalized with CCUG RNA in the nucleus, and *gkt* knockdown reduces this sequestration.

Appendix Figure S3. *TDP1/gkt* knockdown corrected the abnormal localization of MBNL1/mbl. (A) Representative image of colocalized nuclear foci FISH and MBNL1 immunofluorescence. Scale bars 5 μ m. (B) Quantification at least ten animals of each genotype. Data are mean \pm SD. Two-tailed, unpaired t-test. *** $P = 0.0005$, ** $P = 0.005$.

Figure 3E, other splicing could be tested (*Fhos* exon 16, *Serca* exon 13, cf Cerro-Herrerros et al., 2017, etc).

Response: Alternative splicing data has been added in Appendix Fig S4 A-B. The *Fhos* exon 16 and *Serca* exon 13 in the paper were tested. Consistent with our conclusions, the splicing abnormalities of these genes were restored by *gkt* knockdown (Appendix Fig S4 A-B).

Appendix Figure S4. TDP1/*gkt* loss-of-function rescued alternative splicing in DM2. (A) Representative gel image of endogenous *Fhos* exon 10 with RT-PCR in flies at 15-days-old. Tissue from ten animals per sample. Lower panel, quantification of *Fhos* exon 10 inclusion from three biological replicates. Data are mean \pm SD. Two-tailed, unpaired t-test. *** $P = 0.0005$ (CCTG)₁₆ versus (CCTG)₇₂₀, *** $P = 0.0007$ (CCTG)₇₂₀ versus (CCTG)₇₂₀ + *gkt RI 1*, *** $P = 0.0003$ (CCTG)₇₂₀ versus (CCTG)₇₂₀ + *gkt RI 2*. (B) Quantification of *Serca* exon 13 expression in flies at 15-days-old from three biological replicates. Data are mean \pm SD. Two-tailed, unpaired t-test. *** $P = 0.0002$ (CCTG)₁₆ versus (CCTG)₇₂₀, *** $P = 0.0006$ (CCTG)₇₂₀ versus (CCTG)₇₂₀ + *gkt RI 1*, ** $P = 0.0026$ (CCTG)₇₂₀ versus (CCTG)₇₂₀ + *gkt RI 2*.

Introduction sentences 103 to 119, this is a well written and interesting part, but ill-placed and maybe more appropriate for the discussion ?

Response: The section from sentences 103 to 119 has been moved to the third paragraph of the Discussion section, as suggested. The first paragraph of the Discussion summarizes our findings, and the second paragraph addresses the muscle phenotype of DM2 flies. This part, now in the third paragraph, discusses the mechanism of repeat expansion disorders, emphasizing that repeat expansion sequence is central to the pathology of these diseases.

Referee #2 (Comments on Novelty/Model System for Author):

The manuscript submitted by Zhu et al. presents a fly-based chemical screening approach where a new target is identified and validated as potentially therapeutic for rare DM2 disease, with some small molecules also identified as potential target modifiers.

Novelty of the results is high in the area of myotonic dystrophy pathologies (DM1 and DM2), since no drugs are still available for patients. Moreover, the manuscript focuses on the less studied, DM2.

The medical impact at this point can be considered medium since the results presented involve a project still in an early stage of development. Results presented, although promising, still need of more confirmative results.

Otherwise, the manuscript displays deficiencies in the technical aspect, with poor methodological descriptions for some approaches that make it difficult to follow and understand. Important to note, that for the large number of experiments executed, always the same Two-tailed unpaired t-test statistical approach is used. Not clear that appropriated for all of them.

Finally, the repeat expansion fly models were considered "repeat stable" in most cases, with only one to two reports indicating some low instability levels. This description was also done for the specific model used here. The manuscript results are changing this paradigm, which is highly surprising, and not well-addressed in the manuscript. The use of additional confirmatory models is highly suggested for the publication of the results.

Response: We sincerely thank you for your thoughtful and insightful suggestions regarding both the writing and experimental aspects of our manuscript. As suggested, we have revised the statistical analysis for the section comparing DMSO versus TDP1 inhibitors, which involves a multi-group comparison. The analysis method has been changed to one-way ANOVA. This change does not alter the conclusions of the study.

Referee #2 (Remarks for Author):

The manuscript submitted by Zhu et al. and titled "Modulating CCTG Repeat Expansion Toxicity in DM2 Drosophila Model Through TDP1 Inhibition" introduces a novel therapeutic target in a repeat expansion disease, DM2, from a fly-based chemical screening approach. The results presented are promising, not only because of identifying a totally novel disease target, but also presenting molecules able to efficiently target it, and the most exciting result, the in vivo description of a DNA repeat contractions as a result of the target modulation.

However, the whole manuscript, including the writing of some sections but also, the experimental

approach displayed, contains several major concerns at different levels, making difficult to fully validate the results presented. The manuscript needs of large extended or/and additional experimental description and execution before publication.

Major concerns

- The introduction section is written in a dispersed way, without order, and lacking some key recent information. Some examples are:

(i)The random rationale to choose DM2, including too much information about DM1; that connects with...

(ii)...not explanation at all about why drugs being developed for DM1 are not going to be useful for DM2 in case of success

Response: The Introduction has been reorganized to improve clarity and flow.

The first paragraph provides a brief overview of DM.

The second paragraph outlines the differences in the pathogenesis of DM1 and DM2, as well as the current status of DM1 treatment. It highlights that DM2 currently lacks effective therapeutic strategies and explains that, due to mechanistic differences, therapies for DM1 cannot be directly applied to DM2. To address this gap, the DM2 drosophila model was used for large-scale chemical screening.

The third paragraph presents information about the DM2 Drosophila model and TDP1.

The fourth paragraph introduces the primary focus of our study.

(iii)Poor and non-updated description of drug development in DM

Response: The information of phase III drugs for DM1 has been added in the Introduction section.

Add: “Metformin for the splicing defect (phase III)” in line 83-84. “IONIS-DMPKRx (preclinical)” Changed to “AOC 1001 (phase III)” in line 84.

Due to space constraints, we have referred a detailed review of DM1 drugs instead.

Add : “A more comprehensive list of DM1 drugs currently in development can be found in the review by Pascual-Gilabert et al.” in line 87-88.

(iv)Poor description of fly DM2 models, lacking description for some already published, and also poor description for the one used, for example, writing about some of the lethal phenotypes observed.

Response: The description of DM2 drosophila model has been added in the Introduction section.

Added: “In 2017, a DM2 Drosophila model expressing 1,100 CCTG repeats was constructed, with histological analysis revealing severe muscle reduction and cardiac dysfunction. A Drosophila model with 106 CCTG repeats was developed by Bergmann et al. , which displayed disruption of external eye morphology and an apoptotic response but lacked a muscle phenotype.” in line 95-99. Detailed descriptions of lethal phenotypes have been added to the Materials and Methods section.

Added: “In the primary screen, 140 compounds were tested using the *24B-Gal4* driver, which induces broad expression of (CCTG)₇₂₀ from the mesoderm to muscle cells. The expression caused complete lethality in flies, allowing for the identification of compounds that could potentially rescue toxicity by influencing muscle development. For the secondary screen, the *Mef2-Gal4* driver, primarily expressed in mature muscle, was employed. This driver resulted in semi-lethality, providing an easily quantifiable phenotype for further analysis.” in line 410-417.

Regarding the DM2 *Drosophila* phenotypes, Dr. Bonini's study primarily focused on the neurodegeneration in the compound eye, with muscle-related phenotypes examined only for the presence of foci in the body wall muscle of third instar larvae (Yu et al., 2015).

(v) Too much writing in the intro, involving results or discussion text (lines 94-102 and 120-126). This text should be relocated.

Response: Same as comment (i) and (ii) above; see response above.

• The Methods section needs to include larger and more detailed information for different aspects:

(i) The manuscript turns mainly around fly-based experiments with large number of stocks used. It is implied that several of them needed of the generation and use of new transgenic flies including the expression of up to three different transgenes at the same time. The manuscripts lacks description of the crosses and new lines generated making it very difficult to follow and understand the results achieved.

Response: The details of crosses and new line have been added to the Materials and Methods section.

Added: “*Mef2-Gal4* flies were crossed with *UAS-(CCTG)_n* flies, and the female offspring were subsequently crossed with *sco/cyo; TM3/TM6B* male flies. Recombinant offspring *Mef2-Gal4 > (CCTG)_n* were selected based on PCR analysis.” in line 398-400.

(ii) The hits from the second screening (semi-lethality) is based on compounds able to increase >1.5 change fold in the flies that can reach the adult stage. But, how variable was the semi-lethality assay in terms of born flies' percentage? Are the authors seeding always the same number of embryos/larvae? More detail is needed to know about the hits identified.

Response: During the initial protocol development, survival data for DMSO-treated flies remained consistently stable at around 15–20. Due to the limited amount of the compound available, cultures were maintained in 5 mL centrifuge tubes. In the secondary screening, compounds with survival counts exceeding 30–40 (approximately the top 20%) were selected. All experiments were performed in three times. For all other experiments, the flies were cultured in standard glass tubes.

The details of Chemical screening have been added to the Materials and Methods section.

Added: “Chemical screening was performed in 5 mL centrifuge tubes due to the limited availability of compounds.” at line 409-410.

Added: “For the secondary screen, the *Mef2-Gal4* driver, primarily expressed in mature muscle,

was employed. This driver resulted in semi-lethality, providing an easily quantifiable phenotype for further analysis. Compounds with survival counts showing a 1.5-fold increase compared to DMSO-treated flies (>30–40, approximately the top 20%) were selected.” at line 415-418.

(iii) In vivo functional fly-based assays are poorly described, and results have difficult interpretation. Interesting to add videos of the locomotor or climbing assays as Supplementary data. Examples of difficult to evaluate data are: Figure 2A: it is clear present an aging effect for the repeat stocks, more pronounced for the 16 repeat flies that during the 10 days of the assays display a strong reduction of their locomotor levels, also observed for the 720 flies. How are the authors considering it? Also, how considering that during all this time, the 720 flies should be dying... or not? Is there a semi-lethality phenotype, but once the flies born they have the same survival rate during adults? Also these locomotor peaks are so heterogeneous during the whole assay. Otherwise, the level of IFM muscle destruction in 720 repeat stock (Figure 2B and others) is so high at 15 days that these flies shouldn't fly. Why not doing a flight assay? Climbing and locomotor could be linked to other muscles involvement. Why not check what is happening in other fly muscles (histology)?

Response: The survival and locomotor activity of flies are influenced by living space and nutrition. Different experiments were conducted using varying tube sizes and food environment.

A) During chemical screening, 5 mL centrifuge tube was used due to the limited availability of compounds.

B) In the locomotor assay using the Drosophila Activity Monitor (DAM), flies were housed in glass tubes (5 mm in diameter, 6.5 cm in length) containing fly food at one end and an air-permeable closure at the other. The DAM system recorded, in real time and continuously over 24 hours, the time required for flies to cross the midpoint of the glass tube using infrared sensors.

C) For all other experiments, flies were reared in standard glass tubes with ample food supply. Different housing conditions resulted in variations in the survival and phenotypes of these flies. Experimental conclusions were drawn by comparing flies under identical conditions: the (CCTG)₇₂₀+control RI group and the (CCTG)₇₂₀+gkt RI group.

1. Videos of climbing behavior have been added in Appendix to the supplementary materials (Appendix videoS1). Figure 2A shows the continuous 24-hour locomotor of Drosophila over 10 days.

2. In the DAM system, the decreased locomotor level observed in (CCTG)₁₆ flies was attributed to narrow tube. When raised in standard glass tubes, (CCTG)₁₆ flies showed no significant changes in climbing ability on days 1, 7, or 15 (Appendix FigS2 D).

3. In the DAM system, 20 flies per genotype were tested. For (CCTG)₇₂₀ flies, 1–2 individuals generally died during the later stages of monitoring. Data from these individuals, whose movement counts remained at 0, were excluded from the analysis.

4. Flight assay has been added in Appendix Fig. S2 A-B. As shown in the flight ability test results, ~45% of (CCTG)₇₂₀ flies lost flight capability by day 15 (Appendix Fig. S2 A-B).

5. Drosophila muscles are primarily categorized into direct and indirect flight muscles (Jawkar & Nongthomba, 2020; Weitkunat & Schnorrer, 2014). The indirect flight muscles (IFMs), which are the largest and strongest, power flight and resemble vertebrate skeletal muscles in structure,

making them valuable for modeling muscle-related diseases such as Parkinson's disease and muscular dystrophy (Moehlman et al., 2023; Shcherbata et al., 2007). The direct flight muscles (DLMs), responsible for precise wing movements, are much smaller, making histological sectioning infeasible for these muscles. We examined the IFMs, which are the primary and most commonly studied flight muscles.

(iv) Figure 3 is displaying the presence of ribonuclear foci and alternative splicing (AS) issues in the DM2 720 repeat expansion stock. However, the foci observed are so high and show a strong signal. Very different from what is observed in human or mouse DM samples. What is the number of foci x cell? Could be transcription signal what it is observed? Downregulation of TDP1 is displaying reduction of the "foci" area, not the number. Explanation? Regarding AS, the gels are not compare the targeted genes with gene control expressions (actin, GAPDH).

Response: DM is primarily caused by the gain-of-function effects of toxic RNA (Udd & Krahe, 2012). Existing mouse and *Drosophila* models of DM were established through the overexpression of repeat sequences, effectively simulating the phenotypes of nerve and muscle degeneration (Braz et al., 2018; Marzullo et al., 2023).

In this FISH experiment, the strong signal was caused by the labeling of (CAGG)₅ probes with CY3 fluorophores at both ends. In this study, the muscle-specific *Mef2-Gal4* and UAS systems were employed to induce the expression of (CCTG)₇₂₀ in *Drosophila*, resulting in stable expression across nearly all muscle cells.

In addition to CAGG probes, MBNL1 protein immunofluorescence was used to confirm the authenticity of the observed foci via *UAS-hMBNL1-HA* fruit fly strains (Appendix Fig. S3A–B). MBNL1 has been shown to be co-localized with foci in DM. As shown in Appendix Fig. S3A–B, MBNL1 protein colocalized with CCUG RNA in the nucleus.

The foci area reflects the level of toxic RNA. Dr. Bonini, who developed DM2 fly models with varying CCTG repeat sizes, observed that the foci sizes in these flies were consistent with the toxicity associated with CCTG repeats (Yu et al., 2015).

The expression of *RP49* has been added as the control. As a structural constituent of the ribosome, the stable expression of *RP49* makes it particularly suitable for studies on alternative splicing in DM *Drosophila* models (Celotto & Graveley, 2002; Gentile et al., 2005). Following the methodology described by Cerro-Herreros et al., we utilized *RP49* to normalized the total RNA content (Sellier et al., 2018).

(v)The amplification of large repeat tracts by standard (PCR) methods has been displayed as highly difficult for different motifs. No publications indicating the amplification of so large CCTG repeat tract (in this case a tetranucleotide). Currently, having the possibility of long read sequencing methods successful with long repeat tracts, it should be here included to confirm, again, that the results shown are not PCR amplification issues since the lengths representation is so different depending on the stocks, some are displaying repeated peaks, and in other cases we see a straighter line (Fig. 4) Authors should explain about it. Similar concerns with the visualization of repeat bands with the presence of black areas where sizing seems very difficult, or in other cases, very regular bands displayed, shorter, but also larger (expansions?). Here one "silly" question. In the figures the 720 repeats band is located below the 2000 bp. How is this

possible? (720x4 bases=2880 bp)

Response: The (CCTG)₇₂₀ fragments have been specifically amplified. We successfully amplified CCTG repeats of various lengths (16, 200, 475, 525), and the trend of product length was consistent with the number of repeats (Appendix FigS5 A). Additionally, CCTG repeat-primed PCR was employed to validate the expansion of CCTG repeats (Appendix FigS5 B). Following amplification, the products were ligated into a vector, and Sanger sequencing confirmed the presence of CCTG repeats (Appendix FigS5 C). These results demonstrate that our PCR system amplifies the CCTG repeat expansion.

In developing these DM2 Drosophila models, Dr. Bonini verified the the length of repeat fragments using Southern blot analysis with a CCTG probe, which is consistent with our amplified products (Yu et al., 2015).

The CCTG repeats appear smaller than expected on electrophoresis due to faster migration, which is caused by the secondary structures formed by their high GC contents (Chastain et al., 1995; Gomes-Pereira & Monckton, 2017). Based on previous research and our experience with FXS testing, the electrophoretic migration of long repeat sequence was non-linear relative to the actual number of repeats (Chastain et al., 1995). As the repeat number increases, secondary structures have a greater impact on migration, leading to deviations from the expected sizes (Cheng et al., 1996).

Our lab has been working on the amplification of long repeat expansion for over 15 years, and successfully amplify repeats in Fragile X syndrome and other repeat expansion diseases, including CGG, CTG/CAG, and ATTCC repeats. Our lab has established a standard panel for Fragile X syndrome testing, including CGG repeats of various sizes, ranging from pre-mutation to full-mutation (Gao et al., 2020). We worked with long-read sequencing company in China to sequence the 400 CGG repeats. Despite extensive optimization efforts, they were unable to successfully sequence the whole region due to technical issues. To confirm the specificity of PCR amplification, we employed multiple methods, including repeat-primed PCR, Sanger sequencing and DNA chip, all of which consistently supported the accuracy of our amplification results.

•Results

(i)The manuscript points out TDP1 as a novel target for DM2. Its downregulation triggers repeat contraction and reduction of toxic phenotypes in flies. BUT, the manuscript does not include data showing the expression/protein status of the TDP1 homolog in the CCTG repeat stocks in the different approaches executed. It is not shown what happens with the expression of this gene in each CCTG repeat stock or in each developmental stage evaluated in flies, where significant contraction differences are observed. Also at the end, after the computational screening, when some TDP1 inhibitors were evaluated in flies. Thus, it is very important to have pieces of evidence indicating that TDP1 modulation is happening in the model system used, but in the manuscript, no figures or writing showing how effective was the expression or protein reduction levels when using these stocks to correlate with DM2 phenotypes readings. Also not known if differences between the different TDP1 downregulations.

Response:

1. The efficiency of *gkt* knockdown and overexpression in *Drosophila* has been verified (Appendix FigS1 A). Two *gkt* RNAi lines (knockdown efficiencies of 70–80%) and a *gkt* mutant line were tested. The *gkt* mutant line has been confirmed as a complete loss-of-function strain (Guo et al., 2014). Additionally, we constructed a *gkt* overexpressing *Drosophila* strain. Results showed that *gkt* overexpression did not significantly affect the number of repeats or locomotor activity in DM2 flies (Appendix FigS6).
2. The effect of CCTG repeat expansion on *gkt* expression has been added. According flybase gene expression Data, *TDP1/gkt* is expressed at moderate to low levels consistently throughout the entire fly life cycle and across different tissues. qPCR analysis showed no difference in the expression of *gkt* in (CCTG)₇₂₀ flies compared with (CCTG)₁₆ at the 3rd instar larvae, pupa and adult stages (Appendix FigS1 B). Western blot analysis of *gkt*-HA protein showed no change in *gkt* expression between (CCTG)₇₂₀ and (CCTG)₁₆ flies through *UAS-gkt-HA* *drosophila* (Appendix FigS1 C-D).
3. TDP1 activity assay (in vitro and in vivo) and qPCR confirmed that the tested inhibitors specifically inhibited TDP1 protein activity without affecting its expression (Fig. 5C).

(ii) TDP1 downregulation is strongly connected to repeat contraction and alleviation of DM2-related phenotypes through all the manuscript. If this true, why not test overexpression of TDP1 and check for repeat expansions and the appearance of stronger disease-based phenotypes? This would confirm the direct connection between TDP1 and repeat stability.

Response: Following your suggestion, we constructed a *gkt* overexpression *Drosophila* line by integrating *UAS-gkt-HA* into the attP40 site using the PhiC31 system. Validation was carried out through qPCR and Western blot analysis. Overexpression of *gkt* in DM2 flies did not lead to change in either repeat size or phenotype (Appendix FigS6).

(iii) Likewise, to confirm the hypothesis proposed, the manuscript needs to include also biological evidence for the target and the molecules proposed in different DM2 models (cell lines or mouse models).

Response: In addition to the DM2 *Drosophila* model, the effect of TDP1 on repeat size was also examined in the DM1 *Drosophila* model, as well as in HEK293 cell containing long repeats and DM1 patient lymphocytes (Fig EV5).

1. In the DM1 *Drosophila* model, (CTG)₂₇₀ repeat size was analyzed using capillary electrophoresis following the genetic knockdown of *gkt* (Fig EV5 A-B).
2. HEK293 cell models with stable expression of DM1 (CTG)₄₀₀ and DM2 (CCTG)₄₀₀ repeats were constructed. Repeat length was measured using an Agilent DNA chip after siRNA-mediated TDP1 knockdown combined with CPT treatment (Fig EV5 C-D).
3. Since no cases of DM2 have been reported in China, we found five DM1 patients, two of whom provided peripheral blood, from which we successfully established lymphoid cell lines, selecting the one with the longer repeat length for further experiments. In the DM1 patient-derived lymphocytes treated with TDP1 inhibitors, repeat length was also measured using an Agilent DNA chip (Fig EV5 E-F).

Across these models, inhibiting or knocking down TDP1 consistently promoted repeat contraction.

(iv) The mechanism proposed related to the downregulation of TDP1 is not presented as CCTG-specific. Other repeat motifs must be evaluated for TDP1 downregulation to support results presented in DM2 flies.

Response: Same as comment (iii) above; see response above.

(v) Comparing with DM1, DM2 is characterized by less severe human phenotypes. However, DM2 flies display robust toxic phenotypes, even higher than DM1 flies. New locomotor, climbing, and sarcomere histology phenotypes are here described by first time for the DM2 flies pointing out stronger toxic phenotypes. Good characterization and controls are needed. For example, no controls indicate that muscle Gal4 itself is introducing some toxicity degree in the presence of the large repeat. Together, these DM2 flies seems not to be a good model regarding the whole DM2 pathology, although not discarding its valid use for targeting repeat toxicity issues.

Response: Three distinct DM1 *Drosophila* models, expressing 162, 270, and 480 CTG repeats, exhibit muscle atrophy phenotypes when crossed with *MHC-Gal4* (de Haro et al., 2006; Houseley et al., 2005; Yu et al., 2011). As *MHC-Gal4>UAS-(CCTG)₇₂₀* flies did not display notable muscle phenotypes, various muscle-specific GAL4 drivers were tested to identify the most suitable phenotypes for screening and other experiments.

Mef2-Gal4 was selected during protocol development for its clear muscle phenotype. The *Mef2-Gal4* driver, primarily expressed in mature muscle, was commonly used in muscle studies. The controls, including *Mef2-Gal4>Control RI*, *Mef2-Gal4>gkt RI*, and *Mef2-Gal4>(CCTG)₁₆ + Control RI*, showed normal performance in muscle-related phenotypes, such as locomotion, paraffin sections, and immunostaining of muscle fiber morphology (Fig 2A, D, Fig 3A, C, E-F, Fig EV1 A, C, H-I, Fig EV2 A, C, E-F).

DM is primarily caused by the gain-of-function effects of toxic RNA. Existing mouse and *Drosophila* models of DM were established through the overexpression of repeat traits, effectively simulating the phenotypes of neural and muscle degeneration. No DM2 mouse model has been reported to date, making *Drosophila* the only available model for this disease (Marzullo et al., 2023). Currently, three DM2 *Drosophila* models have been developed, effectively simulating patient phenotypes in the nervous system, muscles, and heart (Cerro-Herreros et al., 2017; Yenigun et al., 2017; Yu et al., 2015). *Drosophila* remains the only model suitable for large-scale screening.

(vi) The project rationale lacks of detail in some key steps. Starts with a small molecule screening, with some hits identified after two rounds of screening. But here, the authors do not move forward, as is common for this type of screenings, validating some hits in further steps. Can authors explain why? Here, it is proposed a new selection step based in choosing compounds with notable bioactivity levels. What authors mean with "notable bioactivity levels"? What was the specific rationale/threshold to choose the 16 from the 31? And finally, authors check for biological target coincidence, moving to finally define TDP1 as the gene target, common for five of the hits defined as all inhibitors, to be validated. The manuscript has to include references or links that must show the connection for the five compounds and their TDP1 inhibitor role. Also, at this point, the

authors explain that ESR1 target is discarded because some hits inhibitors but other activators. Fig 1C shows that some other genes that showed up, including ESR1, share hits with TRP1, suggesting a pathway involved, not only a gene. But explanation for this is lacking into the rationale.

Response: Details of chemical screening have been added to the Material Methods section.

Added: “In the primary screen, 140 compounds were tested using the *24B-Gal4* driver, which induces broad expression of *(CCTG)720* from the mesoderm to muscle cells. This expression caused complete lethality in flies, allowing for the identification of compounds that could potentially rescue toxicity by influencing muscle development.” in line 410-414.

“In total, 31 compounds were identified in the secondary screen that exhibited varying degrees of rescue effects against muscle toxicity. Among these, 16 compounds with target information available in the PubChem Compound database were selected for subsequent analysis.” in line 419-421.

In addition to TDP1, other hits, such as ESR1, RYR1, and NF-KB, are being verified. As these findings are beyond the scope of the current study, the corresponding data have not been included.

The “notable bioactivity” was replaced to “target information” at line 154. Among 31 compounds, 16 compounds with target information available in the PubChem Compound database were selected for subsequent analysis.

In the bioassay, the effects of these compounds on TDP1 were primarily assessed based on cell viability. These compounds may influence TDP1 activity either directly or indirectly.

(vii) The images for the IFM muscle packages from the 720 stock displays a very "destroyed" muscle. This is not seen from the DM1 models, where atrophy is evident but as of loss of muscle area. Would be great to know what is contained in the "holes" between the muscles that still remains. Additional staining to see if the holes contain connective tissue (Masson or other histology labelling) should be performed to discard technical issues (poor fixation, etc) during the protocol. Linked to the last, some fusion of muscle packages is observed indicating technical issues (less than 6 packages observed in Fig 2D and Fig EV1D indicating bad fixation process).

Response: Masson staining, which is used to stain connective tissue in mammals, was added to Appendix Fig. S2C. The muscles in *(CCTG)₁₆* flies remained intact, while the unstained holes observed in *(CCTG)₇₂₀* flies suggest tissue damage. Previous studies have shown that *Drosophila* connective tissue does not take up histological stains (Gregor et al., 2022).

In Fig. 2D and Fig. EV1D, the spacing between muscle packages is less distinct, likely due to damage in some muscle fibers. In contrast, the six muscle packages in control flies are clearly visible.

(viii) The project finish with a virtual screening looking for molecules binding TDP1 and acting as inhibitors. Some were found and tested. But, why not using the ones identified in the first screening, also pointed out as TDP1 inhibitors? Did they show up in the virtual screening? They should, at least their MoA different but also triggering TDP1 downregulation.

Response: In the bioassay of NCBI PubChem Compound database, the effects of these compounds on TDP1 were primarily assessed based on cell viability. These compounds may influence TDP1 activity either directly or indirectly. Given that concentrations as high as 40 μ M were used during screening, these compounds cannot be classified as specific TDP1 inhibitors.

Other concerns

(i) What does it mean? Line 158-160 "...restored locomotor activity and climbing ability without affecting normal locomotor activity"

Response: Changed to: "Knockdown of *gkt*, through two RNAi lines and one mutant line, restored locomotor activity and climbing ability in DM2 flies, while *gkt* knockdown alone did not produce any noticeable phenotype in control flies." in line 169-171.

(ii) The manuscript is presenting consistent results indicating the identification of a novel biological target, some molecules able to modulate it in a very promising direction, triggering repeat contractions. This data is the type of data that patients are waiting for, with clear translatability to drugs developments. However, the manuscript not indicate if these assets are protected (patent?). This should be done previously to publication to have chances of reaching patients successfully.

Response: There is currently no patent for TDP1 as a repeat expansion disease target. We have already filed the relevant patent application. The effect of TDP1 inhibition on repeat expansion requires substantial time and effort for validation in other models, such as iPSC-derived muscle tissue from patients, higher animal models, and organoid models, in the future.

Braz, S. O., Acquaire, J., Gourdon, G., & Gomes-Pereira, M. (2018). Of Mice and Men: Advances in the Understanding of Neuromuscular Aspects of Myotonic Dystrophy. *Front Neurol*, 9, 519. <https://doi.org/10.3389/fneur.2018.00519>

Celotto, A. M., & Graveley, B. R. (2002). Exon-specific RNAi: a tool for dissecting the functional relevance of alternative splicing. *RNA*, 8(6), 718-724. <https://doi.org/10.1017/s1355838202021064>

Cerro-Herreros, E., Chakraborty, M., Perez-Alonso, M., Artero, R., & Llamusi, B. (2017). Expanded CCUG repeat RNA expression in *Drosophila* heart and muscle trigger Myotonic Dystrophy type 1-like phenotypes and activate autophagocytosis genes. *Sci Rep*, 7(1), 2843. <https://doi.org/10.1038/s41598-017-02829-3>

Chastain, P. D., 2nd, Eichler, E. E., Kang, S., Nelson, D. L., Levene, S. D., & Sinden, R. R. (1995). Anomalous rapid electrophoretic mobility of DNA containing triplet repeats associated with human disease genes. *Biochemistry*, 34(49), 16125-16131. <https://doi.org/10.1021/bi00049a027>

Cheng, S., Barcelo, J. M., & Korneluk, R. G. (1996). Characterization of large CTG repeat expansions in myotonic dystrophy alleles using PCR. *Hum Mutat*, 7(4), 304-310. [https://doi.org/10.1002/\(SICI\)1098-1004\(1996\)7:4<304::AID-HUMU3>3.0.CO;2-8](https://doi.org/10.1002/(SICI)1098-1004(1996)7:4<304::AID-HUMU3>3.0.CO;2-8)

de Haro, M., Al-Ramahi, I., De Gouyon, B., Ukani, L., Rosa, A., Faustino, N. A., Ashizawa, T.,

- Cooper, T. A., & Botas, J. (2006). MBNL1 and CUGBP1 modify expanded CUG-induced toxicity in a *Drosophila* model of myotonic dystrophy type 1. *Hum Mol Genet*, *15*(13), 2138-2145. <https://doi.org/10.1093/hmg/ddl137>
- Gao, F., Huang, W., You, Y., Huang, J., Zhao, J., Xue, J., Kang, H., Zhu, Y., Hu, Z., Allen, E. G., Jin, P., Xia, K., & Duan, R. (2020). Development of Chinese genetic reference panel for Fragile X Syndrome and its application to the screen of 10,000 Chinese pregnant women and women planning pregnancy. *Mol Genet Genomic Med*, *8*(6), e1236. <https://doi.org/10.1002/mgg3.1236>
- Gentile, C., Lima, J. B., & Peixoto, A. A. (2005). Isolation of a fragment homologous to the rp49 constitutive gene of *Drosophila* in the Neotropical malaria vector *Anopheles aquasalis* (Diptera: Culicidae). *Mem Inst Oswaldo Cruz*, *100*(6), 545-547. <https://doi.org/10.1590/s0074-02762005000600008>
- Gomes-Pereira, M., & Monckton, D. G. (2017). Ethidium Bromide Modifies The Agarose Electrophoretic Mobility of CAG*CTG Alternative DNA Structures Generated by PCR. *Front Cell Neurosci*, *11*, 153. <https://doi.org/10.3389/fncel.2017.00153>
- Gregor, K. M., Becker, S. C., Hellhammer, F., Schon, K., Baumgartner, W., & Puff, C. (2022). Histochemical staining techniques in *Culex pipiens* and *Drosophila melanogaster* (Diptera) with a comparison to mammals. *Vet Pathol*, *59*(5), 836-849. <https://doi.org/10.1177/03009858221088786>
- Guo, D., Dexheimer, T. S., Pommier, Y., & Nash, H. A. (2014). Neuroprotection and repair of 3'-blocking DNA ends by glaikit (gkt) encoding *Drosophila* tyrosyl-DNA phosphodiesterase 1 (TDP1). *Proc Natl Acad Sci U S A*, *111*(44), 15816-15820. <https://doi.org/10.1073/pnas.1415011111>
- Houseley, J. M., Wang, Z., Brock, G. J., Soloway, J., Artero, R., Perez-Alonso, M., O'Dell, K. M., & Monckton, D. G. (2005). Myotonic dystrophy associated expanded CUG repeat muscleblind positive ribonuclear foci are not toxic to *Drosophila*. *Hum Mol Genet*, *14*(6), 873-883. <https://doi.org/10.1093/hmg/ddi080>
- Jawkar, S., & Nongthomba, U. (2020). Indirect flight muscles in *Drosophila melanogaster* as a tractable model to study muscle development and disease. *Int J Dev Biol*, *64*(1-2-3), 167-173. <https://doi.org/10.1387/ijdb.190333un>
- Marzullo, M., Coni, S., De Simone, A., Canettieri, G., & Ciapponi, L. (2023). Modeling Myotonic Dystrophy Type 2 Using *Drosophila melanogaster*. *Int J Mol Sci*, *24*(18). <https://doi.org/10.3390/ijms241814182>
- Moehlman, A. T., Kanfer, G., & Youle, R. J. (2023). Loss of STING in parkin mutant flies suppresses muscle defects and mitochondria damage. *PLoS Genet*, *19*(7), e1010828. <https://doi.org/10.1371/journal.pgen.1010828>
- Sellier, C., Cerro-Herreros, E., Blatter, M., Freyermuth, F., Gaucherot, A., Ruffenach, F., Sarkar, P., Puymirat, J., Udd, B., Day, J. W., Meola, G., Bassez, G., Fujimura, H., Takahashi, M. P., Schoser, B., Furling, D., Artero, R., Allain, F. H. T., Llamusi, B., & Charlet-Berguerand, N. (2018). rbFOX1/MBNL1 competition for CCUG RNA repeats binding contributes to myotonic dystrophy type 1/type 2 differences. *Nat Commun*, *9*(1), 2009. <https://doi.org/10.1038/s41467-018-04370-x>
- Shcherbata, H. R., Yatsenko, A. S., Patterson, L., Sood, V. D., Nudel, U., Yaffe, D., Baker, D., & Ruohola-Baker, H. (2007). Dissecting muscle and neuronal disorders in a *Drosophila*

- model of muscular dystrophy. *EMBO J*, 26(2), 481-493.
<https://doi.org/10.1038/sj.emboj.7601503>
- Udd, B., & Krahe, R. (2012). The myotonic dystrophies: molecular, clinical, and therapeutic challenges. *Lancet Neurol*, 11(10), 891-905. [https://doi.org/10.1016/S1474-4422\(12\)70204-1](https://doi.org/10.1016/S1474-4422(12)70204-1)
- Weitkunat, M., & Schnorrer, F. (2014). A guide to study *Drosophila* muscle biology. *Methods*, 68(1), 2-14. <https://doi.org/10.1016/j.ymeth.2014.02.037>
- Yenigun, V. B., Sirito, M., Amcheslavsky, A., Czernuszewicz, T., Colonques-Bellmunt, J., Garcia-Alcover, I., Wojciechowska, M., Bolduc, C., Chen, Z., Lopez Castel, A., Krahe, R., & Bergmann, A. (2017). (CCUG)(n) RNA toxicity in a *Drosophila* model of myotonic dystrophy type 2 (DM2) activates apoptosis. *Dis Model Mech*, 10(8), 993-1003. <https://doi.org/10.1242/dmm.026179>
- Yu, Z., Goodman, L. D., Shieh, S. Y., Min, M., Teng, X., Zhu, Y., & Bonini, N. M. (2015). A fly model for the CCUG-repeat expansion of myotonic dystrophy type 2 reveals a novel interaction with MBNL1. *Hum Mol Genet*, 24(4), 954-962. <https://doi.org/10.1093/hmg/ddu507>
- Yu, Z., Teng, X., & Bonini, N. M. (2011). Triplet repeat-derived siRNAs enhance RNA-mediated toxicity in a *Drosophila* model for myotonic dystrophy. *PLoS Genet*, 7(3), e1001340. <https://doi.org/10.1371/journal.pgen.1001340>

Referee #3 (Remarks for Author):

1) The technical quality and statistical analysis is good with sufficient numbers and repetitions. For the locomotor graphs, it would help to know what the counts are about (I am assuming it is the number of times the alarm was set off by a fly breaking the beam). This could be made clearer in the figure legend.

Response: Details of locomotor assay has been added to the Materials and Methods section.

Added: “The *Drosophila* Activity Monitor (DAM) utilizes centrally positioned 5 mm activity tubes, which are equipped with fly food at one end and an air-permeable closure at the other. This setup houses individual flies for experiments lasting 15–25 days. The activity tubes are fitted with infrared monitoring system that records whenever a fly crosses the midpoint of the tube. ” in line 428-432.

2) The identification of TDP1 through genetic analysis as well as compounds inhibiting its activity and their subsequent effects on multiple phenotypes was novel, convincing and impactful.

3) The medical impact is too early to comment on, though these experiments provide avenues for further investigations in mouse models, for example.

What is lacking and I think is essential, is to show:

1) what happened to the expression of the toxic RNA (transgene RNA)

Response: The data on toxic RNA has been added in Appendix Fig. S3D. The levels of toxic RNA were assessed by qPCR targeting the flanking sequences of CCUG repeats. As shown in Appendix Fig. S3D, knocking down *gkt* does not alter the levels of toxic CCUG RNA in DM2 flies.

Appendix Figure S3. (D) Relative mRNA level of *CCUG* across different genotypes at 15-days-old. Data are mean \pm SD. Two-tailed, unpaired t-test. *** $P = 0.0002$.

2) what happened to Mbl (drosophila MBNL) expression ?

Response: The experiments involving *mbl*/MBNL1 have been added in Appendix Fig. S3A–C. QPCR analysis showed that *gkt* knockdown does not affect the expression of *mbl* (Appendix Fig. S3C). The *UAS-hMBNL1-HA* Drosophila was used to assess the sequestration of MBNL1/*mbl* proteins by toxic RNA. As shown in Fig. S3A–B, MBNL1 protein colocalized with CCUG RNA in the nucleus, and *gkt* knockdown reduces this sequestration.

Appendix Figure S3. *TDPI/gkt* knockdown corrected the abnormal localization of MBNL1/*mbi*. (A) Representative image of colocalized nuclear foci FISH and MBNL1 immunofluorescence. Scale bars 5 μ m. (B) Quantification at least ten animals of each genotype. Data are mean \pm SD. Two-tailed, unpaired t-test. *** $P = 0.0005$, ** $P = 0.005$. (C) Relative mRNA level of *mbi* across different genotypes at 15-days-old. Data are mean \pm SD. Two-tailed, unpaired t-test.

3) was the stability of the toxic RNA affected?

Response: In this study, TDP1 knockdown reduced CCTG repeats at the DNA level, resulting in shorter CCTG repeat tracts, which in turn, led to smaller foci area and corresponding decrease in RNA toxicity. QPCR results show that *TDPI/gkt* knockdown did not affect CCUG RNA levels in either $(CCTG)_{16}$ or $(CCTG)_{720}$ (Appendix Fig. S3D).

The stability of toxic RNA is primarily influenced by the number of repeats, with longer repeats conferring greater stability (Reddy et al., 2013). Nuclease cleavage and thermal melting experiments revealed that longer repeats facilitate the formation of secondary structures, such as hairpins, which enhance stability (Błaszczuk et al., 2017; Dickson & Wilusz, 2010; Napierala & Krzyzosiak, 1997; Reddy et al., 2013). Dr. Bonini, who developed the DM2 Drosophila model, demonstrated through fluorescence in-situ hybridization that longer CCTG repeats promote increased toxic RNA aggregation (Yu et al., 2015). The above indicates that, while TDP1 knockdown reduces CCTG repeat size and RNA toxicity, the stability of the toxic RNA remains unaffected by TDP1 modulation.

4) was the stability of the transgene construct affected. That is, is it just the repeat length was affected or was the transgene degraded in other ways.

Response: The transgenic structure is stable.

- 1、 In constructing the DM2 Drosophila model, Dr. Bonini demonstrated through Southern blot that the transgenic band with varying CCTG repeats in Drosophila are single and stable (Yu et al., 2015).
- 2、 The amount of transgenes in *CCTG720*, *(CCTG)720 + gkt RI*, and *(CCTG)720 + gkt OE* flies was measured by qPCR using CCTG flanking sequences (Appendix Fig. S3E). No significant differences were observed. We constructed a *gkt* overexpression Drosophila line. Overexpression of *gkt* in DM2 flies did not lead to change in either repeat size or phenotype (Appendix Fig. S6).
- 3、 The toxic RNA expression was evaluated using CCUG flanking sequences. QPCR analysis of toxic RNA in *CCTG720* and *(CCTG)720+gkt RI* showed that *gkt* knockdown did not affect toxic RNA expression (Appendix Fig. S3D).
- 4、 In addition to the transgenic model, we also constructed a DM1 patient-derived lymphoid cell line. In this model, TDP1 inhibitors promoted repeat contraction, indicating that endogenous CTG repeats also contract upon TDP1 inhibition (Fig EV5 E-F).

Appendix S1. (E) Relative DNA level of CCTG transgene across different genotypes at 15-days-old from five biological replicates. Data are mean \pm SD. Two-tailed, unpaired t-test.

Appendix Figure S6. TDP1/gkt overexpression did not affect CCTG repeat size. (A) Relative mRNA level of *gkt* across different genotypes at 7-days-old. Data are mean \pm SD. Two-tailed, unpaired t-test. $***P = 0.0006$. (B) Immunoblot against HA across different genotypes at 7-days-old. (C) Representative gel image of PCR across *Mef2-Gal4>UAS-(CCTG)₇₂₀+atp40* and *Mef2-Gal4>UAS-(CCTG)₇₂₀+UAS-gkt-HA* at 15-days-old and 30-days-old. Tissue from three animals per sample. (D) Quantification of mean length of CCTG repeats from three biological replicates.

Data are mean \pm SD. Two-tailed, unpaired t-test. (E) The average counts of per 30min in flies across different genotypes from 15-days-old to the 25-days-old. Fifteen animals per sample. (F) Quantification of the average counts per day in flies across different genotypes from 15-days-old to the 25-days-old. Five biological replicates. Data are mean \pm SD. Two-tailed, unpaired t-test.

Figure EV5 (E) Representative gel image of DM1 patient-derived lymphocytes following 15 days treatment of different compounds from bioanalyzer. (F) Quantification of (E) from three biological replicates. Data are mean \pm SD. One-way ANOVA. **P = 0.0082 DMSO versus Lawsonsone, ***P = 0.001 DMSO versus SPI-112.

Błaszczuk, L., Rypniewski, W., & Kiliszek, A. (2017). Structures of RNA repeats associated with neurological diseases. *Wiley Interdiscip Rev RNA*, 8(4). <https://doi.org/10.1002/wrna.1412>

Dickson, A. M., & Wilusz, C. J. (2010). Repeat expansion diseases: when a good RNA turns bad. *Wiley Interdiscip Rev RNA*, 1(1), 173-192. <https://doi.org/10.1002/wrna.18>

Napierala, M., & Krzyzosiak, W. J. (1997). CUG repeats present in myotonin kinase RNA form metastable "slippery" hairpins. *J Biol Chem*, 272(49), 31079-31085. <https://doi.org/10.1074/jbc.272.49.31079>

Reddy, K., Zamiri, B., Stanley, S. Y. R., Macgregor, R. B., Jr., & Pearson, C. E. (2013). The disease-associated r(GGGGCC)_n repeat from the C9orf72 gene forms tract length-dependent uni- and multimolecular RNA G-quadruplex structures. *J Biol Chem*, 288(14), 9860-9866. <https://doi.org/10.1074/jbc.C113.452532>

Yu, Z., Goodman, L. D., Shieh, S. Y., Min, M., Teng, X., Zhu, Y., & Bonini, N. M. (2015). A fly model for the CCUG-repeat expansion of myotonic dystrophy type 2 reveals a novel interaction with MBNL1. *Hum Mol Genet*, 24(4), 954-962. <https://doi.org/10.1093/hmg/ddu507>

27th Jan 2025

Dear Prof. Duan,

Thank you for the submission of your revised manuscript to EMBO Molecular Medicine. I am pleased to inform you that we will be able to accept your manuscript pending the following final amendments:

1) Please address all the points raised by the referees. No additional experiments are required. Limitations of the study should be clearly stated taking referee #2 and #3 points in consideration. Please implement referee #1 suggestions. Acceptance of the manuscript will depend on the completeness of your responses included in the next, final version of the manuscript. For this reason, and to save you from any frustrations in the end, I would strongly advise against returning an incomplete revision.

2) Figures:

- Please upload individual, high-resolution files in TIFF, EPS or PDF format for each main and EV figures. All graphs of a figure should fit on one page and all figures should be uploaded as one file per figure. Please check "Author Guidelines" for more information:

<https://www.embopress.org/page/journal/17574684/authorguide#figureformat>

<https://www.embopress.org/page/journal/17574684/authorguide#expandedview>

- We note that the H&E staining images in Figure 2D are reused in Figure EV1D. Please indicate this clearly in the figure legends and make sure that the images displayed in different figures are indeed from the same experiment.

3) Author checklist: Please submit a complete checklist. <https://www.embopress.org/pb-assets/embo-site/EMBO%20Press%20Author%20Checklist-1642513524327.xlsx>

4) In the main manuscript file, please do the following:

- Please address all comments suggested by our data editors listed below:

o Figure legends:

1. Please note that, sub-figures panels 6J-M are not provided in sequence in the merged manuscript. The above sub-figures are provided after figure 5. Please provide sub-figures 6J-M in sequence.

2. Please note that, figure panels for EV5 are provided before EV1 in the merged manuscript file. Please provide all the figure panels in correct sequence.

3. Please note that the exact p values are not provided in the legends of figures 2C, E, G; 3B, D; 4G, 6G, K; EV1 C, G, I; EV2 B, EV4 F, H4.

4. Please note that the scale bar is missing for figures EV1 J

5. Please note that the white arrows are not defined in the legend of figure 6H, EV2 A. This needs to be rectified.

6. Please note that the black arrows are not defined in the legend of figure EV1 F. This needs to be rectified.

- Add callouts for Figure 4B and 6H-I. Please also update the callouts for Suppl. Table 1 and 2, and for FigS1 E to Table EV1 and EV2 and Appendix Figure S1E.

- Rename "Conflict of Interest" to "Disclosure and competing interests statement". We updated our journal's competing interests policy in January 2022 and request authors to consider both actual and perceived competing interests. Please review the policy <https://www.embopress.org/competing-interests> and update your competing interests if necessary.

- Author contributions: Please remove it from the manuscript and specify author contributions in our submission system. CRediT has replaced the traditional author contributions section because it offers a systematic machine-readable author contributions format that allows for more effective research assessment. You are encouraged to use the free text boxes beneath each contributing author's name to add specific details on the author's contribution. More information is available in our guide to authors:

<https://www.embopress.org/page/journal/17574684/authorguide#authorshipguidelines>

- Please include structured Methods section that includes a Reagents and Tools Table (should be uploaded as a separate file) followed by a Methods and Protocols section. More information on how to adhere to this format as well as downloadable templates (.docx) for the Reagents and Tools Table can be found in our author guidelines:

<https://www.embopress.org/page/journal/17574684/authorguide#structuredmethods>

An example of a paper with Structured Methods can be found here:

<https://www.embopress.org/doi/full/10.1038/s44320-024-00037-6#sec-4>

- Indicate in legends number and nature of replicates and exact p= values, not a range, along with the statistical test used. To keep the figures "clear" some authors found providing an Appendix table Sx with all exact p-values preferable. You are welcome to do this if you want to.

- Please remove "Expanded View for this article is available online".

- Correct the reference citation in the reference list. Please remove DOI and where there are more than 10 authors on a paper, 10 will be listed, followed by "et al.". Please check "Author Guidelines" for more information.

<https://www.embopress.org/page/journal/17574684/authorguide#referencesformat>

5) Tables: Please add legends for Table EV1 and EV2.

6) Movies: Please rename Appendix Video S1 to Movie EV1 and zip a legend with a short description with the file.

7) Appendix: Please add page numbers in the table of content and upload the file as a PDF.

8) The Paper Explained: Please provide "The Paper Explained" and add it to the main manuscript text. Please check "Author Guidelines" for more information. <https://www.embopress.org/page/journal/17574684/authorguide#researcharticleguide>

9) Synopsis: Every published paper now includes a 'Synopsis' to further enhance discoverability. Synopses are displayed on the journal webpage and are freely accessible to all readers. They include separate synopsis image and synopsis text.

- Synopsis image: Please provide a striking image or visual abstract as a high-resolution jpeg file 550 px-wide x (250-400)-px high to illustrate your article.

- Synopsis text: Please provide a short standfirst (maximum of 300 characters, including space) as well as 2-5 one sentence bullet points that summarise the paper as a .doc file. Please write the bullet points to summarise the key NEW findings. They should be designed to be complementary to the abstract - i.e. not repeat the same text. We encourage inclusion of key acronyms and quantitative information (maximum of 30 words / bullet point). Please use the passive voice.

10) As part of the EMBO Publications transparent editorial process initiative (see our Editorial at <http://embomolmed.embopress.org/content/2/9/329>), EMBO Molecular Medicine will publish online a Review Process File (RPF) to accompany accepted manuscripts. This file will be published in conjunction with your paper and will include the anonymous referee reports, your point-by-point response and all pertinent correspondence relating to the manuscript. Let us know whether you agree with the publication of the RPF and as here, if you want to remove or not any figures from it prior to publication. Please note that the Authors checklist will be published at the end of the RPF.

11) Please provide a point-by-point letter INCLUDING my comments as well as the reviewer's reports and your detailed responses (as Word file).

I look forward to reading a new revised version of your manuscript as soon as possible.

Yours sincerely,

Zeljko Durdevic

*** Instructions to submit your revised manuscript ***

1) a .docx formatted version of the manuscript text (including Figure legends and tables)

2) Separate figure files*

3) supplemental information as Expanded View and/or Appendix. Please carefully check the authors guidelines for formatting Expanded view and Appendix figures and tables at

<https://www.embopress.org/page/journal/17574684/authorguide#expandedview>

4) a letter INCLUDING the reviewer's reports and your detailed responses to their comments (as Word file).

5) The paper explained: EMBO Molecular Medicine articles are accompanied by a summary of the articles to emphasize the major findings in the paper and their medical implications for the non-specialist reader. Please provide a draft summary of your

article highlighting

6) Author contributions: the contribution of every author must be detailed in a separate section.

7) EMBO Molecular Medicine now requires a complete author checklist

(<https://www.embopress.org/page/journal/17574684/authorguide>) to be submitted with all revised manuscripts. Please use the checklist as guideline for the sort of information we need WITHIN the manuscript. The checklist should only be filled with page numbers where the information can be found. This is particularly important for animal reporting, antibody dilutions (missing) and exact values and n that should be indicated instead of a range.

8) Every published paper now includes a 'Synopsis' to further enhance discoverability. Synopses are displayed on the journal webpage and are freely accessible to all readers. They include a short stand first (maximum of 300 characters, including space) as well as 2-5 one sentence bullet points that summarise the paper. Please write the bullet points to summarise the key NEW findings. They should be designed to be complementary to the abstract - i.e. not repeat the same text. We encourage inclusion of key acronyms and quantitative information (maximum of 30 words / bullet point). Please use the passive voice. Please attach these in a separate file or send them by email, we will incorporate them accordingly.

You are also welcome to suggest a striking image or visual abstract to illustrate your article. If you do please provide a jpeg file 550 px-wide x 300-600px high.

9) A Conflict of Interest statement should be provided in the main text

10) Please note that we now mandate that all corresponding authors list an ORCID digital identifier. This takes <90 seconds to complete. We encourage all authors to supply an ORCID identifier, which will be linked to their name for unambiguous name identification.

Currently, our records indicate that the ORCID for your account is 0000-0001-7117-4487.

Link Not Available

11) Include a Reagents and Tools Table as part of the Methods section, which can be downloaded from our author guidelines (<https://www.embopress.org/page/journal/17574684/authorguide#structuredmethods>)

Photos 400-800 DPI

*Additional important information regarding figures and illustrations can be found at

<https://bit.ly/EMBOPressFigurePreparationGuideline>. See also figure legend preparation guidelines:

<https://www.embopress.org/page/journal/17574684/authorguide#figureformat>

***** Reviewer's comments *****

Referee #1 (Remarks for Author):

The authors adequately addressed my comments, confirming their data in a DM1 fly model and non-physiological HEK or lymphoblastic cells expressing the DM mutations, but which have the merit to be human based models. Thus, these results are now suitable for publication.

PS, the text has been slightly improved, but remains largely opaque and uneasy to follow (there is still zero detail about

compounds and their identities in figure 1, and the new models, including DM1 flies and mammalian cells are expedited in 10 short lines at the very end of the results part). However, these are mostly editorial questions.

Referee #2 (Comments on Novelty/Model System for Author):

The manuscript describes identifying novel compounds in terms of mechanisms of action for "positive" modulation of the type of mutation defined for repeat expansion diseases. The work is focused on a *Drosophila* DM2 model. Although the results are auspicious as the manuscript is now presented, additional data is needed to finally achieve journal standards, indicating "demonstration of human disease relevance", here still not a reached. In addition, the limitation to DM2, barely extended to DM1, and supported by the exciting first mechanistic results, it has also be implemented with new disease inclusion and new models, even for DM2, currently existing.

Not suited to a short report. This work needs to "grow" a provide solider data in more diseases and models.

Referee #2 (Remarks for Author):

The authors have made a good effort to implement the reviewer request, but the final result is only partial responses, more effective for the writing structure than for the results interpretation and new data included. Issues still not addressed:

1. the stated "repeat stability" of TRE fly models is clear in the literature. The explanation indicating where the contractions were observed during the fly development, and how they could be produced is poor and confusing at this moment (lines 228-254). Only introducing more questions, and still not responding why now instability (here contractions) can easily being observed when this not happened before.
2. the fly experiments (crossing) performed are complex. Additional information provided, only explaining that *Mef2*>CCTG flies were obtained, is still incomplete, since also siRNA and overexpression crosses were done. A figure with all the crosses and at what stage the contraction observed is highly recommended for not-in-fly experts.
3. Authors included results with human DM1 cell lines from patients, and Hek293 transfected with CTG/CTGG repeats, but limited to see contraction. Results detecting correction of phenotypes in patient-derived DM1/DM2 cells lines (foci, mbn1 sequestration, miR23b expression, splicing, etc) is needed to assure the translatability of the compounds here indicated. Also, experiments performed with human cells, some done with one compound (not the good ones), and others with the two compounds chosen. Why? They should be done all with the compounds chosen.
4. Although the authors focused on DM2, the results obtained suggest to move, at least, to DMs for the manuscript. As it is now, it is not giving definitive responses of the final applicability.
5. Protection (patent) of the data here described it is not well resolved by the authors. Would be a pity these compounds are not well-protected if their promising activity is finally also observed in human situations.
6. Still not clear to this reviewer why they did not evaluate some of the compounds from the *Drosophila* screening. The authors mentioned the high concentration used in the fly screening...but this initially does not discard a good response in cells...
7. Still some repeat lengths and/or results in figures not well understood. For example, figure EV5-E displays a "normal" repeat band close to 600 bp, or same figure EV5 in C-D, it is stated that a contraction occurring in DM2 situation, but it is very hard to see. In the same EV5-E figure, the expansion is now stated above the 2000 bp ladder band, and not very clear. Before stated around 1600 bp.

Referee #3 (Comments on Novelty/Model System for Author):

It is unclear what the medical impact of this will be. It is probably correct that slowing down expansion rates could slow down disease progression in repeat mediated disorders. However, most of the strategies, including this one, involve affecting DNA replication and error correction, which is a slippery but known slope towards cancer.

Referee #3 (Remarks for Author):

The authors have done a good job of addressin concerns from the three reviewers and the paper is better for it.

However, one thought lingered and on thing occurred to me upon rereading this manuscript. The authors conclude that the TDP1 inhibitors are mediating their effects through impaired repairing repair mechanisms, leading to contractions of the repeat tract. However, this wasn't directly proven. It is correlative.

The question I had was what happened to the various phenotypes (histologic and molecular (splicing, Foci size, etc.) when you inhibited TDP1 and TOP1. You showed that the repeat didn't shrink, but were silent about the phenotypic effects. Did you observe no rescue of phenotype or did you observe a rescue of phenotype despite no change in repeat size. If the latter, then it suggests that inhibition of TDP1 is having its beneficial effects through other mechanisms. Please address.

Referee #1 (Remarks for Author):

The authors adequately addressed my comments, confirming their data in a DM1 fly model and non-physiological HEK or lymphoblastic cells expressing the DM mutations, but which have the merit to be human based models. Thus, these results are now suitable for publication.

PS, the text has been slightly improved, but remains largely opaque and uneasy to follow (there is still zero detail about compounds and their identities in figure 1, and the new models, including DM1 flies and mammalian cells are expedited in 10 short lines at the very end of the results part). However, these are mostly editorial questions.

Response:

Regarding “Compound Identities in Figure 1”, the description has been updated in the Results and Materials Methods sections respectively.

“To find small molecules capable of mitigating muscle toxicity, we conducted a chemical screen of 2,160 compounds from the Johns Hopkins Clinical Compound Library (JHCCL) of FDA approved or approvable drugs. Given that muscle degeneration is the prominent phenotype in DM2 patients, muscular toxicity was used as a basis for our screen. The expression of 720 CCTG repeats across all somatic muscles, under control of the muscle-expressed *24B-Gal4* driver, led to pupal lethality. A total of 140 compounds were identified to rescue the viability of *24B-Gal4 > UAS-(CCTG)₇₂₀*, resulting in the emergence of adults (Fig 1A). To verify the efficacy of these compounds without their effects on muscle development, another muscle-specific *Mef2-Gal4* driver was utilized. The expression of 720 CCTG repeats across late-stage muscle development and mature muscle, under the control of the *Mef2-Gal4* driver, resulted in semi-lethality. The 31 screened compounds showed a

significant increase (≥ 1.5 -fold change compared to DMSO) in survival rates (Fig 1B). Further analysis, referencing the PubChem Compound database, narrowed these to 16 with target information (Fig 1C, Supplementary Table S1). While ESR1 was pinpointed by six compounds, it was excluded due to the inconsistent activity types among these compounds, with three acting as agonists and the others as inhibitors. The inhibition of TDP1 by five compounds, at the top of the screening list, exhibited the greatest potential for modifying muscle toxicity (Table 1).” at the line 147-163 of Result.

“The compounds utilized in this study are as follows: 2,160 compounds from the JHCCL Lawsone (MCE), SPI-112 (MCE), Camptothecin (CPT, TargetMol), Amikacin (TargetMol.), Dimethyl sulfoxide (DMSO, TargetMol).

The compounds from the JHCCL were used for chemical screen at a dose of 40 μ M. Chemical screening was performed in 5 mL centrifuge tubes due to the limited availability of compounds. In the primary screen, 140 compounds were tested using the 24B-Gal4 driver, which induces broad expression of (CCTG)720 from the mesoderm to muscle cells. The expression caused complete lethality in flies, allowing for the identification of compounds that could potentially rescue toxicity by influencing muscle development.

For the secondary screen, the Mef2-Gal4 driver, primarily expressed in mature muscle, was employed. This driver resulted in semi-lethality, providing an easily quantifiable phenotype for further analysis. Compounds with survival counts showing a 1.5-fold increase compared to DMSO-treated flies (>30–40, approximately the top 20%) were selected.

In total, 31 compounds were identified in the secondary screen that exhibited varying degrees of rescue effects against muscle toxicity. Among these, 16 compounds with target information available in the bioassay of PubChem Compound database were selected for subsequent analysis.” has been updated at line 410-428 of Materials and Methods,

Regarding “Description of DM1 Flies and Mammalian Models”, the description has been updated in the Materials Methods sections.

The DM1 fly strains has been updated at Reagents and Tools Table.

“The DM1 patient-derived lymphoblastoid cell lines were established after EBV transformation. This study was approved by the Ethics Committee of Center for Medical Genetics, the School of Life Sciences, Central South University, and informed consent was obtained on all participants (approval number: 2022-1-4).” has been updated at line 571-577 of Materials and Methods.

Referee #2 (Comments on Novelty/Model System for Author):

The manuscript describes identifying novel compounds in terms of mechanisms of action for "positive" modulation of the type of mutation defined for repeat expansion diseases. The work is focused on a *Drosophila* DM2 model. Although the results are auspicious as the manuscript is now presented, additional data is needed to finally achieve journal standards, indicating "demonstration of human disease relevance", here still not reached. In addition, the limitation to DM2, barely extended to DM1, and supported by the exciting first mechanistic results, it has also be implemented with new disease inclusion and new models, even for DM2, currently existing.

Not suited to a short report. This work needs to "grow" a provide solidier data in more diseases and models.

Referee #2 (Remarks for Author):

The authors have made a good effort to implement the reviewer request, but the final result is only partial responses, more effective for the writing structure than for the results interpretation and new data included. Issues still not addressed:

1. the stated "repeat stability" of TRE fly models is clear in the literature. The

explanation indicating where the contractions were observed during the fly development, and how they could be produced is poor and confusing at this moment (lines 228-254). Only introducing more questions, and still not responding why now instability (here contractions) can easily be observed when this not happened before.

Response:

In the DM2 drosophila model, Knockdown of TDP1/gkt could induce time-dependent contraction of large CCTG repeat via TOP1 in muscle cells. Due to methodological limitations, further study into the molecular mechanism underlying TDP1-mediated repeat contraction needs to be conducted in cell model. Possible mechanisms have been hypothesized in the paragraph 4 of Discussion. Addressing these hypotheses requires extensive investigations, probably including the following experiments.

1. Establishing DOX-inducible cell models expressing varying lengths of CCTG repeats to determine whether repeat contraction is transcription-dependent or DNA replication-dependent. 2. Defining the relationship between repeat length and R-loop formation, elucidating the role of R-loops in repeat contraction, and investigating how TDP1 knockdown and R-loops interact in this process. 3. Characterizing the type of DNA damage (single-strand or double-strand break) induced by the synergistic effects of TDP1 knockdown and R-loop accumulation, and identifying the downstream damage repair pathways and associated molecules.

2. the fly experiments (crossing) performed are complex. Additional information provided, only explaining that Mef2>CCTG flies were obtained, is still incomplete, since also siRNA and overexpression crosses were done. A figure with all the crosses and at what stage the contraction observed is highly recommended for not-in-fly experts.

Response:

The figure of crosses is shown below.

A $(CCTG)_{16}/Gmr-Gal4/+$, $Gmr-Gal4/+$; $(CCTG)_{720}/+$, $(CCTG)_{16}/+$; $Mef2-Gal4/+$, $+/+$; $(CCTG)_{720}/Mef2-Gal4$ cross follows the similar scheme

$$\begin{array}{c} \text{♀ } \frac{+}{+}; \frac{Gmr-Gal4}{Gmr-Gal4}; \frac{+}{+} \times \text{♂ } \frac{+}{+}; \frac{+}{+}; \frac{UAS-(CCTG)720}{UAS-(CCTG)720} \\ \downarrow \\ \frac{+}{+}; \frac{Gmr-Gal4}{+}; \frac{UAS-(CCTG)720}{+} \end{array}$$

B $Gmr-Gal4/+$; $(CCTG)_{720}/gkt$ mut, $Gmr-Gal4 > UAS-(CCTG)_{16} + \text{control RI}$, $Gmr-Gal4 > UAS-(CCTG)_{720} + \text{control RI}$, $Gmr-Gal4 > UAS-(CCTG)_{720} + gkt$ RI, $Mef2-Gal4 > UAS-(CCTG)_{16} + \text{control RI}$ cross follows the similar scheme

$$\begin{array}{c} \text{♀ } \frac{+}{+}; \frac{Sco}{Cyo}; \frac{TM3}{TM6B} \times \text{♂ } \frac{+}{+}; \frac{+}{+}; \frac{UAS-(CCTG)720}{UAS-(CCTG)720} \quad \text{♀ } \frac{+}{+}; \frac{Sco}{Cyo}; \frac{TM3}{TM6B} \times \text{♂ } \frac{+}{+}; \frac{Gmr-Gal4}{Gmr-Gal4}; \frac{+}{+} \\ \downarrow \qquad \qquad \qquad \downarrow \\ \text{♀ } \frac{+}{+}; \frac{Cyo}{+}; \frac{UAS-(CCTG)720}{TM6B} \qquad \qquad \times \qquad \qquad \text{♂ } \frac{+}{+}; \frac{Gmr-Gal4}{Sco}; \frac{TM6B}{+} \\ \downarrow \\ \text{♀ } \frac{+}{+}; \frac{Gmr-Gal4}{Cyo}; \frac{UAS-(CCTG)720}{TM3} \times \text{♂ } \frac{+}{+}; \frac{UAS-gkt RI}{UAS-gkt RI}; \frac{+}{+} \\ \downarrow \\ \frac{+}{+}; \frac{Gmr-Gal4}{UAS-gkt RI}; \frac{UAS-(CCTG)720}{TM3} \end{array}$$

C $Mef2-Gal4 > UAS-(CCTG)_{720} + \text{control RI}$, $Mef2-Gal4 > UAS-(CCTG)_{720} + gkt$ RI cross follows the similar scheme.

$$\begin{array}{c} \text{♀ } \frac{+}{+}; \frac{+}{+}; \frac{Mef2-Gal4}{Mef2-Gal4} \times \text{♂ } \frac{+}{+}; \frac{+}{+}; \frac{UAS-(CCTG)720}{UAS-(CCTG)720} \\ \downarrow \\ \text{♀ } \frac{+}{+}; \frac{+}{+}; \frac{UAS-(CCTG)720}{Mef2-Gal4} \times \text{♂ } \frac{+}{+}; \frac{Sco}{Cyo}; \frac{TM3}{TM6B} \\ \text{recombination} \downarrow \\ \text{♀ } \frac{+}{+}; \frac{Cyo}{+}; \frac{Mef2-Gal4, UAS-(CCTG)720}{TM6B} \times \text{♂ } \frac{+}{+}; \frac{UAS-gkt RI}{UAS-gkt RI}; \frac{+}{+} \\ \downarrow \\ \frac{+}{+}; \frac{UAS-gkt RI}{+}; \frac{Mef2-Gal4, UAS-(CCTG)720}{+} \end{array}$$

3. Authors included results with human DM1 cell lines from patients, and Hek293 transfected with CTG/CTGG repeats, but limited to see contraction. Results detecting correction of phenotypes in patient-derived DM1/DM2 cells lines (foci, mbn1 sequestration, miR23b expression, splicing, etc) is needed to assure the translatability of the compounds here indicated. Also, experiments performed with human cells, some done with one compound (not the good ones), and others with the two compounds chosen. Why? They should be done all with the compounds chosen.

Response:

The phenotypes (foci, MBNL1 sequestration, miR-23b expression, and splicing alterations) are primarily observed in muscle cells. In DM, skeletal muscle is one of the most affected tissues, and MBNL1 sequestration leads to splicing alterations of muscle-related genes. It is unavailable to detect these phenotypes in human DM1 patient-derived lymphoblastoid cells and HEK293 cells, as the expression levels of muscle-related genes are low or absent in these cell types.

The difference in compound treatments across cell types was due to their distinct biological characteristics. In HEK293 cells, rapid division results in frequent DNA replication, while our study suggests that TDP1 knockdown induces repeat contraction more effectively in non-dividing muscle cells. To enhance the effects of TDP1 siRNA, CPT, a compound with high cytotoxicity but strong TOP1-mediated DNA break formation, was included, thereby amplifying the effects of TDP1 knockdown. El-Khamisy et al. (2005) treated TDP1-mutant patient cells with CPT to enhance the consequences of TDP1 mutations (El-Khamisy *et al*, 2005).

The DM1 patient-derived lymphoblastoid cells are suspension cells with a slow division rate, allowing for long-term compound treatment. Therefore, only TDP1 inhibitors were used in these cells, without the addition of CPT.

4. Although the authors focused on DM2, the results obtained suggest to move, at least, to DMs for the manuscript. As it is now, it is not giving definitive responses of the final applicability.

Response:

We have added experiments on the effects of TDP1 inhibition on repeat length in DM1 *Drosophila*, DM1 patient-derived lymphoblastoid cells, and HEK293 cells expressing CTG/CCTG repeats. Expanding these findings to DMs requires extensive studies, including

1. Developing DM1 *Drosophila* models with varying CTG repeat lengths and assessing muscle degeneration, molecular pathology, and repeat contraction upon genetic and pharmacological TDP1 inhibition.
2. Using DM1 patient-derived iPSC-induced muscle fibers to evaluate the therapeutic effects of TDP1 knockdown through genetic editing (knockdown via gene editing or AAV delivery) and pharmacological inhibition. It involves optimizing dosage and delivery methods for TDP1 inhibitors.

3. Investigating TDP1 inhibition in DM1 mouse models through both genetic and pharmacological approaches, assessing improvements in motor function and disease phenotypes. Generation of DM2 mouse models and treatment of similar way.

Added the Limitation of the Study section:” This study highlights the potential of TDP1 inhibition in promoting repeat contraction. Further study is needed to fully elucidate the molecular mechanisms underlying repeat contraction. Validation in patient-derived muscle fibers and mouse models will be essential for clinical translation. Moreover, investigating whether TDP1 inhibition could have therapeutic effects in DM1 and even other repeat expansion disorders remains an important direction for future research.”

5. Protection (patent) of the data here described it is not well resolved by the authors.

Would be a pity these compounds are not well-protected if their promising activity is finally also observed in human situations.

Response:

We have already applied for a patent in China in April 2023 for TDP1 and its inhibitors as therapeutic targets for DM2, with the patent number CN202310462486.X.

6. Still not clear to this reviewer why they did not evaluate some of the compounds from the Drosophila screening. The authors mentioned the high concentration used in the fly screening...but this initially does not discard a good response in cells.

Response:

The target information for screening compounds was obtained from the bioassay of the PubChem Compound Database. In the bioassay of TDP1 inhibitors, compounds were tested on chicken DT40 cells, in which endogenous chicken TDP1 was knocked out and replaced with human TDP1, to observe cell survival. These compounds may directly or indirectly affect TDP1 function, meaning that some may not act directly on TDP1 itself but rather on downstream or parallel pathways, leading to similar phenotypic effects. To ensure specificity, the TDP1 inhibitors used in subsequent experiments were validated in both in vitro and in vivo assays, confirming their direct action on TDP1.

7. Still some repeat lengths and/or results in figures not well understood. For example, figure

EV5-E displays a "normal" repeat band close to 600 bp, or same figure EV5 in C-D, it is stated that a contraction occurring in DM2 situation, but it is very hard to see. In the same EV5-E figure, the expansion is now stated above the 2000 bp ladder band, and not very clear. Before stated around 1600 bp.

Response:

In patient-derived lymphoblastoid cells with 25/500 CTG repeats (Figure EV5-E, Agilent DNA chip gel image), the normal (CTG)₂₅ allele is 627 bp, while the expanded (CTG)₅₀₀ allele is around 2000 bp. The smear bands below 2000 bp are contracted CTG repeat fragments.

In HEK293 cells expressing (CTG)₄₀₀ and (CCTG)₄₀₀ repeats (Figure EV5-C, Agilent DNA chip gel image), the repeats are located within the pcDNA3.1 plasmid. The expanded (CTG)₄₀₀ is around 1300 bp, while the expanded (CCTG)₄₀₀ is around 1800 bp.

Reference

El-Khamisy SF, Saifi GM, Weinfeld M, Johansson F, Helleday T, Lupski JR, Caldecott KW (2005) Defective DNA single-strand break repair in spinocerebellar ataxia with axonal neuropathy-1. *Nature* 434: 108-113

Referee #3 (Comments on Novelty/Model System for Author):

It is unclear what the medical impact of this will be. It is probably correct that slowing down expansion rates could slow down disease progression in repeat mediated disorders. However, most of the strategies, including this one, involve affecting DNA replication and error correction, which is a slippery but known slope towards cancer.

Referee #3 (Remarks for Author):

The authors have done a good job of addressin concerns from the three reviewers and the paper is better for it.

However, one thought lingered and on thing occurred to me upon rereading this manuscript. The authors conclude that the TDP1 inhibitors are mediating their effects through impaired repairing repair mechanisms, leading to contractions of the repeat tract. However, this wasn't directly proven. It is correlative.

Response:

In the DM2 drosophila model, Knockdown of TDP1/gkt could induce time-dependent contraction of large CCTG repeat via TOP1 in muscle cells. Due to methodological limitations, further investigation into the mechanism underlying TDP1-mediated repeat contraction needs to be conducted in human cell model. Possible mechanisms have been hypothesized in the paragraph 4 of Discussion. Addressing these hypotheses requires extensive and complex investigations, including the following experiments.

Added the Limitation of the Study section:” This study highlights the potential of TDP1 inhibition in promoting repeat contraction. Further study is needed to fully elucidate the molecular mechanisms underlying repeat contraction. Validation in patient-derived muscle fibers and mouse models will be essential for clinical translation. Moreover, investigating whether TDP1 inhibition could have therapeutic effects in DM1 even other repeat expansion disorders remains an important direction for future research.”

The question I had was what happened to the various phenotypes (histologic and molecular (splicing, Foci size, etc.) when you inhibited TDP1 and TOP1. You showed that the repeat didn't shrink, but were silent about the phenotypic effects. Did you observe no rescue of phenotype or did you observe a rescue of phenotype despite no change in repeat size. If the latter, then it suggests that inhibition of TDP1 is having its

beneficial effects through other mechanisms. Please address.

Response:

In the TOP1+TDP1 double knockdown experiments, we primarily focused on repeat length changes between TDP1 knockdown alone and TDP1+TOP1 knockdown to determine the role of TOP1 in repeat contraction.

The phenotypes were not observed because TOP1 knockdown itself induces abnormalities in *Drosophila*. Previous studies have shown that TOP1 knockdown leads to growth, developmental defects, and neurodegeneration, as TOP1 functions as a nuclease that resolves supercoiled DNA structures, facilitating DNA replication and transcription (Song *et al*, 2007; Zhang *et al*, 2000). In repeat expansion disorders, repeat length is directly correlated with disease severity, making it the primary focus of our investigation.

Reference

Song J, Hu J, Tanouye M (2007) Seizure suppression by top1 mutations in *Drosophila*. *J Neurosci* 27: 2927-2937

Zhang CX, Chen AD, Gettel NJ, Hsieh TS (2000) Essential functions of DNA topoisomerase I in *Drosophila melanogaster*. *Dev Biol* 222: 27-40

5th Mar 2025

Dear Prof. Duan,

We are pleased to inform you that your manuscript is accepted for publication and is now being sent to our publisher to be included in the next available issue of EMBO Molecular Medicine.

Zeljko Durdevic
Senior Editor
EMBO Molecular Medicine
